# ONCE-FOR-ALL: SCALABLE SIMULTANEOUS FORECASTING VIA EQUILIBRIUM STATE ESTIMATION

## ABSTRACT

We introduce Equilibrium State Estimation (ESE), a novel paradigm for simultaneous prediction, where multiple interacting systems require separate yet coordinated forecasts. Such scenarios often arise in real-world such as economics and healthcare modeling. Unlike existing approaches that predict one system at a time, ESE forecasts all systems in a single pass. It first estimates the equilibrium state across systems, then generates holistic forecasts based on the difference between the current state and the estimated equilibrium. Extensive experiments on synthetic and real-world datasets, including currency exchange and COVID-19 spread modeling, demonstrate that ESE is at least as accurate as state-of-the-art (SOTA) methods while being significantly faster. In addition, ESE integrates seamlessly with conventional predictors, combining their accuracy with its exceptional efficiency and delivering a 10–70× speedup. With linear-time complexity, ESE scales far better than SOTA methods as the number of systems increases. Moreover, it remains accurate under diverse perturbations, establishing ESE as a fast, generalizable, robust, and scalable multi-prediction method. Source code and data are available at https://anonymous.4open.science/r/ESE-C339.

## 1 INTRODUCTION

The aim of this study is to establish a new prediction approach for scenarios involving multiple interacting systems, where each system influences and is influenced by others. Predicting each system individually is feasible but repetitive, costly, and may overlook the interactions among systems, especially as the number of systems increases. This raises a fundamental question: can all systems be predicted simultaneously in a unified manner? To address this, we propose a holistic prediction approach based on the concept of equilibrium, enabling a one-step, joint prediction across all systems.

Equilibrium states are commonly observed in the real-world, for instance isostatic equilibrium in mechanics Hemingway & Matsuyama (2017) and homeostasis in biology Hegyi et al. (2012). In economics, equilibrium plays a central role in modeling large market dynamics Cole & Tao (2016). Equilibrium is also used to model decision-making behaviors, for example Nash equilibrium in game theory Farina & Sandholm (2021), and regression equilibrium in market competition Ben-Porat & Tennenholtz (2019). Equilibrium-based models have been proposed to analyze socioeconomic impacts Nagurney & Salarpour (2021) for the COVID-19 pandemic, and to design optimal social distancing strategies Bairagi et al. (2020).

Equilibrium is not inherently predictive, rather, it describes a state in which all competing influences within a system are relatively balanced Daskalakis et al. (2009). When this balance is disturbed, the system begins to destabilize. For example, a system in thermal equilibrium becomes unstable when exposed to a heat source, triggering adjustments that may eventually lead to a new equilibrium. By viewing interacting systems as components of a larger super-system, we can estimate their collective equilibrium state. This estimation reveals the tendencies of change across systems. From this perspective, the future states of all member systems within the ensemble can be predicted jointly. Building on this hypothesis, we propose Equilibrium State Estimation (ESE), a novel multi-prediction model grounded in the analysis of equilibrium dynamics, in particular the direction of changes.

Equilibrium is naturally multi-dimensional and multifaceted, making it well-suited for handling multiple targets, in contrast to typical time-series approaches which focus on a single output. For example, consider an epidemic spreading across multiple regions within a large geographical area. Each region can be treated as an individual system, influenced by and influencing its neighboring

regions. Instead of predicting for each region, our approach is to estimate the equilibrium state of the entire area and predicts all regional states in a single run. Our ESE consists of two key components: (1) Equilibrium State Estimator, which infers the latent equilibrium across systems; (2) Predictor, which forecasts system states based on the equilibrium estimation. ESE can function independently or be integrated with existing methods, where the base model predicts overall trends, and ESE handles the distribution across systems for unified multi-prediction. **Our three key contributions** are summarized as below:

- We propose ESE, a novel mechanism for forecasting multiple systems via equilibrium state estimation. Unlike conventional time-series models, ESE predicts all systems in a single run. Its linear complexity makes it especially advantageous as the number of systems grows.
- We conduct extensive experiments on both synthetic and real-world datasets (exchange rate and COVID-19), under varying input lengths, prediction horizons, and granularities. Results show that ESE is not only fast and accurate, but also more flexible than SOTA methods.
- We demonstrate that ESE can be integrated with SOTA prediction models, enabling them to perform simultaneous multi-prediction with enhanced accuracy and significant speedup.

## 2 EQUILIBRIUM IN MULTIPLE SYSTEMS

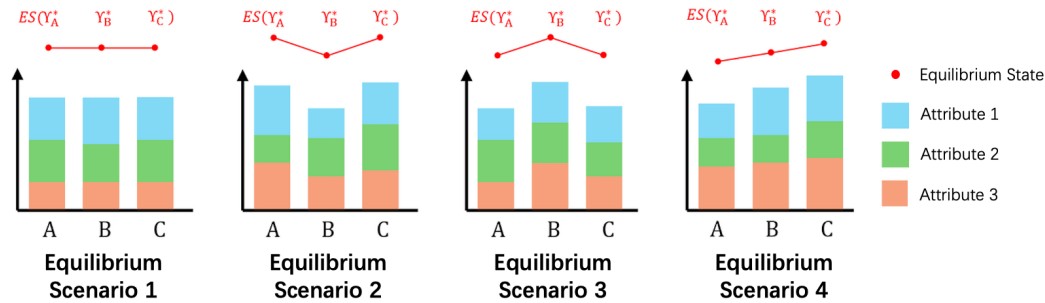

Figure 1: Illustration of a scenario consisting of three systems, A, B, and C, reaching equilibrium in four scenarios. Each system is characterized by the same set of three attributes. The red solid line above A, B, C in each scenario represents the corresponding equilibrium state ($\mathcal{ES}$). Each point on the line indicates the system's target value at $\mathcal{ES}$ - $\gamma_A^*, \gamma_B^*$, and $\gamma_C^*$, respectively [* for equilibrium].

**Equilibrium**  Equilibrium has seen limited adoption in numerical prediction, as this is not its conventional role. However, its potential in improving efficiency and enabling scalable learning is gaining recognition. For instance, it is used in decentralized learning for Markov games Foster et al. (2023). The deep equilibrium model integrates equilibrium into deep learning to enable memory-efficient training and inference, regardless of network depth Bai et al. (2019); Yang et al. (2023). This approach has also shown advantages in computer vision Bai et al. (2022; 2020); Graf et al. (2022). In our study, equilibrium refers to the balanced state among multiple interacting systems, as illustrated in Fig. 1. When changes occur within these systems, such as shifts in their attributes, the equilibrium adjusts accordingly. For example, when the economic and trade conditions of all countries remain stable, the exchange rates of their currencies tend to remain steady. However, if one country's economy grows or declines, the overall dynamics will change, eventually settling into a new equilibrium.

**Forecasting multiple systems**  It is to simultaneously forecast multiple target variables. Each system represents an individual predictive target that is associated with attribute variables. For example, in exchange rate prediction, a single system corresponds to one specific exchange rate. Our approach is to forecast multiple different exchange rates at the same time. Note, attribute variables are not always necessary in traditional time series forecasting tasks, such as ARIMA and VAR models, but essential in our approach. Moreover, variables in multi-variate time series pertain to a single underlying system, whereas our approach spans multiple systems. It is particularly important to distinguish this study from similar-looking prediction tasks, such as *multi-target*, *multi-variate*, and *multi-compartment*, which represent fundamentally different goals [1].

---

[1]See **Appendix A** for more detailed explanation of these similar-looking concepts.

**Definition 1:** An ensemble of multiple systems $\mathcal{MS}$ consists of $n$ systems, denoted as the set $s = [s_1, \ldots, s_n] \in \mathbb{R}^n$, where $n \in \mathbb{N}^+; n \geq 2$, and each $s_i$ is a distinct system.

This definition shows our approach require two or more systems. For simplicity of notation, we use $s_i$ to denote the target variable of System $s_i$. For example, in the context of epidemic forecasting, $s_i$ represents the number of daily new cases in System $s_i$, or region $i$. Similarly, we use $\mathcal{MS}$ to denote the ensemble of multiple systems, as well as the aggregate of the target variables across all systems. That is, $\mathcal{MS} = \sum_{i=1}^{n} s_i$, $n \geq 2$.

**Definition 2:** When the ensemble of multiple systems $\mathcal{MS}$ is viewed in its entirety, the change in proportion in one system complements the total changes in proportion from other systems.

$$\triangle\gamma_i = -\sum_{j}^{n} \triangle\gamma_j \,, n \geq 2, i \in [1, n], j \in [1, n], i \neq j, \tag{1}$$

This can be expressed as in Eq. 1, where $\gamma_i$ is the proportion of system $s_i$ to $\mathcal{MS}$, as $\gamma_i = s_i/\mathcal{MS}$. Eq. 1 is to simplify and unify the relationships among systems in $\mathcal{MS}$ through proportional changes in their target values, making the model independent of the specific type or range of target values. Specifically, a change in one system corresponds to the net change across all other systems in proportion. Note that the members of the ensemble should be consistent. No system shall be removed from or added to $\mathcal{MS}$. Therefore, constraints must be imposed on ESE.

**Constraints 1 & 2:** (1) When modeling the ensemble of multiple systems, all systems are considered, and none are left out, as specified in Eq. 2. Hence the aggregate of $\gamma_i$ is always **1**. By "considering all elements", we do not mean that the ensemble must model the entire scenario. For example, although COVID-19 is global, city-level forecasting does not require modeling the entire world. However, when modeling the entire city as an ensemble of suburbs, all of its suburbs should be included in $\mathcal{MS}$. (2) The sum of proportional changes of all systems is **zero**, as expressed in Eq. 3. This is derived from Eq. 1, as $\triangle\gamma_i + \sum_{j=1}^{n} \triangle\gamma_j = 0 \,; i \neq j$.

$$\sum_{i=1}^{n} \gamma_i = 1 \,, n \geq 2, \tag{2} \qquad\qquad \sum_{i=1}^{n} \triangle\gamma_i = 0 \,, n \geq 2. \tag{3}$$

**Definition 3:** Every system $s_{1:n}$ within $\mathcal{MS}$ has identical attribute set $\mathcal{A}$. Collectively, these attributes determine the state of $\mathcal{MS}$. The attribute values of $\mathcal{A}$ for a system $s_i$ can be expressed as $\mathcal{A}_i = \{\alpha_{i,1}, \alpha_{i,2}, \ldots, \alpha_{i,m}\}$, where $m$ is the number of attributes.

Note that these three definitions and two constraints are introduced for the mathematical formulation of the prediction task, not for the task itself, and therefore do not limit its applicability.

**Equilibrium State** One core idea in Nash equilibrium Kreps (1989) is the estimation of a system's state by analyzing the internal competition among players. Leveraging this concept, we estimate the equilibrium state by examining the interactions among "*internal*" systems, specifically, through the analysis of their attributes. At equilibrium, the attribute sets of the $n$ systems, $\mathcal{A}_{1:n}^*$, maximize the utility : $U(\mathcal{A}_1^*, \mathcal{A}_2^*, \ldots, \mathcal{A}_n^*) \geq \cdots \geq U(\mathcal{A}_1, \mathcal{A}_2 \ldots, \mathcal{A}_n)$ [2]. The process is detailed in Section 3.1.

## 3 EQUILIBRIUM STATE ESTIMATION FOR PREDICTION

The above descriptions have not incorporated any temporal elements. Here, we introduce time dependency, as our proposed Equilibrium State Estimation (ESE) is inherently dynamic method that operates without assuming a static equilibrium, resembling time-varying models Gao et al. (2024). ESE comprises two key components: (1) Equilibrium State Estimator, for estimating the target variable values at equilibrium, and (2) Predictor, for forecasting the future states of the systems based on the distance of each system from equilibrium. A simplified illustration of this prediction process is provided in Fig. 2.

---

[2]See **Appendix B and C** for more detailed explanation of equilibrium state equation.

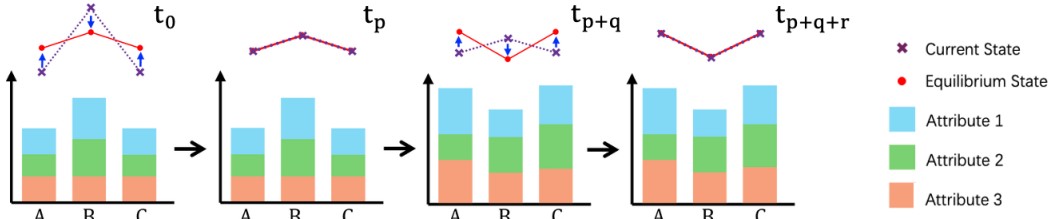

Figure 2: Illustration of the transition between states over time in $\mathcal{MS}$. Each purple dotted line represents the actual state $[\gamma_A, \gamma_B, \gamma_C]$ at a given time point. The red lines, consistent with those in Fig. 1, represent the corresponding equilibrium state $\mathcal{ES}$: $[\gamma_A^*, \gamma_B^*, \gamma_C^*]$. At time $t_0$, the ensemble $\mathcal{MS}$ is out of equilibrium, and the target values begin to move toward the equilibrium state. When a change occurs at $t_{p+q}$, the systems reconverge toward a new equilibrium by $t_{p+q+r}$. The blue arrows illustrate the predicted direction of state transitions, inferred by comparing the current state with the estimated equilibrium state under the current attribute configuration of Systems A, B and C.

### 3.1 Estimating the Equilibrium States

**System State** The current state of $\mathcal{MS}$ at time $t$, $\mathcal{ST}_t = [\gamma_{1,t}, \ldots, \gamma_{n,t}] \in \mathbb{R}^n$, $n \in \mathbb{N}^+$.

**Equilibrium State** The equilibrium state for $\mathcal{MS}$ at time $t$ can be expressed as $\mathcal{ES}_t = [\gamma_{1,t}^*, \ldots, \gamma_{n,t}^*] \in \mathbb{R}^n$, $n \in \mathbb{N}^+$. In most cases, $\mathcal{MS}$ is not in equilibrium, e.g. $\mathcal{ST}_t \neq \mathcal{ES}_t$.

By comparing $\mathcal{ES}_t$ and $\mathcal{ST}_t$, we can assess whether the system is in equilibrium, and, if not, estimate the magnitude of deviation from equilibrium. To estimate $\mathcal{ES}_t$, we begin by normalizing each attribute $\alpha_{i,k}(t)$ of all systems at time $t$, resulting in $\alpha'_{i,k}(t)$, as defined in Eq. 4:

$$\alpha'_{i,k}(t) = \frac{\alpha_{i,k}(t) - \overline{\alpha_k(t)}}{U[\alpha_k(t)] - L[\alpha_k(t)]}, \alpha'_{i,k}(t) \in \mathbb{R}^{m \times n}, i \in [1, n], k \in [1, m], \tag{4}$$

where $n$ denotes the number of systems, and $m$ is the number of attributes per system. For the $k$th attribute at time $t$, $\overline{\alpha_k(t)}$, $U[\alpha_k(t)]$ and $L[\alpha_k(t)]$ represent the average, upper bound, and lower bound values across all systems, respectively. U and L represent theoretical extremes, not the observed range in the dataset. For instance, the theoretical minimum for a region's population is 0, whereas its practical minimum in our data is over 100. With $\alpha'_{i,k}(t) \in \mathbb{R}^{m \times n}$ values available for all attributes of all systems, we can obtain the initial equilibrium parameters $\mathcal{ES}_t^{[0]}$, as shown in Eq. 5:

$$\mathcal{ES}_t^{[0]} = \frac{1 + \frac{1}{m} \sum_{k=1}^m \psi_{k,t} \alpha'_{1:n,k}(t)}{n}, \tag{5}$$

where $\psi_k$ is the influence of $\alpha'_k$, e.g. the $k$th normalized attribute values across all systems, obtained through MLE Wooldridge (2016), as Eq. 6. $A'_t \in \mathbb{R}^{m \times n}$ denotes the matrix containing all $\alpha'_{i,k}(t)$. $\Psi_t \in \mathbb{R}^m$ is the set of coefficients, containing $m$ elements $\psi_t$. $\sigma^2$ is the variance of the residuals, representing the magnitude of variation that the model cannot capture.

$$\ell(\Psi_t, \sigma_t^2) = -\frac{n}{2} \log(2\pi\sigma_t^2) - \frac{1}{2\sigma_t^2}(\mathcal{S}_t - A'_t \Psi_t)^\top (\mathcal{S}_t - A'_t \Psi_t) \tag{6}$$

The equilibrium state cannot be determined from attributes alone. To work out the equilibrium, an n-element correction vector, denoted as $\mathcal{L} = [l_1, \ldots, l_n]$ is introduced to guide the convergence. Each element of $\mathcal{L}$ maps to a target value of $\mathcal{ES}$, e.g. the target of one system. The values will be updated at every epoch of the estimation process of $\mathcal{ES}$, which is detailed in Algorithm 1. The progression of $\mathcal{L}$ over epochs, e.g. from epoch $e$ to $e+1$ can be expressed as $\mathcal{L}^{[e+1]} = f(\mathcal{L}^{[e]})$, $e \in [0 : E)$. $E$ is the maximum number of epochs. Note that the process will not converge if all $\psi_t$ values are set to zero, effectively disabling the influence of all attributes in Eq. 5. As illustrated in Fig. 2, attributes are fundamental to the ESE framework. The model relies on the pattern of attributes to estimate pattern of targets at the equilibrium state; without attributes, the estimation procedure would fail to converge.

---

**Algorithm 1** The Equilibrium Estimation Process of $\mathcal{ES}_t$

---

**Input:** $\mathcal{ST}_{(t-p):t} = \gamma_{1:n,(t-p):t},\ \mathcal{ES}_t^{[0]}$    /* $p$ is the sampling length from the past. */
**Output:** $\mathcal{ES}_t$
1:  $e \leftarrow 0;\ \mathcal{L}^{[0]} \leftarrow 1_n^T$    /* Initialization from Epoch 0, all $n$ elements of $\mathcal{L}^{[0]}$ set to 1. */
2:  **while** $\mathcal{ST}_{(t-p):t} = \gamma_{1:n,(t-p):t}$ and $\mathcal{ES}_t^{[e]}$ are not cointegrated **and** $e < E$ **do**
3:    **for** $t'$ in $[(t-p), t]$ **do**
4:      $\mathcal{L}^{[e]} \leftarrow (\mathcal{ES}_t^{[e]} - \mathcal{ST}_{t'} + \mathcal{L}^{[e]}) \cdot \lambda$    /* $\lambda$, the damping coefficient, set to 0.5. */
5:    **end for**
6:    $\mathcal{ES}_t^{[e]} \leftarrow \mathcal{ES}_t^{[e]} - (\mathcal{L}^{[e]} \cdot \lambda)$
7:    $e \leftarrow e + 1$
8:  **end while**

---

The cointegration test is used as a convergence criterion in Algorithm 1, which is a common statistical method for assessing the existence of a long-run equilibrium Abadir (2004); Enders & Siklos (2001). Algorithm 1 iteratively adjusts the estimated equilibrium $\mathcal{ES}_t$ until it establishes a stable, long-run statistical relationship (cointegration) with the recent historical data $\mathcal{ST}_{(t-p):t}$. In our study, $E = \infty$. A finite E can be set if an upper time limit is needed.

In this study, we assume that is true, if $\mathcal{ST}_{(t-p):t}$ and $\mathcal{ES}_t$ are cointegrated. Otherwise, $\mathcal{ES}_t$ will keep converging until cointegration is reached. The convergence analysis and the cointegration equations are in **Appendix D**, which shows the progression of p-values from the cointegration test at each epoch. Once the p-value drops below $0.05$, the long-run equilibrium is considered true. The process will then terminate and output the estimated equilibrium state $\mathcal{ES}_t$. From a utility standpoint, the estimation process is to maximize confidence in the cointegration between $\mathcal{ST}_{(t-p):t}$ and $\mathcal{ES}_t$, until the confidence reaches a level above 0.95. ESE does not attempt to drive the system toward true equilibrium state. Instead, it estimates what the system would be if it were in equilibrium, enabling predictions based on the deviation of the current state from this estimated equilibrium. The convergence process in ESE is purely part of the estimation procedure and does not suggest that the actual system moves toward equilibrium. It should be emphasized that the notion of equilibrium in ESE refers specifically to a statistical equilibrium state, not a game-theoretic equilibrium. See Appendix O for the detail.

## 3.2 PREDICTOR

Based on the estimated equilibrium $\mathcal{ES}_t$, the prediction for all systems can be defined as in Eq. 7.

$$\widehat{s_{1:n,t+h}} = \theta_{t+h} \cdot \mathcal{MS}_t \cdot \mathcal{ES}_t + \varepsilon_{t+h},\ \varepsilon_{t+h} \sim N(0, \sigma^2), \tag{7}$$

where $h$ is the prediction horizon, $\mathcal{MS}_t$ is the total target value of the $\mathcal{MS}$ at time $t$. The parameter $\theta_{t+h}$ is estimated by maximizing the log-likelihood function of a linear autoregressive model for $\mathcal{MS}$. The last part, $\varepsilon_{t+q}$ is the residual. The parameter $\theta_{t+h}$ captures the overall trend of $\mathcal{MS}$ projected $h$ steps into the future. A value of $\theta_{t+h} = 1$ indicates no change, while $\theta_{t+h} > 1$ suggests an increasing trend, and $\theta_{t+h} < 1$ implies a decreasing trend. Note that it is possible for $\mathcal{MS}$ to be out of equilibrium even when $\theta_{t+h} = 1$, as deviations among individual systems may complement or cancel each other out, resulting in no net change in the aggregate trend (See **Appendix E** for the detail of proof). Conversely, a $\mathcal{MS}_t$ in equilibrium may exhibit $\theta_{t+h} \neq 1$ if all constituent systems change in sync. In such cases, the system maintains internal balance despite experiencing a uniform upward or downward trend. Detailed explanation of Eq. 7, especially the calculation of $\theta_t$, is in **Appendix F**.

The proportions of a multiple system ensemble are stable, even as its aggregate level fluctuates. This leads to a two-stage procedure: first, we obtain $\widehat{\mathcal{MS}_{t+h}}$, which is the overall system forecast; This value is then disaggregated into forecasts for individual systems using the fixed proportions from $\mathcal{ES}_t$. Therefore, an important feature of ESE is that any prediction models can be integrated[3], e.g. LSTM, SCINet etc. When using other tools with ESE, we only need the simplified version as shown in Eq. 8:

---

[3]Certain multi-variate methods (e.g., VAR) are unsuitable for direct integration due to their inapplicability to univariate forecasting as per Eq. 8. However, they can be combined indirectly as detailed in **Appendix Q**.

$$\widehat{s_{1:n,t+h}} = \widehat{\mathcal{MS}_{t+h}} \mathcal{ES}_t, \tag{8}$$

where $\widehat{\mathcal{MS}_{t+h}}$ is the predicted overall value of $\mathcal{MS}$ for time $t+h$, obtained from an external prediction model. In this scenario, the external model is responsible for predicting the overall trend, while ESE works out the internal distribution across individual systems.

## 4 EXPERIMENTS

Our validation involves two components: synthetic data and real-world datasets, including currency exchange rates and COVID-19 spreading. They will be made publicly available, as there is currently no benchmark designed for this type of forecasting involving multiple systems. Existing time-series benchmarks are unsuitable because they address only single-system settings and do not consider targets across multiple homogeneous systems. Moreover, they typically lack the attribute data needed to model system-level interactions as outlined above.

A collection of SOTA forecasting methods is used as our baseline set. Unlike ESE, they do not leverage attribute data because they are time-series–only methods. Even when attribute data is included, the impact on their predictive performance is rather limited (see Section. 5). In the following experiments, a 90:10 train-test split is used for models that require training.

### 4.1 SYNTHETIC DATA

Three synthetic datasets (detailed in **Appendix G**) are used for the comparison with six SOTA prediction methods: ARIMA ArunKumar et al. (2021), LSTM Feng et al. (2022), DLinear Zeng et al. (2023), Informer Zhou et al. (2021), DeepAR Le Guen & Thome (2020), and PatchTST[4]Nie et al. (2023). Table 1 provides a snapshot of the results for a single configuration, while the full results, spanning multiple input lengths, prediction horizons, and system sizes, are in **Appendix H.1** for prediction accuracy (Tables 7, 8, and 9) and **Appendix H.2** for computational cost (Tables 10, 11, and 12). Table 1 aligns with the broader trends observed in these extended results. Prediction performance is measured using average RMSE and MAE, while computational cost is in minutes [5]. Each SOTA method is estimated both independently ("No ESE") and when combined with ESE ("With ESE").

We can observe that **(1)** ESE alone is consistently competitive in accuracy and never the worst; **(2)** When combined with ESE, all predictors either improve or maintain their performance; **(3)** The lowest error in each column is always achieved

Table 1: Comparison on a synthetic dataset with 10 systems with 20 input steps and 1 step horizon. Results involving ESE are shown in light purple. The best results in each column are highlighted in bold with a light red background.

| Models | | RMSE | MAE | Cost (mins) |
|---|---|---|---|---|
| ESE | – | 0.248 | 0.228 | 0.23 |
| ARIMA | No ESE | 0.249 | 0.243 | **0.09** |
| | With ESE | 0.247 | 0.243 | 0.24 |
| LSTM | No ESE | 0.263 | 0.216 | 2.47 |
| | With ESE | 0.263 | 0.212 | 0.48 |
| Dlinear | No ESE | 0.256 | 0.221 | 2.93 |
| | With ESE | 0.264 | 0.221 | 0.52 |
| Informer | No ESE | 0.244 | 0.236 | 1.69 |
| | With ESE | **0.241** | 0.232 | 0.40 |
| DeepAR | No ESE | 0.271 | 0.218 | 2.29 |
| | With ESE | 0.271 | **0.210** | 0.53 |
| PatchTST | No ESE | 0.263 | 0.224 | 1.84 |
| | With ESE | 0.265 | 0.224 | 0.41 |

either by ESE alone or by a SOTA predictor augmented with ESE; **(4)** ESE incurs significantly lower computational cost than other methods, except ARIMA, which requires no training, hence is fast on small datasets, but not on larger ones; **(5)** When paired with ESE, the cost of SOTA predictors can be reduced significantly, by a factor of 2-10, excluding ARIMA; **(6)** The runtime increases with the number of systems estimated.

---

[4]All PatchTST models used in this study are PatchTST/64.

[5]All computational costs are measured on a system with an AMD Ryzen 9 7950X 16-Core 4.50 GHz CPU, 64 GB RAM, and an NVIDIA 4090 GPU with 24 GB memory.

## 4.2 Read-world Data

The economics and healthcare domains are highly representative of the forecasting settings defined in Section 2. Accordingly, currency exchange rates and COVID-19 spread are used.

**Currency Exchange Rates**  An exchange rate reflects the value of one country's currency relative to another Bodie & Kane (2020). As a classic form of time series data, it is commonly used as a benchmark in prediction studies. Here, we use 16 exchange rates relative to the USD, representing 16 non-USD currencies from the G20, which comprises the world's most significant economies. These currencies exhibit mutual influence, while the impact of economies outside the G20 is negligible. Hence, these exchange rate scenarios can be modeled in our ESE, where each country represents a system with its exchange rate as the prediction target. We collected daily exchange rate data relative to the USD over a five-year period, from November 11, 2019 to October 31, 2024. Moreover, macroeconomic data for the G20 countries over the same period were also gathered (more in **Appendix I**).

Twelve SOTA predictors are included in this comparison: the six used for the synthetic data, along with VAR Hyndman & Athanasopoulos (2018), NLinear Zeng et al. (2023), FiLM Zhou et al. (2022a), SCINet Liu et al. (2022a), KVAE Tang & Matteson (2021), and TPGNN Liu et al. (2022b). Similar to Table 1, Table 2 provides a snapshot of the results under a single configuration: 100 input steps and a 1-step prediction horizon. The full results on performance accuracy are in **Appendix J.1** (Table 16 and 17), while the full results on computational cost are in **Appendix J.2** (Table 18). Given the substantial variation in exchange rate magnitudes, for example, a more than 20,000-fold difference between the Indonesian Rupiah (IDR) and the British Pound (GBP), we report both raw and normalized metrics: RMSE and MAE, as well as their normalized counterparts, RMSE$^*$ and MAE$^*$. The normalization (detailed in **Appendix K**) mitigates scale bias in the estimation and ensures fair comparison across currencies. Overall, the observations from synthetic data hold, particularly: **(1)** ESE alone is

Table 2: Comparison on currency exchange rate prediction with 12 SOTA methods, using 100 input steps and 1-step horizon across 16 currencies relative to the USD. RMSE$^*$ and MAE$^*$ denote normalized values, scaled by a factor of 100.

| Models | | RMSE | MAE | RMSE$^*$ | MAE$^*$ | Cost (mins) |
|---|---|---|---|---|---|---|
| ESE | – | 6.01 | 5.520 | 1.183 | 0.600 | **0.22** |
| VAR | – | 6.823 | 5.917 | 1.248 | 0.653 | – |
| ARIMA | No ESE | 6.274 | 5.681 | 1.280 | 0.629 | 0.26 |
| | With ESE | 5.709 | 5.291 | 1.197 | 0.604 | 0.24 |
| LSTM | No ESE | 6.168 | 5.668 | 1.218 | 0.604 | 7.65 |
| | With ESE | 5.621 | 5.257 | 1.190 | 0.601 | 0.70 |
| Dlinear | No ESE | 5.878 | 5.405 | 1.212 | 0.603 | 8.27 |
| | With ESE | **5.461** | **5.102** | 1.174 | **0.586** | 0.74 |
| Nlinear | No ESE | 5.963 | 5.459 | 1.214 | 0.608 | 8.83 |
| | With ESE | 5.518 | 5.171 | 1.179 | 0.593 | 0.77 |
| Informer | No ESE | 6.012 | 5.585 | 1.219 | 0.608 | 5.08 |
| | With ESE | 5.563 | 5.204 | 1.182 | 0.595 | 0.54 |
| FiLM | No ESE | 5.929 | 5.544 | 1.218 | 0.612 | 9.41 |
| | With ESE | 5.508 | 5.219 | 1.179 | 0.597 | 0.81 |
| SCINet | No ESE | 5.899 | 5.502 | 1.184 | 0.603 | 11.42 |
| | With ESE | 5.481 | 5.148 | **1.174** | 0.590 | 0.93 |
| DeepAR | No ESE | 6.078 | 5.606 | 1.219 | 0.610 | 7.15 |
| | With ESE | 5.622 | 5.287 | 1.189 | 0.604 | 0.67 |
| KVAE | No ESE | 6.054 | 5.591 | 1.204 | 0.604 | 5.83 |
| | With ESE | 5.538 | 5.192 | 1.182 | 0.595 | 0.58 |
| TPGNN | No ESE | 6.088 | 5.633 | 1.182 | 0.601 | 8.37 |
| | With ESE | 5.512 | 5.183 | 1.177 | 0.594 | 0.74 |
| PatchTST | No ESE | 6.039 | 5.568 | 1.209 | 0.607 | 5.55 |
| | With ESE | 5.546 | 5.203 | 1.181 | 0.595 | 0.57 |

a top-tier predictor, if not the best, in most cases, especially for longer input lengths; **(2)** ESE consistently improves the performance for most SOTA predictors[6]; **(3)** ESE substantially reduces computational cost. For example, when combined with SCINet, it reduces runtime by more than an order of magnitude. Note: VAR can model multiple systems jointly based on cross-system correlations and is therefore not compatible with ESE. Moreover, because VAR requires manual parameter tuning, its computational cost is excluded from the comparison.

**COVID-19 Spread**  COVID-19 is a viral epidemic that has had profound impacts on healthcare systems, economies, and social dynamics worldwide. Prediction of its spreading or new cases across

---

[6]The performance improvements presented in Table 2, e.g. "With ESE" vs "No ESE", are all statistically significant, with only two exceptions: PatchTST at MAE$^*$ and FiLM at RMSE$^*$.

Table 3: Comparison on COVID-19 data with 12 SOTA methods on prediction accuracy and computational cost, using input of 100 steps and 1-step horizon for 20, 79, 320 systems respectively.

| Models | | 20 Regions | | | 79 Cities | | | 320 Suburbs | | |
|---|---|---|---|---|---|---|---|---|---|---|
| | | RMSE | MAE | Cost (mins) | RMSE | MAE | Cost (mins) | RMSE | MAE | Cost (mins) |
| ESE | – | 84.54 | 83.91 | 1.31 | 55.54 | 50.85 | 2.10 | 4.83 | 4.58 | **2.23** |
| VAR | – | 95.79 | 90.86 | – | 90.85 | 81.74 | – | 8.07 | 7.68 | – |
| ARIMA | No ESE | 79.70 | 79.59 | **0.33** | 77.72 | 67.41 | **1.31** | 7.44 | 6.24 | 5.07 |
| | With ESE | 75.31 | 73.64 | 1.33 | 54.15 | 50.65 | 2.12 | 4.74 | 4.41 | 2.30 |
| LSTM | No ESE | 78.45 | 74.96 | 9.23 | 61.42 | 57.26 | 39.96 | 6.23 | 5.47 | 149.26 |
| | With ESE | 72.79 | 69.47 | 1.80 | 55.42 | 49.93 | 2.59 | 4.68 | 4.37 | 2.72 |
| Dlinear | No ESE | 79.47 | 72.41 | 10.40 | 56.16 | 53.95 | 40.98 | 5.93 | 5.10 | 163.57 |
| | With ESE | 73.61 | 66.94 | 1.86 | 56.14 | 50.12 | 2.68 | 4.51 | 4.34 | 2.87 |
| Nlinear | No ESE | 74.96 | 70.77 | 11.56 | 55.62 | 52.23 | 45.67 | 5.47 | 4.99 | 160.09 |
| | With ESE | 71.64 | 69.98 | 1.87 | 55.14 | 51.33 | 2.70 | 4.47 | 4.18 | 2.86 |
| Informer | No ESE | 76.84 | 71.79 | 5.95 | 62.01 | 60.57 | 26.97 | 6.05 | 5.80 | 100.08 |
| | With ESE | 73.43 | 71.48 | 1.66 | 55.48 | 49.23 | 2.40 | 4.60 | 4.37 | 2.62 |
| FiLM | No ESE | 76.84 | 73.85 | 11.70 | 56.21 | 51.34 | 48.55 | 5.74 | 4.96 | 181.06 |
| | With ESE | 71.68 | 69.15 | 1.86 | 5.49 | 4.90 | 2.74 | **4.36** | **3.96** | 2.84 |
| SCINet | No ESE | 79.75 | 69.94 | 14.18 | 55.86 | 53.56 | 62.27 | 5.94 | 5.59 | 206.06 |
| | With ESE | 71.84 | 69.71 | 2.04 | 54.04 | 50.34 | 2.82 | 4.46 | 4.24 | 2.94 |
| DeepAR | No ESE | 83.41 | 74.65 | 9.22 | 56.34 | 54.32 | 37.25 | 6.29 | 5.04 | 130.67 |
| | With ESE | 74.82 | 72.91 | 1.77 | **50.99** | 50.96 | 2.52 | 4.79 | 4.46 | 2.73 |
| KVAE | No ESE | 70.85 | 64.44 | 6.80 | 53.12 | 52.71 | 32.41 | 5.29 | 4.94 | 109.62 |
| | With ESE | 71.94 | 68.41 | 1.67 | 52.13 | 51.24 | 2.50 | 4.76 | 4.26 | 2.63 |
| TPGNN | No ESE | 78.37 | 69.86 | 10.58 | 59.22 | 58.13 | 42.92 | 6.06 | 5.41 | 158.84 |
| | With ESE | 72.24 | 70.64 | 1.81 | 56.95 | 54.99 | 2.65 | 4.68 | 4.24 | 2.86 |
| PatchTST | No ESE | **70.01** | **63.52** | 6.92 | 56.44 | 52.41 | 27.70 | 5.11 | 4.43 | 109.89 |
| | With ESE | 70.74 | 66.97 | 1.79 | 53.71 | **47.25** | 2.46 | 4.51 | 4.36 | 2.66 |

regions is obviously critical. In this scenario, each region can be viewed as a system, with the number of its new cases serving as its prediction target. We collected daily cases from the state[7] government of Victoria, Australia, ranging from Jan. 1, 2022, to Sep. 16, 2022 [8]. During the pandemic, the Victorian government reported daily epidemic data for all its 79 municipalities, making the dataset one of the most comprehensive regional-level data in the world, hence used in this study. To estimate ESE's performance across different levels of granularity, we aggregated the 79 municipalities into 20 larger regions and also partitioned them into 320 smaller regions based on postcodes. At the level of 320 regions, the only attributes are population and band (See details in **Appendix L**).

The same twelve SOTA predictors are used in this part. Table 3 shows their results across three levels of granularity, 20 regions, 79 regions, and 320 regions. Full comparisons of prediction accuracy are in **Appendix M.1**, while computational costs are in **Appendix M.3**. Overall, the observations are consistent with those from the exchange rate experiments: **(1)** ESE alone outperforms other predictors in many cases, particularly when predicting all 320 regions; **(2)** ESE consistently boosts the performance of other predictors, with the best results in most columns achieved either by ESE alone or by combining with ESE, except for the 20-region setting, where PatchTST performs best; **(3)** ESE significantly reduces costs for all methods when integrated. When applied to FiLM and SCINet on the 320-region dataset, ESE enables speedups exceeding 70×. Also, ESE demonstrates strong scalability with increasing region count, while most other methods either degrade in performance or show limited improvement. Another notable strength of ESE is its ability to effectively handle large input lengths: the lowest RMSE values for inputs longer than 50 steps are predominantly achieved by ESE or SOTA methods augmented with our ESE.

## 5 ANALYSIS AND DISCUSSION

**Computational Complexity** As shown in Line 3 of Algorithm 1, the number of iterations is proportional to the time step $p$, indicating its linear time complexity. Specifically, its computational

---

[7]We use "state" for two unrelated concepts: the condition of a system, and a constituent unit of a nation.

[8]After Sep/16/22, data are no longer published daily but weekly, making it unsuitable for this study.

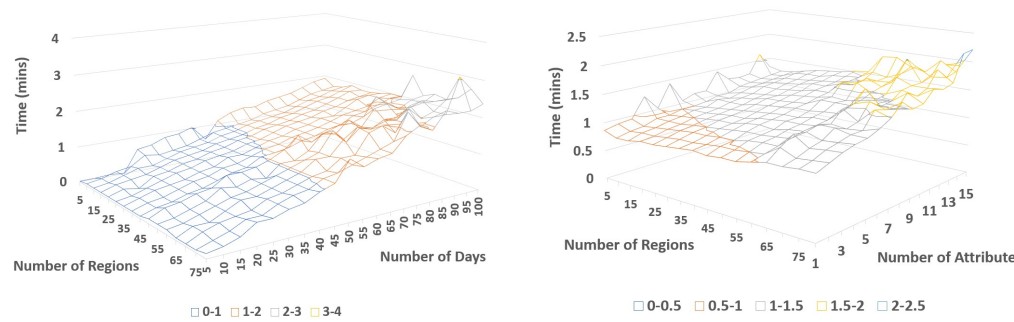

Figure 3: Complexity analysis of ESE. (Left): ESE's cost relative to the number of regions and the number of days for COVID-19 data (with 9 attributes). (Right): ESE's cost relative to the number of regions and the number of attributes for COVID-19 data (input size = 150).

cost scales linearly with the number of systems, the number of attributes, and the time steps. This is consistent with our analysis using the COVID-19 data, presented in Fig.3. In the left figure, the X-axis represents the number of regions (ranging from 1 to 79), while the Y-axis denotes the number of input steps (ranging from 5 to 100 days). Data points are grouped into four color-coded bands to better visualize the trends. A similar linear pattern is evident on the right side of Fig.3, showing a linear increase in cost with respect to both the number of regions and the number of attributes, making the forecasting of multiple systems scalable.

**Robustness**  ESE simultaneously predicts all systems in $\mathcal{MS}$, making it robust to localized perturbations, as variations in a single system have relatively low impact on the prediction due to its small proportion in the entire ensemble of systems. Table 4 presents a robustness analysis using the exchange rate data, where random noise of 5%, 10%, and 20% was added to 1, 2, and 5 currencies, respectively. As expected, performance degrades with increasing noise levels and the number of perturbed systems. However, the degradation is marginal when compared to the baseline performance without noise, normalized RMSE* and MAE* of **1.18** and **0.60**, respectively (as shown in Table 2).

Table 4: Robustness analysis of ESE

| Noise | 1 | 2 | 5 | systems |
|-------|------|------|------|---------|
| 5%  | 1.18 | 1.20 | 1.25 | RMSE* |
|     | 0.64 | 0.63 | 0.68 | MAE* |
| 10% | 1.25 | 1.24 | 1.28 | RMSE* |
|     | 0.60 | 0.66 | 0.75 | MAE* |
| 20% | 1.25 | 1.32 | 1.46 | RMSE* |
|     | 0.63 | 0.74 | 1.08 | MAE* |

**Multi-variate Prediction**  Most of the twelve predictors are multi-variate, capable of predicting multiple targets simultaneously. Therefore, the above tasks can be treated as a form of multi-variate prediction. Table 5 presents the results on exchange rate data using SCINet, a top predictor for this task (see

Table 5: ESE vs. multi-variate prediction - analysis using currency exchange rate data w/ and w/o attributes

| | RMSE | MAE | Time (sec) |
|---|------|------|------------|
| ESE alone | 6.08 | 5.52 | 13.2 |
| SCINet + ESE | 5.48 | 5.15 | 55.8 |
| SCINet (multiple predictions combined) | 5.90 | 5.50 | 685.2 |
| SCINet (multi-variate, without attributes) | 98.44 | 21.18 | 121.2 |
| SCINet (multi-variate, with attributes) | 96.23 | 18.99 | 129.5 |

Table 2), to further illustrate the advantages of ESE. When applied repeatedly to predict one system at a time, SCINet achieves high accuracy but also incurs a high cost. Predicting all systems jointly, without using attribute data, reduces its runtime from 685.2 to 121.2 sec, but at the expense of accuracy. Adding attribute data (bottom row) results in only a minor improvement.

In contrast, ESE delivers high accuracy and efficiency, whether used alone or in combination with SCINet. Similarly, this is observed with other predictors, such as PatchTST and Informer.

**Analysis of Completeness** As discussed in *Constraints 1 & 2*, ESE does not require modeling the entire scenario. Table 6 shows ESE prediction for four currencies (CAD, GBP, EUR, and JPY) at three evaluation points (Apr. 22, May 17, Jun. 14, 2024), when modeling only G7 countries. The G7 results are reasonable, but the G20 model yields forecasts closer

Table 6: Analysis on $\mathcal{MS}$ Completeness: G7 vs. G20

| Sample | Setting | Predicted Values | Observed values + RMSE*s |
|--------|---------|------------------|--------------------------|
| 1 | G7 | 1.36, 0.93, 0.78, 153.22 | 1.37, 0.93, 0.79, 157.70 |
|   | G20 | 1.38, 0.93, 0.82, 157.17 | 0.27 vs. 1.10 |
| 2 | G7 | 1.36, 0.93, 0.79, 159.16 | 1.36, 0.92, 0.79, 156.37 |
|   | G20 | 1.38, 0.95, 0.82, 157.91 | 0.98 vs. 0.61 |
| 3 | G7 | 1.38, 0.94, 0.82, 159.78 | 1.37, 0.94, 0.81, 154.81 |
|   | G20 | 1.39, 0.94, 0.82, 153.60 | 0.78 vs. 1.17 |

to the observed values and a lower mean RMSE* (0.68 vs. 0.96). Completeness is not required in ESE, though a more comprehensive composition enhances performance by better capturing interactions between member systems. [9]

# 6 RELATED WORK

In time series forecasting, auto-regressive models are the most common, such as ARIMA ArunKumar et al. (2021); Benvenuto et al. (2020) and DeepAR Le Guen & Thome (2020). ARIMA is a linear regression model adept at capturing trends and seasonality. DeepAR offers a probabilistic forecasting approach to model by using dense connections. Recurrent neural networks can model temporal dependencies, such as LSTM Feng et al. (2022); Omran et al. (2021); Sah et al. (2022); Shahid et al. (2020). Motivated by attention mechanisms, several transformer-based methods emerged, such as Informer Zhou et al. (2021), FEDformer Zhou et al. (2022b), and PatchTST Nie et al. (2023), especially Time-MoE Xiaoming et al. (2025) that is designed for billion-scale time series. Convolutional methods, e.g., TCN Bai et al. (2018) uses causal convolution to ensure predictions depend on past data, avoiding future information affecting the past. The use of large language models (LLMs) for time series forecasting has recently received growing attention, such as Time-LLM Jin et al. (2024) and AutoTimes Liu et al. (2024).

Multi-variate forecasting requires modeling inter-dependencies between variables. Commonly used methods include VAR Hyndman & Athanasopoulos (2018) and SEM Kang & Ahn (2021), which examines the covariation between variables. As the number of time series increases, the complexity of inter-variable relationships grows rapidly. While effective for small-scale tasks, multi-variate methods become inadequate and inefficient when applied to high-dimensional time series, such as predicting exchange rates and COVID-19 spread.

# 7 CONCLUSION

This study proposes ESE, a novel paradigm for simultaneous forecasting of multiple systems. Unlike conventional methods, ESE does not treat interacting systems as multi-variate time series but as a group. By holistically analyzing the equilibrium state, ESE enables the forecasting of all systems in a single run. It achieves performance comparable to SOTA methods but at much lower cost. It can integrate with existing methods to achieve the best prediction performance while maintaining high efficiency. Its low complexity makes it highly scalable for scenarios involving a large number of systems. Moreover, ESE is not sensitive to noise, even when several systems are subject to perturbations. In sum, ESE is accurate, efficient, robust, and scalable for predicting multiple systems.

**Limitations:** (1) ESE is based on equilibrium, so not suitable for scenarios that do not satisfy Definitions 1, 2, and 3, e.g. multiple unrelated systems. (2) ESE performs better with long inputs, as short-inputs may lack sufficient data for equilibrium estimation (see Appendix M.1, Tables 21 -24).

---

[9]Additional analyses on weighting strategy and true equilibrium are viewable in **Appendix N and O.** Weighting strategy shows that naive weights perform substantially worse than equilibrium estimation, as they fail to capture meaningful long-run relationships.

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

## A  DIFFERENTIATING SIMILAR-LOOKING CONCEPTS

There exist several concepts similar to our ESE prediction of multiple systems, such as multi-variate time series, and multi-compartment models. Here, we clarify the distinctions between them.

**Multi-variate** time series is a data structure consisting of multiple time series, such as ETT (Electricity Transformer Temperature)Zhou et al. (2021), MIMIC-III (The Medical Information Mart for Intensive Care III)LSAEW & Pollard (2016), Electricity (Individual Household Electric Power Consumption Dataset)Hebrail & Berard (2012), Weather (Max-Planck-Institut Weather Dataset for Long-term Time Series Forecasting)Kolle (2025), characterized by the observation of several variables at the same time points. This temporal alignment implies that the variables are correlated (e.g., through lagged correlations or dynamic dependencies), thereby providing additional information to improve predictive accuracy, and any variable can serve as the prediction target. This stands in fundamental difference to our settings, where target variable cannot swap with attribute variables.

**Multi-compartment** prediction models a system that is divided into multiple compartments, for example, predicting drug concentration over time across compartments like blood plasma, tissues, or organs Rescigno (1960); Nakata et al. (2012). These compartments are often heterogeneous. In our case, all systems in the ensemble are similar in nature, e.g., each being a country or a suburb.

**Multi-target** prediction refers to the forecast of multiple target variables from a shared set of input features. For example, (1) the number of daily visits, (2) sales and (3) net profits can be predicted together for one shop Del Valle et al. (2023). Although bearing some similarity this is different from our ESE prediction, which simultaneously predicts the visits or sales of multiple stores.

**Multi-objective** refers specifically to optimization tasks that involve multiple, potentially conflicting or independent, estimation objectives that need to be considered concurrently. Game-theoretic time series models scenarios where the underlying dynamics are based on strategic interactions among multiple decision-making agents or systems. These two concepts are different from standard time-series modeling and not related to this study.

## B  MORE DETAILED DESCRIPTION OF EQUILIBRIUM STATE

As stated in the introduction section, our equilibrium state estimation method (ESE) analyzes the equilibrium state of an ensemble of multiple systems $\mathcal{MS}$. The core idea is similar to Nash equilibrium Kreps (1989), performing state estimation by analyzing the internal competitive relationship between systems. ESE also relates to the concept of deep equilibrium model Bai et al. (2019) and image information transformation Xu & Song (2022). That is, the changes in the state of $\mathcal{MS}$ can be obtained by studying the internal competitive relationship between "internal" systems. This relationship is estimated by feature information of attributes from all "internal" systems (introducing at Section 3.1).

**Equilibrium Conditions**

$$
\begin{aligned}
u_1(\alpha_1^*, \alpha_2^*) \geq u_1(\alpha_1, \alpha_2^*); \\
u_2(\alpha_1^*, \alpha_2^*) \geq u_2(\alpha_1^*, \alpha_2),
\end{aligned}
\tag{9}
$$

To estimate the equilibrium state of $\mathcal{MS}$, the relationships between the systems need to be analyzed. According to Nash equilibrium's basic payoff function, we can have Eq. 9 Eatwell et al. (1989); Von Neumann & Morgenstern (2007). They are for a two-player game with a single decision-making point, e.g. betray or not. Function $u_i()$ represents the payoff of player $i$, and $\alpha_i$ represents the decision of player $i$. Note that $\alpha_1$ and $\alpha_2$ are in the same decision space. That is if $\alpha_1$ is a decision $\{yes, no\}$, $\alpha_2$ is also of decision $\{yes, no\}$. Further, $(\alpha_1^*, \alpha_2^*)$ are the decisions under Nash Equilibrium. Thus, we can obtain the conditions for being in an equilibrium state as below.

**Lemma 1:** The equilibrium state of the integrated systems $\mathcal{MS}$ means that all systems of $\mathcal{MS}$ reach their maximum benefit, or proportion in the system $\mathcal{MS}$, under mutual influence based on their attributes $\mathcal{A}$.

$$
\begin{aligned}
U_1(\mathcal{A}_1^*, \mathcal{A}_2^*, \ldots, \mathcal{A}_n^*) \geq U_1(\mathcal{A}_1, \mathcal{A}_2^* \ldots, \mathcal{A}_n^*) \\
U_2(\mathcal{A}_1^*, \mathcal{A}_2^*, \ldots, \mathcal{A}_n^*) \geq U_2(\mathcal{A}_1^*, \mathcal{A}_2 \ldots, \mathcal{A}_n^*) \\
\vdots \\
U_n(\mathcal{A}_1^*, \mathcal{A}_2^*, \ldots, \mathcal{A}_n^*) \geq U_n(\mathcal{A}_1^*, \mathcal{A}_2^*, \ldots, \mathcal{A}_n),
\end{aligned}
\tag{10}
$$

In the case of $\mathcal{MS}$ with a group of attributes, we can define its equilibrium conditions as in Eq. 10. It is a generalization of Eq. 9. As shown in Eq. 10, changes in any attribute of any system will lead to changes in other systems. Based on Constraints 1 and 2 in Section 2, we can only focus on the relationships between systems of $\mathcal{MS}$. Therefore, the payoff function $U()$ can be unified, as Eq. 11[10]. In this way, internal variations within the system, such as interactions and feedback loops between systems, can be transformed into a distribution of proportions, significantly reducing the complexity of equilibrium state analysis. The trends of each $s_i$ can be reflected in proportions more clearly and consistently, without considering the tendency of the $\mathcal{MS}$. The stronger the $s_i$, the greater the proportion. No matter how the $\mathcal{MS}$ develops, the proportions of the $s_{1:n}$ will not change if no change in the attributes of $s_{1:n}$. Note, we do not require $\mathcal{MS}$ to reach a true equilibrium to estimate its equilibrium state for ESE.

$$U(\mathcal{A}_1^*, \mathcal{A}_2^*, \ldots, \mathcal{A}_n^*) \geq \cdots \geq U(\mathcal{A}_1, \mathcal{A}_2 \ldots, \mathcal{A}_n). \tag{11}$$

Based on the above, the equilibrium state of a $\mathcal{MS}$ can be estimated by estimating the proportion of each system based on their attribute values $\mathcal{A}$, as presented in Section 3.1. This study leverages the concept of equilibrium from Nash equilibrium to model the relationships among systems within the same $\mathcal{MS}$, as proportional changes relative to one another.

## C  SIMPLIFYING EQUILIBRIUM STATE FOR ESE

As described in **Appendix B**, the concept of equilibrium in this study derives from Nash equilibrium. Hence the equilibrium conditions, Eq. 9, can be extended as Eq. 10 according to **Lemma 1:**, which is also shown below as Eq. 12.

$$\begin{aligned}
U_1(\mathcal{A}_1^*, \mathcal{A}_2^*, \ldots, \mathcal{A}_n^*) &\geq U_1(\mathcal{A}_1, \mathcal{A}_2^* \ldots, \mathcal{A}_n^*) \\
U_2(\mathcal{A}_1^*, \mathcal{A}_2^*, \ldots, \mathcal{A}_n^*) &\geq U_2(\mathcal{A}_1^*, \mathcal{A}_2 \ldots, \mathcal{A}_n^*) \\
&\vdots \\
U_n(\mathcal{A}_1^*, \mathcal{A}_2^*, \ldots, \mathcal{A}_n^*) &\geq U_n(\mathcal{A}_1^*, \mathcal{A}_2^*, \ldots, \mathcal{A}_n),
\end{aligned} \tag{12}$$

This compound equation is not friendly to compute. To simplify it, we can start from a two-player scenario, where the payoff functions can be expressed as Eq. 13:

$$\begin{aligned}
U_1(\mathcal{A}_1, \mathcal{A}_2) &= \sum_{j=1}^{J} \sum_{k=1}^{K} u_1(\theta_{1,j} \cdot \alpha_{1,j}, \ \phi_{1,k} \cdot \alpha_{2,k}); \\
U_2(\mathcal{A}_1, \mathcal{A}_2) &= \sum_{j=1}^{J} \sum_{k=1}^{K} u_2(\theta_{2,j} \cdot \alpha_{1,j}, \ \phi_{2,k} \cdot \alpha_{2,k}),
\end{aligned} \tag{13}$$

where $U_1()$ and $U_2()$ are the payoff functions for players 1 and 2 under multiple decisions (attributes). $\mathcal{A}_1$ is the decision (attribute) set of player 1, containing $J$ different decisions, $(\alpha_{1,1}, \ldots, \alpha_{1,J})$. $\mathcal{A}_2$ is the decision (attribute) set of player 2, containing $K$ different decisions, $(\alpha_{2,1}, \ldots, \alpha_{2,K})$. All attributes, e.g. $\alpha_{1,j}$ and $\alpha_{2,k}$ are not independent and may influence each other. In the equations, $\theta_{1,j}$ is the coefficient on attribute $\alpha_{1,j}$ of player 1, while $\phi_{1,k}$ is the coefficient on attribute $\alpha_{1,k}$ of player 2, both on the payoff function of player 1, $U_1(\mathcal{A}_1, \mathcal{A}_2)$. Similarly, $\theta_{2,j}$ and $\phi_{2,k}$ are the corresponding coefficients on the payoff function of player 2, $U_2(\mathcal{A}_1, \mathcal{A}_2)$. To simplify the ESE process, we assume the attribute set $\mathcal{A}$ of every player is identical (**Definition 3, Section 2**). Therefore, we can reduce Eq. 13 to Eq. 14, as below:

$$\begin{aligned}
U_1(\mathcal{A}_1, \mathcal{A}_2) &= \sum_{j=1}^{J} u(\psi_j \cdot \alpha_{1,j}, \ \psi_j \cdot \alpha_{2,j}); \\
U_2(\mathcal{A}_1, \mathcal{A}_2) &= \sum_{j=1}^{J} u(\psi_j \cdot \alpha_{1,j}, \ \psi_j \cdot \alpha_{2,j}),
\end{aligned} \tag{14}$$

In ESE, the payoff functions $U_1()$ and $U_2()$ of the two players are considered identical, as both players are to be predicted for the same target value. Therefore, $U_1(\mathcal{A}_1, \mathcal{A}_2)$ and $U_2(\mathcal{A}_1, \mathcal{A}_2)$ are the same and can be unified as $U(\mathcal{A}_1, \mathcal{A}_2)$. As a result, the payoff functions for both players can be

---

[10]The deductive reasoning process of the payoff function is presented in **Appendix C**.

expressed as one function $U(\mathcal{A}_1, \mathcal{A}_2)$. For $n$-player scenarios, that would still hold true, as long as $n >= 2$. Hence the equilibrium state of $\mathcal{MS}$ with $n$ systems can be simplified as Eq. 15, which is Eq.. 11 in the main paper:

$$U(\mathcal{A}_1^*, \mathcal{A}_2^*, \ldots, \mathcal{A}_n^*) \geq \cdots \geq U(\mathcal{A}_1, \mathcal{A}_2 \ldots, \mathcal{A}_n). \tag{15}$$

## D  COINTEGRATION AND CONVERGENCE IN ESE

One core component of ESE's Equilibrium Estimation process, outlined in **Algorithm 1**, is the test of long-run equilibrium using cointegration. At any timestep $t$, when $\mathcal{ST}_t$ and $\mathcal{ES}_t^{[e]}$, the estimated state at Epoch $e$, are cointegrated, their relationship can be expressed as: $\mathcal{ST}_t = \beta_0 + \beta_1 \mathcal{ES}_t^{[e]} + \epsilon_t$, where $\beta_0$ and $\beta_1$ are estimated using Ordinary Least Squares (OLS). Hence, the residual term $\epsilon_t$ can be obtained as in Eq. 16:

$$\epsilon_t = \mathcal{ST}_t - \beta_0 - \beta_1 \mathcal{ES}_t^{[e]}. \tag{16}$$

Cointegration between $\mathcal{ST}_t$ and $\mathcal{ES}_t^{[e]}$ is then tested by checking whether $\epsilon_t$ is stationary[11] (i.e., white noise). A p-value below 0.05 indicates statistical significance, confirming that $\epsilon_t$ is stationary and that cointegration exists between $\mathcal{ST}_t$ and $\mathcal{ES}_t^{[e]}$.

If systems' attributes are random nor do not show influence on the prediction targets, then Algorithm 1 will not converge, e.g. cointegration would never be satisfied. For example, when we set $\psi_t$ values to zero, the influence of attributes are effectively disabled (Eq. 5). Algorithm 1 will fail to converge.

### D.1  CONVERGENCE CURVES

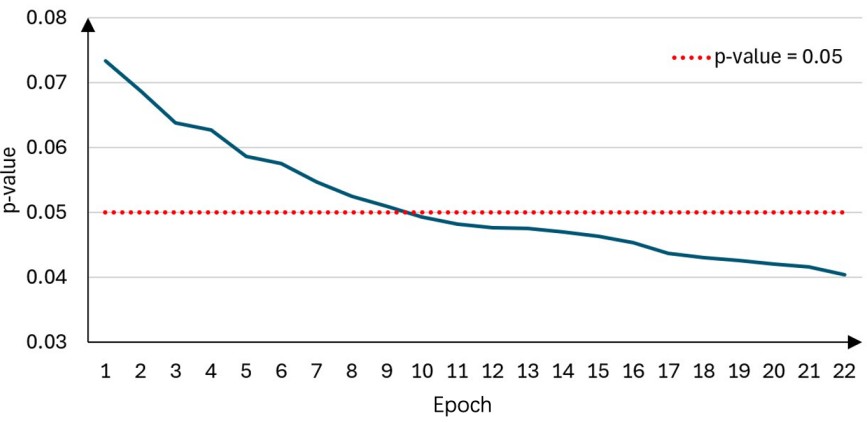

Figure 4: Convergence of ESE on Synthetic Data, 20 Systems. The blue line represents the p-values obtained at each epoch of ESE convergence. The red dotted line represents a p-value of 0.05, the threshold for rejecting the null hypothesis for the existence of a long-run equilibrium.

Figures 4 and 5 show the convergence process during ESE's equilibrium estimation, on synthetic data and COVID data, respectively. P-values are obtained from the cointegration test (Step 2, **Algorithm 1**), where the null hypothesis is rejected if the value is lower than 0.05. For illustration purposes, we allow the convergence to continue beyond 0.05 on these two figures. During an actual convergence, it will stop once the p-value reaches 0.05, showing the existence of a long-run equilibrium. From the figures, we can see ESE convergence process is steady and effective. In addition, the convergence process on COVID takes longer (43 epochs) to go below 0.05 than that for synthetic data (10 epochs).

---

[11]Estimation and cointegration test are implemented by existing packages *statsmodels.tsa.vector_ar.vecm* Seabold & Perktold (2010)

That suggests more effort is needed for the estimation process in the case of COVID, consistent with the complex nature of the data.

Because of this analysis, we can confirm that stochastic Fleming & Rishel (2012) and oscillation behaviors Morin (2008) are not a concern, as they can often be observed in real-world scenarios, like epidemic spreading. More details about conintegration and long-run equilibrium can be found in Chen et al. (2009); Maki & Kitasaka (2006).

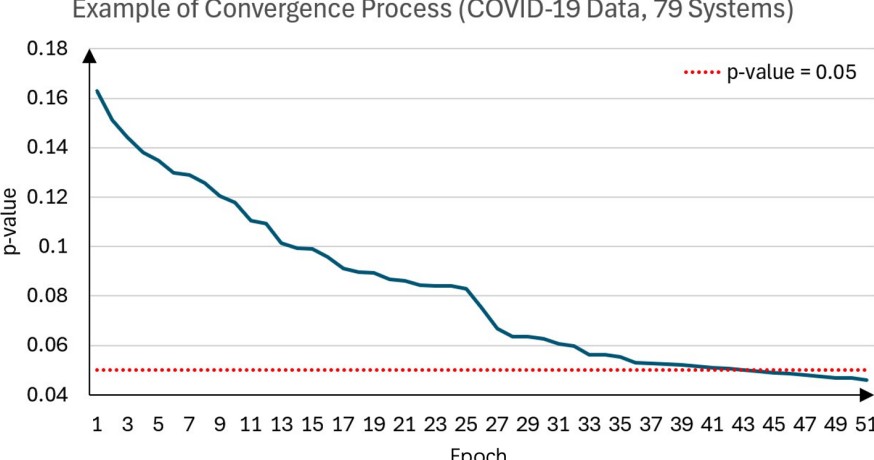

Figure 5: Convergence of ESE on COVID-19 Data, 79 Systems. The blue line represents p-values obtained from the cointegration test at each epoch of ESE convergence. The red dotted line is the threshold for rejecting the null hypothesis for the existence of a long-run equilibrium.

### D.2 P-VALUE, CONTINUOUS STOPPING, OPTIONAL STOPPING

Setting the p-value threshold at 0.05 is a common practice in statistical analysis, as referenced in Kremers et al. (1992); Yussuf (2022). However, this does not imply that the threshold must be fixed at 0.05. Below, we show a sequence of test results under different p-values, using currency exchange rate data. Overall, we observe that RMSE and MAE tend to improve with smaller p-values, although the improvement below 0.05 is marginal.

| p-value | 0.01 | 0.02 | 0.03 | 0.04 | 0.05 | 0.06 | 0.07 | 0.08 | 0.09 | 0.10 | 0.11 | 0.12 | 0.13 | 0.14 | 0.15 |
|---|---|---|---|---|---|---|---|---|---|---|---|---|---|---|---|
| RMSE | 5.87 | 5.68 | 6.04 | 5.88 | 6.01 | 6.58 | 7.41 | 6.95 | 7.68 | 8.38 | 7.63 | 9.77 | 8.78 | 9.66 | 10.39 |

### D.3 DAMPING VALUE $\lambda$

As for the damping coefficient being 0.5, the choice follows directly from the stability constraints of ESE's iterative update in Line 4 of Algorithm 1: $\mathcal{L}^{[e]} \leftarrow \mathcal{ES}\_t^{[e]} - \mathcal{ST}\_t' + (\mathcal{L}^{[e]}) \cdot \lambda$. In this update, the term $\mathcal{ES}\_t^{[e]} - \mathcal{ST}\_t'$ has a maximum magnitude of 1 due to normalization. Therefore, if $\lambda > 0.5$, the recursive update may grow in magnitude rather than contract, causing $L^{[e]}$ to diverge instead of converge. If $\lambda < 0.5$, the update becomes conservative, slowing down convergence unnecessarily. The table below is an empirical comparison of different $\lambda$ values under 0.5. Smaller $\lambda$ values take longer to converge, increasing total computation time without affecting the final equilibrium, as all settings converged to the same final state.

Thus, $\lambda$=0.5 is the maximum stable damping factor that still ensures efficient convergence.

| | Damping Value | Epoch | Time (sec) |
|---|---|---|---|
| 1 | 0.5 | 14 | 9.37 |
| 2 | 0.4 | 18 | 11.13 |
| 3 | 0.3 | 21 | 12.44 |
| 4 | 0.2 | 27 | 14.42 |
| 5 | 0.1 | 36 | 16.61 |

## E  PROOF OF SEPARABILITY BETWEEN OVERALL TREND AND EQUILIBRIUM STATE

This section proves a point used in the predictor (Section 3.2): the change in the overall multiple system value $\Delta \mathcal{M} \mathcal{S}_{t+h}$ is mathematically separable from the dynamics of the proportional states (Equilibrium State) $\gamma_{i,t}$. This allows us to model the overall trend and the internal distribution independently.

Recall the definitions:

- $s_{i,t} = \gamma_{i,t} \cdot \mathcal{M} \mathcal{S}_t$

- $\sum_{i=1}^{n} \gamma_{i,t} = 1$

The change in the value of a single system $i$ over a horizon $h$, defined as:

$$\Delta s_{i,t+h} = s_{i,t+h} - s_{i,t} \tag{17}$$

The core of our approach is to forecast the future state based on the current proportional distribution and a predicted overall change. Therefore, in this forecasting model, we express the future value $s_{i,t+h}$ as:

$$s_{i,t+h} = \gamma_{i,t} \cdot (\mathcal{M} \mathcal{S}_t + \Delta \mathcal{M} \mathcal{S}_{t+h}) \tag{18}$$

Eq. 18 is a modeling choice that assumes the internal distribution relative to the equilibrium is stable over the short-term forecast horizon $h$, which is central to the ESE.

Substituting the definition of $s_{i,t+h}$ and $s_{i,t}$ into the definition of $\Delta s_{i,t+h}$, we get:

$$\begin{aligned}
\Delta s_{i,t+h} &= [\gamma_{i,t} \cdot (\mathcal{M} \mathcal{S}_t + \Delta \mathcal{M} \mathcal{S}_{t+h})] - [\gamma_{i,t} \cdot \mathcal{M} \mathcal{S}_t] \\
&= \gamma_{i,t} \cdot \mathcal{M} \mathcal{S}_t + \gamma_{i,t} \cdot \Delta \mathcal{M} \mathcal{S}_{t+h} - \gamma_{i,t} \cdot \mathcal{M} \mathcal{S}_t \\
&= \gamma_{i,t} \cdot \Delta \mathcal{M} \mathcal{S}_{t+h}
\end{aligned} \tag{19}$$

Equ. 19 shows that under our forecasting model, the change in an individual system's value is proportional to the change in the overal value, scaled by its current proportion $\gamma_{i,t}$. The term $\Delta \mathcal{M} \mathcal{S}_{t+h}$ depends on the overall trend of the ensemble, while $\gamma_{i,t}$ depends on the current system state. This demonstrates that these two components are separate and independent factors in the forecast of the change $\Delta s_{i,t+h}$. The aggregate trend does not dictate the internal distribution, and vice-versa.

## F  PREDICTOR

Eq. 20, 21, 22 show the estimation process of Eq. 7 by log maximum likelihood.

$$L(\theta) \overset{def}{=} p(\mathcal{MS}_1, \ldots, \mathcal{MS}_k|\theta) = p(\mathcal{MS}_1) \left(\frac{1}{\sigma\sqrt{2\pi}}\right)^{k-1} exp\left\{-\frac{1}{2\sigma^2}\sum_{t=2}^{k}(\mathcal{MS}_t - \theta\mathcal{MS}_{t-1})^2\right\},$$

$$logL(\theta) = logp(\mathcal{MS}_1) - (k-1)log(\sigma\sqrt{2\pi}) - \frac{1}{2\sigma^2}(\mathcal{MS}_t - \theta\mathcal{MS}_{t-1})^2 \tag{20}$$

$$\hat{\theta_{mle}} = argmax\ logL(\theta), \tag{21}$$

$$\mathcal{MS}_{t+i} = \theta_{t+i}\mathcal{MS}_t + \epsilon_{t+i},\ \epsilon_{t+i} \overset{i.i.d.}{\sim} N(0,\sigma^2);\ \ \mathcal{MS}_t = \sum_{i=1}^{n} c_{t,i}, \tag{22}$$

where $\mathcal{MS}_t$ is the total target value of the $\mathcal{MS}$ at time $t$. $\theta_t$ is the coefficient parameter for the linear relationship of the model at time $t$, which was estimated by log maximum likelihood.

## G  SYNTHETIC DATA

To validate ESE, three synthetic systems are created, consisting of 5, 10, and 20 systems, respectively. Each system contains one series of targets and two series of attributes for 1000 time points. They are generated by Eq. 23,

$$y_t = log(C + \beta y_{t-1} + e_t) \tag{23}$$

where $\beta$ is an adjustable coefficient, with a value of $\beta = 1.2$ in this study. $C$ is the intercept, which is randomly chosen from $[50, 100]$ for the target and from $[1, 10]$ for the attributes. To add some white noise, a random number ($e_t$) is added in the range of $[-1, 1]$ based on Gaussian distribution.

## H  FULL COMPARISONS ON SYNTHETIC DATA

### H.1  COMPARING PREDICTION PERFORMANCE - SYNTHETIC DATA

Table 7: Comparison of prediction performance using 5/10/20 systems on synthetic data. RMSE and MAE rows show the means and standard deviations of RMSE and MAE results of all runs of one method, respectively. The experiments are repeated for three input lengths: 10, 20 and 50. The prediction target is fixed at a horizon of 1. We can observe that **(1)** ESE alone is consistently competitive in accuracy and never the worst; **(2)** When combined with ESE, all predictors either improve or maintain their performance; **(3)** The lowest error in each column is almost achieved either by ESE alone or by a SOTA predictor augmented with ESE.

| Models | | Metric | Prediction Horizon = 1 | | |
| --- | --- | --- | --- | --- | --- |
| | | | *Input Length = 10* | *Input Length = 20* | *Input Length = 50* |
| ESE | | RMSE | 0.246±0.01 / 0.246±0.01 / 0.245±0.01 | 0.252±0.01 / 0.248±0.01 / 0.255±0.01 | 0.295±0.01 / 0.270±0.01 / 0.282±0.01 |
| | | MAE | 0.208±0.01 / 0.207±0.01 / 0.207±0.01 | 0.210±0.01 / 0.228±0.01 / 0.216±0.01 | 0.229±0.01 / 0.261±0.01 / 0.257±0.01 |
| ARIMA | No ESE | RMSE | 0.241±0.01 / 0.241±0.01 / 0.241±0.01 | 0.243±0.02 / 0.249±0.02 / 0.262±0.01 | **0.276±0.01** / 0.289±0.01 / 0.314±0.02 |
| | | MAE | 0.222±0.01 / 0.222±0.01 / 0.222±0.01 | 0.229±0.01 / 0.243±0.01 / 0.235±0.01 | 0.248±0.01 / 0.283±0.01 / 0.246±0.01 |
| | With ESE | RMSE | 0.243±0.02 / 0.243±0.01 / **0.239±0.02** | **0.240±0.02** / 0.247±0.01 / 0.261±0.02 | 0.276±0.02 / 0.291±0.01 / 0.312±0.02 |
| | | MAE | 0.221±0.01 / 0.220±0.01 / 0.223±0.01 | 0.228±0.01 / 0.243±0.01 / 0.233±0.01 | 0.247±0.01 / 0.285±0.01 / 0.248±0.01 |
| LSTM | No ESE | RMSE | 0.250±0.01 / 0.250±0.01 / 0.260±0.01 | 0.258±0.01 / 0.263±0.01 / 0.284±0.01 | 0.298±0.01 / 0.297±0.01 / 0.297±0.01 |
| | | MAE | 0.203±0.01 / 0.203±0.01 / **0.202±0.01** | 0.212±0.01 / 0.216±0.01 / 0.216±0.01 | 0.226±0.01 / 0.245±0.01 / 0.247±0.01 |
| | With ESE | RMSE | 0.249±0.01 / 0.250±0.01 / 0.259±0.01 | 0.255±0.01 / 0.263±0.01 / 0.280±0.01 | 0.298±0.01 / 0.296±0.01 / 0.296±0.01 |
| | | MAE | 0.204±0.01 / **0.202±0.01** / 0.202±0.01 | 0.212±0.01 / 0.212±0.01 / 0.213±0.01 | 0.224±0.01 / 0.247±0.01 / 0.246±0.01 |
| Dlinear | No ESE | RMSE | 0.244±0.01 / 0.244±0.01 / 0.243±0.01 | 0.257±0.01 / 0.256±0.01 / 0.264±0.01 | 0.309±0.01 / 0.289±0.01 / 0.267±0.01 |
| | | MAE | **0.203±0.01** / 0.205±0.01 / 0.211±0.01 | 0.214±0.01 / 0.221±0.01 / **0.204±0.01** | 0.223±0.01 / 0.230±0.01 / 0.234±0.01 |
| | With ESE | RMSE | 0.242±0.01 / **0.241±0.01** / 0.245±0.01 | 0.254±0.01 / 0.264±0.01 / 0.260±0.01 | 0.309±0.01 / **0.287±0.01** / **0.265±0.01** |
| | | MAE | 0.215±0.01 / 0.214±0.01 / 0.203±0.01 | 0.215±0.01 / 0.221±0.01 / 0.204±0.01 | 0.225±0.01 / **0.232±0.01** / 0.232±0.01 |
| Informer | No ESE | RMSE | 0.243±0.01 / 0.243±0.01 / 0.249±0.01 | 0.248±0.01 / 0.244±0.01 / 0.252±0.01 | 0.283±0.01 / 0.283±0.01 / 0.281±0.01 |
| | | MAE | 0.225±0.01 / 0.237±0.01 / 0.233±0.01 | 0.213±0.01 / 0.236±0.01 / 0.245±0.01 | 0.227±0.01 / 0.278±0.01 / 0.268±0.01 |
| | With ESE | RMSE | 0.242±0.01 / 0.242±0.01 / 0.251±0.01 | 0.246±0.01 / **0.241±0.01** / 0.251±0.01 | 0.283±0.01 / 0.282±0.01 / 0.279±0.01 |
| | | MAE | 0.226±0.01 / 0.237±0.01 / 0.246±0.01 | 0.239±0.01 / 0.232±0.01 / 0.234±0.01 | 0.228±0.01 / 0.256±0.01 / 0.271±0.01 |
| DeepAR | No ESE | RMSE | 0.248±0.01 / 0.248±0.01 / 0.255±0.01 | 0.252±0.01 / 0.271±0.01 / 0.279±0.01 | 0.281±0.01 / 0.315±0.01 / 0.335±0.01 |
| | | MAE | 0.203±0.01 / 0.204±0.01 / 0.209±0.01 | 0.215±0.01 / 0.218±0.01 / 0.214±0.01 | 0.251±0.01 / 0.244±0.01 / 0.257±0.01 |
| | With ESE | RMSE | 0.249±0.01 / 0.250±0.01 / 0.257±0.01 | 0.248±0.01 / 0.271±0.01 / 0.278±0.01 | 0.280±0.01 / 0.314±0.01 / 0.334±0.01 |
| | | MAE | 0.202±0.01 / 0.205±0.01 / 0.204±0.01 | 0.214±0.01 / **0.210±0.01** / 0.211±0.01 | 0.253±0.01 / 0.243±0.01 / 0.256±0.01 |
| PatchTST | No ESE | RMSE | 0.241±0.01 / 0.241±0.01 / 0.241±0.01 | 0.251±0.01 / 0.262±0.01 / 0.247±0.01 | 0.294±0.01 / 0.263±0.01 / 0.291±0.01 |
| | | MAE | 0.207±0.01 / 0.207±0.01 / 0.208±0.01 | 0.208±0.01 / 0.219±0.01 / 0.225±0.01 | 0.211±0.01 / 0.233±0.01 / 0.229±0.01 |
| | With ESE | RMSE | **0.240±0.01** / 0.243±0.01 / 0.243±0.01 | 0.250±0.01 / 0.260±0.01 / **0.246±0.01** | 0.292±0.01 / **0.261±0.01** / 0.291±0.01 |
| | | MAE | 0.209±0.01 / 0.208±0.01 / 0.208±0.01 | **0.207±0.01** / 0.218±0.01 / 0.225±0.01 | **0.211±0.01** / 0.232±0.01 / **0.229±0.01** |

Table 8: Comparison of prediction performance using 5/10/20 systems on synthetic data. RMSE and MAE rows show the means and standard deviations of RMSE and MAE results of all runs of one method, respectively. The experiments are repeated for three input lengths: 10, 20 and 50. The prediction target is fixed at a horizon of 2. We can observe that (1) ESE alone is consistently competitive in accuracy and never the worst; (2) When combined with ESE, all predictors either improve or maintain their performance; (3) The lowest error in each column is almost achieved either by ESE alone or by a SOTA predictor augmented with ESE.

| Models | | Metric | Prediction Horizon = 2 Input Length = 10 | Input Length = 20 | Input Length = 50 |
|---|---|---|---|---|---|
| ESE | | RMSE | 0.269±0.01 / 0.267±0.01 / 0.266±0.01 | 0.284±0.01 / 0.276±0.01 / 0.285±0.01 | 0.336±0.01 / 0.283±0.01 / 0.320±0.01 |
| | | MAE | 0.218±0.01 / 0.218±0.01 / 0.215±0.01 | 0.234±0.01 / 0.236±0.01 / 0.228±0.01 | 0.264±0.01 / 0.266±0.01 / 0.266±0.01 |
| ARIMA | No ESE | RMSE | 0.254±0.01 / 0.244±0.01 / 0.253±0.02 | 0.246±0.01 / 0.256±0.02 / 0.258±0.02 | 0.293±0.02 / 0.270±0.01 / **0.279±0.02** |
| | | MAE | 0.216±0.01 / 0.215±0.01 / 0.213±0.01 | 0.235±0.01 / 0.220±0.01 / 0.221±0.01 | 0.246±0.01 / 0.235±0.01 / 0.261±0.01 |
| | With ESE | RMSE | 0.258±0.01 / 0.256±0.02 / 0.255±0.02 | 0.259±0.01 / **0.253±0.01** / **0.256±0.02** | 0.290±0.01 / 0.272±0.02 / 0.281±0.02 |
| | | MAE | 0.218±0.01 / 0.215±0.01 / 0.216±0.01 | 0.234±0.01 / 0.219±0.01 / 0.217±0.01 | 0.245±0.01 / **0.233±0.01** / 0.263±0.01 |
| LSTM | No ESE | RMSE | 0.269±0.01 / 0.267±0.01 / 0.266±0.01 | 0.292±0.01 / 0.282±0.01 / 0.278±0.01 | 0.337±0.01 / 0.290±0.01 / 0.304±0.01 |
| | | MAE | 0.215±0.01 / 0.214±0.01 / 0.213±0.01 | 0.226±0.01 / 0.223±0.01 / 0.219±0.01 | 0.268±0.01 / 0.266±0.01 / 0.252±0.01 |
| | With ESE | RMSE | 0.270±0.01 / 0.267±0.01 / 0.269±0.01 | 0.289±0.01 / 0.277±0.01 / 0.275±0.01 | 0.341±0.01 / 0.292±0.01 / 0.305±0.01 |
| | | MAE | 0.215±0.01 / 0.217±0.01 / 0.216±0.01 | 0.222±0.01 / 0.221±0.01 / 0.215±0.01 | 0.268±0.01 / 0.264±0.01 / 0.243±0.01 |
| Dlinear | No ESE | RMSE | 0.257±0.01 / 0.251±0.01 / 0.250±0.01 | 0.281±0.01 / 0.259±0.01 / 0.272±0.01 | 0.282±0.01 / 0.268±0.01 / 0.296±0.01 |
| | | MAE | 0.214±0.01 / 0.224±0.01 / 0.213±0.01 | 0.226±0.01 / 0.219±0.01 / 0.216±0.01 | 0.256±0.01 / 0.244±0.01 / 0.246±0.01 |
| | With ESE | RMSE | 0.260±0.01 / 0.256±0.01 / 0.253±0.01 | 0.278±0.01 / 0.256±0.01 / 0.268±0.01 | 0.284±0.01 / **0.268±0.01** / 0.295±0.01 |
| | | MAE | 0.215±0.01 / 0.214±0.01 / 0.213±0.01 | 0.227±0.01 / 0.216±0.01 / **0.214±0.01** | 0.255±0.01 / 0.245±0.01 / 0.246±0.01 |
| Informer | No ESE | RMSE | 0.254±0.01 / **0.244±0.01** / 0.253±0.01 | 0.272±0.01 / 0.273±0.01 / 0.270±0.01 | 0.293±0.01 / 0.316±0.01 / 0.283±0.01 |
| | | MAE | 0.218±0.01 / 0.217±0.01 / 0.227±0.01 | 0.228±0.01 / 0.218±0.01 / 0.234±0.01 | 0.262±0.01 / 0.235±0.01 / 0.242±0.01 |
| | With ESE | RMSE | 0.255±0.01 / 0.254±0.01 / 0.258±0.01 | 0.270±0.01 / 0.266±0.01 / 0.276±0.01 | 0.296±0.01 / 0.318±0.01 / 0.283±0.01 |
| | | MAE | 0.218±0.01 / 0.221±0.01 / 0.222±0.01 | 0.226±0.01 / 0.217±0.01 / 0.231±0.01 | 0.261±0.01 / 0.236±0.01 / 0.245±0.01 |
| DeepAR | No ESE | RMSE | **0.249±0.01** / 0.257±0.01 / 0.255±0.01 | 0.250±0.01 / 0.266±0.01 / 0.260±0.01 | 0.277±0.01 / 0.275±0.01 / 0.299±0.01 |
| | | MAE | 0.212±0.01 / **0.211±0.01** / **0.206±0.01** | 0.223±0.01 / **0.214±0.01** / 0.226±0.01 | 0.244±0.01 / 0.241±0.01 / 0.238±0.01 |
| | With ESE | RMSE | 0.251±0.01 / 0.258±0.01 / **0.248±0.01** | **0.246±0.01** / 0.264±0.01 / 0.259±0.01 | **0.276±0.01** / 0.286±0.01 / 0.302±0.01 |
| | | MAE | 0.215±0.01 / 0.214±0.01 / 0.212±0.01 | **0.209±0.01** / 0.220±0.01 / 0.225±0.01 | 0.253±0.01 / 0.243±0.01 / **0.237±0.01** |
| PatchTST | No ESE | RMSE | 0.254±0.01 / 0.253±0.01 / 0.255±0.01 | 0.260±0.01 / 0.260±0.01 / 0.263±0.01 | 0.289±0.01 / 0.278±0.01 / 0.286±0.01 |
| | | MAE | 0.212±0.01 / 0.216±0.01 / 0.216±0.01 | 0.217±0.01 / 0.224±0.01 / 0.229±0.01 | 0.229±0.01 / 0.242±0.01 / 0.250±0.01 |
| | With ESE | RMSE | 0.253±0.01 / 0.251±0.01 / 0.258±0.01 | 0.258±0.01 / 0.258±0.01 / 0.263±0.01 | 0.290±0.01 / 0.278±0.01 / 0.283±0.01 |
| | | MAE | **0.211±0.01** / 0.216±0.01 / 0.216±0.01 | 0.217±0.01 / 0.225±0.01 / 0.228±0.01 | **0.228±0.01** / 0.243±0.01 / 0.249±0.01 |

Table 9: Comparison of prediction performance using 5/10/20 systems on synthetic data. RMSE and MAE rows show the means and standard deviations of RMSE and MAE results of all runs of one method, respectively. The experiments are repeated for three input lengths: 10, 20 and 50. The prediction target is fixed at a horizon of 5. We can observe that (1) ESE alone is consistently competitive in accuracy and never the worst; (2) When combined with ESE, all predictors either improve or maintain their performance; (3) The lowest error in each column is almost achieved either by ESE alone or by a SOTA predictor augmented with ESE.

| Models | | Metric | Prediction Horizon = 5 Input Length = 10 | Input Length = 20 | Input Length = 50 |
|---|---|---|---|---|---|
| ESE | | RMSE | 0.277±0.01 / 0.270±0.01 / 0.267±0.01 | 0.286±0.01 / 0.280±0.01 / 0.289±0.01 | 0.317±0.01 / 0.312±0.01 / 0.340±0.01 |
| | | MAE | **0.223±0.01** / **0.223±0.01** / 0.232±0.01 | 0.241±0.01 / 0.256±0.01 / 0.249±0.01 | 0.266±0.01 / 0.268±0.01 / 0.267±0.01 |
| ARIMA | No ESE | RMSE | 0.280±0.01 / 0.277±0.01 / 0.271±0.02 | 0.283±0.01 / 0.295±0.01 / 0.294±0.02 | 0.337±0.02 / 0.353±0.01 / 0.304±0.02 |
| | | MAE | 0.231±0.01 / 0.230±0.01 / **0.229±0.01** | 0.243±0.01 / 0.237±0.01 / 0.259±0.01 | 0.303±0.01 / 0.264±0.01 / 0.278±0.01 |
| | With ESE | RMSE | 0.280±0.01 / 0.286±0.02 / 0.280±0.02 | 0.293±0.01 / 0.291±0.02 / 0.280±0.02 | 0.323±0.02 / 0.341±0.02 / 0.291±0.02 |
| | | MAE | 0.234±0.01 / 0.239±0.01 / 0.230±0.01 | 0.243±0.01 / **0.233±0.01** / 0.248±0.01 | 0.288±0.01 / 0.254±0.01 / 0.270±0.01 |
| LSTM | No ESE | RMSE | 0.271±0.01 / **0.261±0.01** / **0.271±0.01** | 0.292±0.01 / 0.284±0.01 / 0.286±0.01 | 0.335±0.01 / 0.351±0.01 / 0.302±0.01 |
| | | MAE | 0.232±0.01 / 0.228±0.01 / 0.232±0.01 | 0.240±0.01 / 0.241±0.01 / 0.252±0.01 | 0.275±0.01 / 0.251±0.01 / 0.268±0.01 |
| | With ESE | RMSE | **0.276±0.01** / 0.271±0.01 / 0.266±0.01 | 0.287±0.01 / **0.279±0.01** / 0.274±0.01 | 0.319±0.01 / 0.335±0.01 / **0.289±0.01** |
| | | MAE | 0.233±0.01 / 0.237±0.01 / 0.235±0.01 | **0.236±0.01** / 0.238±0.01 / 0.248±0.01 | 0.265±0.01 / **0.241±0.01** / 0.260±0.01 |
| Dlinear | No ESE | RMSE | 0.272±0.01 / 0.270±0.01 / 0.271±0.01 | 0.296±0.01 / 0.286±0.01 / 0.302±0.01 | 0.339±0.01 / 0.341±0.01 / 0.357±0.02 |
| | | MAE | 0.231±0.01 / 0.233±0.01 / 0.238±0.01 | 0.246±0.01 / 0.242±0.01 / 0.255±0.01 | 0.267±0.01 / 0.260±0.01 / 0.273±0.01 |
| | With ESE | RMSE | 0.277±0.01 / 0.275±0.01 / 0.272±0.01 | 0.293±0.01 / 0.293±0.01 / 0.289±0.01 | 0.323±0.01 / 0.327±0.01 / 0.343±0.01 |
| | | MAE | 0.234±0.01 / 0.236±0.01 / 0.238±0.01 | 0.244±0.01 / 0.238±0.01 / 0.245±0.01 | 0.256±0.01 / 0.249±0.01 / 0.262±0.01 |
| Informer | No ESE | RMSE | 0.264±0.01 / 0.274±0.01 / 0.254±0.01 | 0.278±0.01 / 0.282±0.01 / 0.311±0.01 | 0.328±0.01 / 0.322±0.01 / 0.339±0.01 |
| | | MAE | 0.280±0.01 / 0.278±0.01 / 0.279±0.01 | 0.277±0.01 / 0.281±0.01 / 0.298±0.01 | 0.318±0.01 / 0.310±0.01 / 0.327±0.01 |
| | With ESE | RMSE | 0.273±0.01 / 0.262±0.01 / 0.272±0.01 | **0.276±0.01** / 0.281±0.01 / **0.271±0.01** | **0.291±0.01** / **0.290±0.01** / 0.298±0.01 |
| | | MAE | 0.237±0.01 / 0.235±0.01 / 0.236±0.01 | 0.245±0.01 / 0.246±0.01 / 0.249±0.01 | 0.278±0.01 / 0.279±0.01 / **0.256±0.01** |
| DeepAR | No ESE | RMSE | **0.261±0.01** / 0.271±0.01 / 0.273±0.01 | 0.278±0.01 / 0.288±0.01 / 0.304±0.01 | 0.329±0.01 / 0.357±0.01 / 0.326±0.01 |
| | | MAE | 0.232±0.01 / 0.237±0.01 / 0.231±0.01 | 0.248±0.01 / 0.243±0.01 / 0.254±0.01 | 0.265±0.01 / 0.274±0.01 / 0.291±0.01 |
| | With ESE | RMSE | 0.273±0.02 / 0.262±0.02 / 0.272±0.02 | 0.278±0.02 / 0.284±0.02 / 0.287±0.02 | 0.317±0.02 / 0.343±0.02 / 0.310±0.02 |
| | | MAE | 0.234±0.01 / 0.233±0.01 / 0.231±0.02 | 0.244±0.01 / 0.241±0.01 / **0.240±0.01** | **0.255±0.01** / 0.262±0.01 / 0.278±0.01 |
| PatchTST | No ESE | RMSE | 0.277±0.01 / 0.272±0.01 / 0.274±0.01 | 0.281±0.01 / 0.293±0.01 / 0.329±0.01 | 0.304±0.01 / 0.295±0.01 / 0.340±0.01 |
| | | MAE | 0.233±0.01 / 0.234±0.01 / 0.234±0.01 | 0.251±0.01 / 0.239±0.01 / 0.245±0.01 | 0.289±0.01 / 0.245±0.01 / 0.265±0.01 |
| | With ESE | RMSE | 0.278±0.01 / 0.274±0.01 / 0.272±0.01 | 0.284±0.01 / 0.293±0.01 / 0.287±0.01 | 0.305±0.01 / 0.297±0.01 / 0.340±0.01 |
| | | MAE | 0.233±0.01 / 0.234±0.01 / 0.235±0.01 | 0.254±0.01 / 0.239±0.01 / 0.244±0.01 | 0.289±0.01 / 0.262±0.01 / 0.267±0.01 |

## H.2 COMPARING COMPUTATIONAL COSTS - SYNTHETIC DATA

Table 10: Comparison of computational cost (in minutes) using 5/10/20 systems on synthetic data. The experiments are repeated for three input lengths: 10, 20 and 50. The prediction target is fixed at a horizon of 1. We can observe that **(1)** ESE incurs significantly lower computational cost than other methods, except ARIMA, which requires no training, hence is fast on small datasets, but not on larger ones; **(2)** When paired with ESE, the cost of SOTA predictors can be reduced significantly, by a factor of 2-10, excluding ARIMA; **(3)** The runtime increases with the number of systems estimated.

| Models | | Costs (mins) for Horizon = 1 | | |
| | | *Input Length* = 10 | *Input Length* = 20 | *Input Length* = 50 |
|---|---|---|---|---|
| ESE | | 0.19 / 0.24 / 0.27 | 0.20 / 0.23 / 0.29 | 0.20 / 0.23 / 0.30 |
| ARIMA | No ESE | **0.04 / 0.09 / 0.17** | **0.04 / 0.09 / 0.17** | **0.04 / 0.08 / 0.18** |
| | With ESE | 0.20 / 0.25 / 0.28 | 0.21 / 0.24 / 0.30 | 0.21 / 0.24 / 0.31 |
| LSTM | No ESE | 1.27 / 2.34 / 5.00 | 1.22 / 2.47 / 4.69 | 1.18 / 2.35 / 4.94 |
| | With ESE | 0.44 / 0.47 / 0.52 | 0.44 / 0.48 / 0.53 | 0.43 / 0.47 / 0.55 |
| Dlinear | No ESE | 1.47 / 2.93 / 5.53 | 1.39 / 2.93 / 5.80 | 1.41 / 2.75 / 5.68 |
| | With ESE | 0.48 / 0.53 / 0.55 | 0.47 / 0.52 / 0.58 | 0.48 / 0.51 / 0.59 |
| Informer | No ESE | 0.90 / 1.71 / 3.44 | 0.86 / 1.69 / 3.40 | 0.83 / 1.66 / 3.38 |
| | With ESE | 0.37 / 0.41 / 0.45 | 0.37 / 0.40 / 0.46 | 0.36 / 0.40 / 0.47 |
| DeepAR | No ESE | 1.17 / 2.35 / 4.54 | 1.10 / 2.29 / 4.53 | 1.16 / 2.37 / 4.76 |
| | With ESE | 0.61 / 0.51 / 0.57 | 0.51 / 0.53 / 0.52 | 0.51 / 0.64 / 0.53 |
| PatchTST | No ESE | 0.90 / 1.87 / 3.78 | 0.90 / 1.84 / 3.70 | 0.94 / 1.83 / 3.67 |
| | With ESE | 0.37 / 0.43 / 0.46 | 0.38 / 0.41 / 0.48 | 0.38 / 0.42 / 0.49 |

Table 11: Comparison of computational cost (in minutes) using 5/10/20 systems on synthetic data. The experiments are repeated for three input lengths: 10, 20 and 50. The prediction target is fixed at a horizon of 2. We can observe that **(1)** ESE incurs significantly lower computational cost than other methods, except ARIMA, which requires no training, hence is fast on small datasets, but not on larger ones; **(2)** When paired with ESE, the cost of SOTA predictors can be reduced significantly, by a factor of 2-10, excluding ARIMA; **(3)** The runtime increases with the number of systems estimated.

| Models | | Costs (mins) for Horizon = 2 | | |
| | | *Input Length* = 10 | *Input Length* = 20 | *Input Length* = 50 |
|---|---|---|---|---|
| ESE | | 0.18 / 0.25 / 0.28 | 0.20 / 0.24 / 0.28 | 0.19 / 0.24 / 0.31 |
| ARIMA | No ESE | **0.04 / 0.09 / 0.18** | **0.04 / 0.09 / 0.17** | **0.04 / 0.08 / 0.17** |
| | With ESE | 0.19 / 0.26 / 0.29 | 0.21 / 0.25 / 0.29 | 0.20 / 0.25 / 0.32 |
| LSTM | No ESE | 1.27 / 2.50 / 4.76 | 1.28 / 2.37 / 5.10 | 1.21 / 2.47 / 4.89 |
| | With ESE | 0.43 / 0.50 / 0.51 | 0.46 / 0.48 / 0.54 | 0.43 / 0.48 / 0.56 |
| Dlinear | No ESE | 1.47 / 2.92 / 5.89 | 1.48 / 2.90 / 5.92 | 1.36 / 2.82 / 5.71 |
| | With ESE | 0.47 / 0.54 / 0.57 | 0.50 / 0.53 / 0.58 | 0.46 / 0.52 / 0.60 |
| Informer | No ESE | 0.88 / 1.80 / 3.31 | 0.90 / 1.76 / 3.51 | 0.84 / 1.68 / 3.49 |
| | With ESE | 0.36 / 0.43 / 0.44 | 0.38 / 0.42 / 0.46 | 0.36 / 0.41 / 0.48 |
| DeepAR | No ESE | 1.14 / 2.31 / 4.68 | 1.11 / 2.41 / 4.79 | 1.15 / 2.29 / 4.41 |
| | With ESE | 0.41 / 0.48 / 0.51 | 0.42 / 0.48 / 0.52 | 0.42 / 0.47 / 0.53 |
| PatchTST | No ESE | 0.97 / 1.82 / 3.67 | 0.95 / 1.91 / 3.59 | 0.97 / 1.96 / 3.63 |
| | With ESE | 0.37 / 0.43 / 0.46 | 0.39 / 0.43 / 0.46 | 0.38 / 0.43 / 0.49 |

Table 12: Comparison of computational cost (in minutes) using 5/10/20 systems on synthetic data. The experiments are repeated for three input lengths: 10, 20 and 50. The prediction target is fixed at a horizon of 5. We can observe that **(1)** ESE incurs significantly lower computational cost than other methods, except ARIMA, which requires no training, hence is fast on small datasets, but not on larger ones; **(2)** When paired with ESE, the cost of SOTA predictors can be reduced significantly, by a factor of 2-10, excluding ARIMA; **(3)** The runtime increases with the number of systems estimated.

| Models | | Costs (mins) for Horizon = 5 | | |
| :---: | :---: | :---: | :---: | :---: |
| | | *Input Length = 10* | *Input Length = 20* | *Input Length = 50* |
| ESE | | 0.20 / 0.25 / 0.29 | 0.20 / 0.24 / 0.33 | 0.21 / 0.24 / 0.35 |
| ARIMA | No ESE | **0.05 / 0.09 / 0.17** | **0.05 / 0.09 / 0.17** | **0.05 / 0.09 / 0.18** |
| | With ESE | 0.21 / 0.26 / 0.29 | 0.21 / 0.24 / 0.34 | 0.22 / 0.25 / 0.36 |
| LSTM | No ESE | 1.27 / 2.56 / 5.12 | 1.27 / 2.55 / 4.93 | 1.24 / 2.38 / 4.74 |
| | With ESE | 0.45 / 0.50 / 0.54 | 0.46 / 0.49 / 0.58 | 0.46 / 0.48 / 0.59 |
| Dlinear | No ESE | 1.40 / 2.80 / 5.55 | 1.46 / 2.99 / 5.91 | 1.41 / 2.87 / 5.96 |
| | With ESE | 0.48 / 0.53 / 0.56 | 0.50 / 0.53 / 0.63 | 0.50 / 0.52 / 0.65 |
| Informer | No ESE | 0.84 / 1.77 / 3.52 | 0.88 / 1.65 / 3.35 | 0.85 / 1.82 / 3.48 |
| | With ESE | 0.37 / 0.42 / 0.46 | 0.38 / 0.40 / 0.50 | 0.38 / 0.42 / 0.52 |
| DeepAR | No ESE | 1.14 / 2.29 / 4.68 | 1.19 / 2.39 / 4.73 | 1.17 / 2.38 / 4.63 |
| | With ESE | 0.43 / 0.47 / 0.52 | 0.44 / 0.48 / 0.57 | 0.45 / 0.47 / 0.58 |
| PatchTST | No ESE | 0.90 / 1.96 / 3.77 | 0.95 / 1.93 / 3.82 | 0.92 / 1.91 / 3.73 |
| | With ESE | 0.38 / 0.44 / 0.47 | 0.39 / 0.43 / 0.52 | 0.40 / 0.43 / 0.54 |

# I CURRENCY EXCHANGE RATE DATA

Table 13: G20 non-USD currencies and their issuing countries.

| Code | Country and Region | Currency |
| :---: | :--- | :---: |
| ARS | Argentina | Argentine Peso |
| AUD | Australia | Australian Dollar |
| BRL | Brazil | Brazilian Real |
| CAD | Canada | Canadian Dollar |
| CNY | China | Renminbi (Yuan) |
| EUR | France; Germany; Italy; European Union | Euro (EUR) |
| GBP | United Kingdom | Pound Sterling |
| IDR | Indonesia | Indonesian Rupiah |
| INR | India | Indian Rupee |
| JPY | Japan | Japanese Yen |
| KRW | South Korea | South Korean Won |
| MXN | Mexico | Mexican Peso |
| RUB | Russia | Russian Ruble |
| SAR | Saudi Arabia | Saudi Riyal |
| TRY | Turkey | Turkish Lira |
| ZAR | South African | South African Rand |

The G20 includes 19 major countries and 17 major currencies, with Table 13 displaying 16 non-USD currencies and their corresponding countries or organizations. The US dollar is excluded as it is the base currency against all other currencies, hence always considered as 1. We collected daily exchange rates of all currencies' ratios to the USD from November 11, 2019 to October 31, 2024, for 5 years.

Table 14 describes the six columns of collected exchange rate data, including, the date, the opening price, highest price, lowest price, closing price, and the adjusted closing price. The last column, 'Adj Closed', is used as the target.

Table 14: Six columns of exchange rate daily data

| Variable | Description | Data Type |
|----------|-------------|-----------|
| Date | Trading date | Date |
| Open | Opening price of the day | Float |
| High | Highest price of the day | Float |
| Low | Lowest price of the day | Float |
| Close | Closing price of the day | Float |
| Adj Close | Adjusted closing price | Float |

Table 15: Attribute variables descriptions for 16 countries or regions

| Variable | Description | Data Type |
|----------|-------------|-----------|
| DateTime | Date | Date |
| Population (million) | Total population | Float |
| GDP (billion USD) | Gross Domestic Product in USD | Float |
| External Debt (million USD) | Total external debt in USD | Float |
| Gold Reserves (tonnes) | Total gold reserves in tonnes | Float |
| Current Account (billion USD) | Current account balance in USD | Float |
| Inflation Rate | Annual inflation rate in percentage | Float |
| Import (million USD) | Total imports in USD | Float |
| Export (million USD) | Total exports in USD | Float |
| Foreign Direct Investment (million USD) | Foreign direct investment in USD | Float |
| Tourist Arrivals | Number of tourist arrivals | Float |
| Food Inflation | Food inflation rate in percentage | Float |
| Consumer Price Index CPI | Consumer Price Index | Float |
| Money Supply M1 (billion USD) | Narrow money supply in USD | Float |
| Money Supply M2 (billion USD) | Broad money supply in USD | Float |
| Money Supply M3 (billion USD) | Total money supply in USD | Float |

Table 15 list the economic and financial indicators for G20 countries for the same time periods of exchange rate data. It includes data such as the population, GDP, external debt, gold reserves, current account balance, inflation rate, trade figures (imports and exports), foreign direct investment, and various monetary supply measures. Additionally, we include tourist arrivals, food inflation, and the consumer price index (CPI) as attributes. It should be noted that the attribute data related to the Euro is from the European Union rather than individual countries such as France or Germany. The data is sourced from World Bank Open Data and the International Monetary Fund.

## J FULL COMPARISON ON EXCHANGE RATE DATA

### J.1 COMPARING PREDICTION PERFORMANCE - CURRENCY EXCHANGE RATES

Table 16: Comparison of prediction performance on exchange rate data with four horizons (1/2/5/10). RMSE and MAE rows show the means of all runs of one method over 16 currencies. The experiments are repeated for four input lengths: 20, 50, 100 and 200. Row "No ESE" is the method acting alone, while Row "With ESE" is the method augmented with ESE. We can observe that **(1)** ESE alone is a top-tier predictor, if not the best, in most cases, especially for longer input lengths; **(2)** ESE consistently improves the performance for most SOTA predictors.

| Models | | Metric | Prediction Horizon = 1/2/5/10 | | | |
| | | | Input Length = 20 | Input Length = 50 | Input Length = 100 | Input Length = 200 |
|---|---|---|---|---|---|---|
| ESE | – | RMSE | 6.86 / 8.44 / 12.38 / 19.88 | 6.48 / 8.17 / 11.20 / 15.97 | 6.01 / 7.60 / 10.69 / 15.31 | 6.10 / 7.50 / 10.30 / 16.26 |
| | | MAE | 6.03 / 7.33 / 11.03 / 17.54 | 5.38 / 6.82 / 9.82 / 15.73 | 5.52 / 6.75 / 9.87 / 14.49 | 5.26 / 6.57 / **9.29** / 13.96 |
| VAR | – | RMSE | 6.86 / 8.67 / 14.79 / 24.53 | 6.46 / 8.08 / 11.28 / 17.07 | 6.82 / 7.84 / 10.84 / 15.46 | 6.33 / 8.19 / 11.32 / 15.69 |
| | | MAE | 5.91 / 7.35 / 12.90 / 21.32 | 5.74 / 7.08 / 10.23 / 15.64 | 5.92 / 7.10 / 9.93 / 15.03 | 5.39 / 6.65 / 9.44 / 14.13 |
| ARIMA | No ESE | RMSE | 6.86 / 8.67 / 14.79 / 24.53 | 6.46 / 8.08 / 11.28 / 17.07 | 6.82 / 7.84 / 10.84 / 15.46 | 6.33 / 8.19 / 11.32 / 15.69 |
| | | MAE | 5.91 / 7.27 / 11.50 / 18.90 | 5.60 / 7.05 / 10.24 / 15.29 | 5.68 / 7.02 / 10.06 / 15.14 | 5.28 / 6.65 / 9.53 / 13.93 |
| | With ESE | RMSE | 7.44 / 9.06 / 14.20 / 22.87 | 6.52 / 8.03 / 11.46 / 15.83 | 5.71 / 7.63 / 10.58 / 14.85 | 5.56 / 7.02 / 9.23 / 13.36 |
| | | MAE | 7.06 / 8.18 / 12.42 / 20.02 | 5.73 / 7.14 / 10.22 / 15.16 | 5.29 / 6.84 / 9.62 / 13.55 | 4.93 / 6.22 / 8.66 / 12.89 |
| LSTM | No ESE | RMSE | 6.61 / 8.47 / 12.46 / 18.79 | 6.27 / 8.01 / 11.05 / 16.21 | 6.17 / 7.71 / 10.77 / 15.37 | 6.05 / 8.061 / 10.87 / 16.02 |
| | | MAE | 5.81 / 7.24 / 10.74 / 16.41 | 5.69 / 6.84 / 9.83 / 15.70 | 5.67 / 6.87 / 9.87 / 14.71 | 5.39 / 6.58 / 9.26 / 13.91 |
| | With ESE | RMSE | 7.13 / 8.97 / 13.50 / 21.35 | 6.33 / 7.83 / 11.28 / 15.65 | 5.62 / 7.58 / 10.54 / 14.71 | 5.31 / 6.75 / 9.07 / 12.81 |
| | | MAE | 7.05 / 8.15 / 11.75 / 17.65 | 5.61 / 6.81 / 10.01 / 14.61 | 5.26 / 6.73 / 9.49 / 13.36 | 5.01 / 6.13 / 8.48 / 12.62 |
| Dlinear | No ESE | RMSE | 6.55 / 8.16 / **11.74** / **17.92** | 6.29 / 8.04 / 10.99 / 15.99 | 5.88 / 7.55 / 10.59 / 15.18 | 6.05 / 7.65 / 10.98 / 16.24 |
| | | MAE | 5.82 / 7.13 / **10.15** / **15.68** | 5.38 / 6.81 / 9.86 / 15.26 | 5.41 / 6.68 / 9.71 / 14.21 | 5.22 / 6.60 / 9.36 / 14.03 |
| | With ESE | RMSE | 7.06 / 8.73 / 13.3 / 19.31 | 6.19 / 7.95 / 11.10 / 15.43 | **5.46** / 7.41 / 10.37 / 14.55 | 5.14 / 6.61 / 9.06 / 12.72 |
| | | MAE | 6.90 / 8.26 / 11.09 / 17.02 | 5.40 / 6.76 / 9.81 / 14.12 | **5.10** / **6.57** / 9.45 / 12.95 | **4.91** / 6.09 / 8.39 / 12.49 |
| Nlinear | No ESE | RMSE | 6.49 / 8.22 / 11.90 / 18.08 | 6.34 / 8.00 / 10.96 / 16.34 | 5.96 / 7.58 / 10.60 / 15.06 | 6.11 / 7.76 / 10.87 / 16.00 |
| | | MAE | 5.80 / 7.12 / 10.35 / 15.98 | 5.39 / 6.75 / **9.70** / 15.56 | 5.46 / 6.79 / 9.85 / 14.17 | 5.25 / 6.65 / 9.41 / 14.06 |
| | With ESE | RMSE | 7.05 / 8.76 / 13.61 / 19.70 | 6.34 / 7.92 / 11.11 / 15.49 | 5.52 / 7.44 / 10.36 / **14.54** | 5.19 / 6.65 / 9.07 / 12.76 |
| | | MAE | 6.91 / 8.12 / 11.29 / 17.27 | 5.53 / 6.72 / 9.82 / 14.26 | 5.17 / 6.62 / 9.40 / 12.96 | 4.91 / 6.12 / 8.43 / 12.58 |
| Informer | No ESE | RMSE | 6.74 / 8.23 / 11.99 / 20.21 | 6.44 / 7.79 / 11.22 / 16.12 | 6.01 / 7.67 / 10.60 / 15.29 | 5.83 / 7.59 / 10.56 / 14.98 |
| | | MAE | 5.80 / 7.12 / 10.35 / 15.98 | 5.39 / 6.75 / 9.70 / 15.56 | 5.46 / 6.79 / 9.85 / 14.17 | 5.25 / 6.65 / 9.41 / 14.06 |
| | With ESE | RMSE | 7.27 / 8.81 / 13.43 / 21.22 | 6.49 / 7.89 / 11.19 / 15.59 | 5.56 / 7.49 / 10.54 / 14.66 | 5.27 / 6.77 / 9.12 / 12.95 |
| | | MAE | 6.82 / 8.06 / 11.59 / 18.30 | 5.55 / 7.07 / 9.88 / 14.47 | 5.20 / 6.61 / 9.59 / 13.34 | 5.00 / 6.17 / 8.61 / 12.66 |
| FiLM | No ESE | RMSE | 6.83 / 8.41 / 12.17 / 19.70 | 6.50 / 8.15 / 11.29 / 16.58 | 5.93 / 7.72 / 10.76 / 15.34 | 6.02 / 7.88 / 11.32 / 14.90 |
| | | MAE | 5.89 / 7.20 / 10.81 / 17.13 | 5.50 / 6.76 / 10.03 / 15.25 | 5.54 / 7.02 / 9.73 / 14.45 | 5.34 / 6.63 / 9.42 / 13.95 |
| | With ESE | RMSE | 7.09 / 8.72 / 13.76 / 20.42 | 6.57 / 8.12 / 11.29 / 15.67 | 5.51 / 7.47 / 10.54 / 14.57 | 5.32 / 6.80 / 9.18 / 12.96 |
| | | MAE | 6.93 / 8.20 / 11.78 / 18.32 | 5.70 / 6.93 / 10.07 / 14.63 | 5.22 / 6.84 / 9.40 / 13.40 | 5.04 / 6.19 / 8.63 / 12.76 |
| SCINet | No ESE | RMSE | **6.42** / 8.22 / 11.94 / 18.27 | 6.14 / 7.98 / 10.93 / 16.02 | 5.90 / 7.60 / 10.47 / 15.30 | 5.85 / 7.88 / 10.84 / 15.85 |
| | | MAE | **5.62** / 7.15 / 10.42 / 16.11 | 5.50 / **6.64** / 9.90 / 15.26 | 5.50 / 6.94 / 9.58 / 14.87 | 5.28 / 6.53 / 9.41 / 13.93 |
| | With ESE | RMSE | 7.02 / 8.64 / 13.28 / 19.98 | 6.27 / 8.09 / **10.92** / **15.42** | 5.48 / 7.44 / **10.33** / 14.57 | **5.10** / 6.75 / 9.11 / 12.89 |
| | | MAE | 6.73 / 8.07 / 11.48 / 17.60 | 5.51 / 6.83 / 9.75 / **14.05** | 5.15 / 6.65 / 9.31 / 13.28 | 5.00 / 6.14 / 8.56 / 12.64 |
| DeepAR | No ESE | RMSE | 6.83 / 8.46 / 12.32 / 19.64 | 6.22 / 7.95 / 11.13 / 16.53 | 6.08 / 7.62 / 10.62 / 15.20 | 6.37 / 7.91 / 10.90 / 15.77 |
| | | MAE | 5.93 / 7.26 / 10.91 / 17.14 | 5.59 / 7.10 / 10.20 / 15.14 | 5.61 / 6.86 / 9.90 / 14.53 | 5.29 / 6.70 / 9.38 / 13.91 |
| | With ESE | RMSE | 7.30 / 8.84 / 13.61 / 21.36 | 6.40 / 7.93 / 11.40 / 15.79 | 5.62 / 7.52 / 10.53 / 14.65 | 5.48 / 6.84 / 9.22 / 13.05 |
| | | MAE | 7.13 / 8.29 / 11.79 / 18.42 | 5.79 / 7.11 / 10.21 / 15.02 | 5.29 / 6.74 / 9.61 / 13.59 | 4.97 / 6.21 / 8.65 / 12.82 |
| KVAE | No ESE | RMSE | 6.76 / 8.37 / 12.26 / 18.84 | 6.57 / 8.10 / 11.30 / 16.61 | 6.05 / 7.65 / 10.81 / 15.24 | 6.17 / 7.71 / 10.94 / 15.86 |
| | | MAE | 5.72 / 7.05 / 10.80 / 16.50 | 5.46 / 6.69 / 10.24 / 15.31 | 5.59 / 6.94 / 9.83 / 14.56 | 5.29 / 6.58 / 9.30 / 14.01 |
| | With ESE | RMSE | 7.43 / 9.00 / 13.35 / 21.69 | 6.48 / 8.18 / 11.30 / 15.69 | 5.54 / 7.56 / 10.67 / 14.75 | 5.33 / 6.78 / 9.12 / 12.95 |
| | | MAE | 6.83 / 8.05 / 11.76 / 17.87 | 5.45 / 6.84 / 10.13 / 14.89 | 5.19 / 6.84 / 9.49 / 13.63 | 5.00 / 6.19 / 8.63 / 12.76 |
| TPGNN | No ESE | RMSE | 6.59 / **7.96** / 12.00 / 19.81 | **6.11** / 7.81 / 11.06 / 16.21 | 6.09 / 7.53 / 10.64 / 15.24 | 6.13 / 7.77 / 10.88 / 15.86 |
| | | MAE | 5.86 / 7.17 / 10.51 / 17.02 | 5.59 / 7.05 / 10.07 / 15.35 | 5.63 / 6.76 / 9.75 / 14.88 | 5.19 / 6.53 / 9.37 / 13.97 |
| | With ESE | RMSE | 7.06 / 8.51 / 13.03 / 20.37 | 6.31 / **7.76** / 11.25 / 15.63 | 5.51 / 7.45 / 10.51 / 14.81 | 5.43 / 6.58 / 9.03 / 12.63 |
| | | MAE | 7.04 / 8.31 / 11.56 / 18.33 | 5.70 / 7.05 / 9.97 / 14.54 | 5.18 / 6.72 / 9.51 / 13.48 | 4.94 / 6.06 / 8.40 / 12.45 |
| PatchTST | No ESE | RMSE | 6.82 / 8.29 / 11.93 / 18.73 | 6.33 / 7.74 / 11.13 / 16.33 | 6.04 / 7.45 / 10.63 / 15.32 | 6.09 / 7.50 / 10.43 / 16.20 |
| | | MAE | 5.85 / **6.99** / 10.50 / 16.48 | **5.29** / 6.89 / 9.82 / 15.73 | 5.57 / 6.72 / 9.63 / 14.70 | 5.30 / 6.51 / 9.32 / 13.86 |
| | With ESE | RMSE | 7.15 / 8.95 / 12.73 / 20.27 | 6.46 / 7.80 / 11.16 / 15.61 | 5.55 / **7.41** / 10.39 / 14.61 | 5.25 / **6.55** / **9.02** / **12.53** |
| | | MAE | 6.97 / 7.94 / 11.54 / 17.77 | 5.52 / 6.87 / 9.83 / 14.37 | 5.20 / 6.60 / **9.30** / 13.32 | 4.92 / **5.98** / 8.36 / **12.30** |

Table 17: Comparison of normalized prediction performance on exchange rate data with four horizons (1/2/5/10). RMSE* and MAE* rows show the means of normalized means (RMSE* ($\times 100$) and MAE* ($\times 100$)) of one method over 16 currencies. The experiments are repeated for four input lengths: 20, 50, 100 and 200. Row "No ESE" is the method acting alone, while Row "With ESE" is the method augmented with ESE. We can observe that **(1)** ESE alone is a top-tier predictor, if not the best, in most cases, especially for longer input lengths; **(2)** ESE consistently improves the performance for most SOTA predictors.

| Models | | Metric | Prediction Horizon = 1/2/5/10 | | | |
| --- | --- | --- | --- | --- | --- | --- |
| | | | *Input Length = 20* | *Input Length = 50* | *Input Length = 100* | *Input Length = 200* |
| ESE | – | RMSE* | 1.24 / 1.49 / 2.29 / 3.58 | 1.25 / 1.51 / 2.13 / 3.06 | 1.18 / 1.50 / 2.08 / 2.91 | 1.20 / 1.49 / 2.05 / 2.79 |
| | | MAE* | 0.63 / 0.85 / 1.40 / 2.25 | 0.60 / 0.80 / 1.23 / 1.81 | 0.60 / 0.82 / 1.24 / 1.81 | 0.59 / 0.79 / 1.21 / 1.75 |
| VAR | – | RMSE* | 1.27 / 1.58 / 2.79 / 4.55 | 1.29 / 1.54 / 2.19 / 3.19 | 1.25 / 1.48 / 2.11 / 2.99 | 1.22 / 1.50 / 2.07 / 2.85 |
| | | MAE* | 0.64 / 0.86 / 1.60 / 2.68 | 0.62 / 0.81 / 1.25 / 1.83 | 0.65 / 0.81 / 1.26 / 1.83 | 0.62 / 0.81 / 1.23 / 1.77 |
| ARIMA | No ESE | RMSE* | 1.26 / 1.53 / 2.45 / 3.91 | 1.27 / 1.54 / 2.18 / 3.16 | 1.28 / 1.54 / 2.11 / 2.96 | 1.24 / 1.49 / 2.06 / 2.84 |
| | | MAE* | 0.65 / 0.84 / 1.46 / 2.38 | 0.61 / 0.81 / 1.25 / 1.83 | 0.63 / 0.83 / 1.25 / 1.81 | 0.60 / 0.80 / 1.22 / 1.77 |
| | With ESE | RMSE* | 1.55 / 1.79 / 2.55 / 4.72 | 1.25 / 1.59 / 2.19 / 3.15 | 1.20 / 1.48 / 2.05 / 2.71 | 1.02 / 1.16 / 1.37 / 2.06 |
| | | MAE* | 0.81 / 0.91 / 1.57 / 2.50 | 0.68 / 0.85 / 1.31 / 1.87 | 0.60 / 0.82 / 1.15 / 1.60 | 0.45 / 0.57 / 0.89 / 1.48 |
| LSTM | No ESE | RMSE* | 1.23 / 1.53 / 2.17 / 3.52 | 1.24 / **1.49** / 2.18 / 3.08 | 1.21 / 1.48 / 2.09 / 2.92 | 1.23 / 1.48 / 2.05 / 2.82 |
| | | MAE* | 0.66 / 0.85 / 1.34 / 2.05 | 0.60 / 0.79 / 1.24 / 1.82 | 0.60 / 0.81 / 1.24 / 1.80 | 0.60 / 0.80 / 1.23 / 1.77 |
| | With ESE | RMSE* | 1.53 / 1.79 / 2.48 / 4.58 | 1.24 / 1.57 / 2.17 / 3.10 | 1.19 / 1.48 / 2.05 / 2.69 | 1.00 / 1.13 / 1.35 / 2.01 |
| | | MAE* | 0.81 / 0.91 / 1.51 / 2.30 | 0.67 / 0.82 / 1.29 / 1.85 | 0.60 / 0.80 / 1.14 / 1.58 | 0.46 / 0.56 / 0.88 / 1.46 |
| Dlinear | No ESE | RMSE* | 1.21 / 1.52 / 2.24 / **3.36** | 1.23 / 1.51 / **2.11** / 3.01 | 1.21 / **1.43** / 2.09 / 2.84 | 1.21 / 1.48 / 2.04 / 2.81 |
| | | MAE* | 0.65 / 0.86 / **1.29** / **1.92** | **0.60** / **0.79** / 1.22 / 1.80 | 0.60 / 0.80 / 1.25 / 1.78 | 0.60 / 0.80 / 1.23 / 1.76 |
| | With ESE | RMSE* | 1.52 / 1.76 / 2.47 / 4.40 | **1.22** / 1.58 / 2.16 / 3.05 | 1.17 / 1.46 / 2.03 / **2.68** | 0.98 / **1.12** / **1.35** / 2.00 |
| | | MAE* | 0.79 / 0.92 / 1.45 / 2.23 | 0.65 / 0.81 / 1.27 / 1.83 | **0.59** / 0.79 / 1.13 / **1.55** | 0.45 / 0.56 / 0.87 / 1.45 |
| Nlinear | No ESE | RMSE* | 1.22 / 1.50 / 2.24 / 3.39 | 1.25 / 1.51 / 2.16 / 3.06 | 1.21 / 1.47 / 2.06 / 2.85 | 1.20 / 1.47 / 2.04 / 2.82 |
| | | MAE* | 0.65 / 0.86 / 1.30 / 1.95 | 0.60 / 0.80 / 1.24 / 1.81 | 0.61 / 0.81 / 1.23 / 1.78 | 0.60 / 0.79 / 1.22 / 1.76 |
| | With ESE | RMSE* | 1.52 / 1.76 / 2.49 / 4.44 | 1.24 / 1.58 / 2.16 / 3.06 | 1.18 / 1.46 / 2.03 / 2.68 | 0.99 / 1.13 / 1.35 / 2.00 |
| | | MAE* | 0.80 / 0.91 / 1.47 / 2.26 | 0.66 / 0.81 / 1.27 / 1.84 | 0.59 / 0.80 / 1.13 / 1.55 | 0.45 / 0.56 / 0.87 / 1.45 |
| Informer | No ESE | RMSE* | 1.27 / 1.52 / 2.21 / 3.52 | 1.26 / 1.52 / 2.15 / 3.06 | 1.22 / 1.46 / 2.10 / 2.92 | 1.24 / 1.56 / 2.08 / 2.83 |
| | | MAE* | 0.63 / 0.85 / 1.33 / 2.08 | 0.60/ 0.80 / 1.24 / 1.81 | 0.61 / 0.81 / 1.25 / 1.80 | 0.60 / 0.80 / 1.22 / 1.77 |
| | With ESE | RMSE* | 1.54 / 1.77 / 2.48 / 4.57 | 1.25 / 1.58 / 2.17 / 3.09 | 1.18 / 1.47 / 2.04 / 2.69 | 0.99 / 1.14 / 1.36 / 2.02 |
| | | MAE* | 0.79 / 0.90 / 1.50 / 2.35 | 0.66 / 0.84 / 1.28 / 1.85 | 0.60 / 0.79 / 1.15 / 1.58 | 0.46 / 0.56 / 0.89 / 1.46 |
| FiLM | No ESE | RMSE* | 1.21 / **1.48** / 2.24 / 3.49 | 1.27 / 1.52 / 2.15 / 3.10 | 1.22 / 1.46 / 2.07 / 2.90 | 1.33 / 1.49 / 2.05 / 2.83 |
| | | MAE* | 0.65 / 0.85 / 1.37 / 2.13 | 0.61 / 0.80 / 1.24 / 1.82 | 0.61 / 0.80 / 1.24 / 1.81 | 0.62 / 0.80 / 1.23 / 1.77 |
| | With ESE | RMSE* | 1.52 / 1.76 / 2.51 / 4.50 | 1.26 / 1.60 / 2.18 / 3.10 | 1.18 / 1.47 / 2.05 / 2.68 | 1.00 / 1.14 / 1.37 / 2.02 |
| | | MAE* | 0.80 / 0.91 / 1.51 / 2.35 | 0.67 / 0.83 / 1.29 / 1.86 | 0.60 / 0.81 / 1.13 / 1.59 | 0.46 / 0.57 / 0.89 / 1.47 |
| SCINet | No ESE | RMSE* | **1.17** / 1.49 / 2.22 / 3.41 | 1.25 / 1.55 / 2.12 / **3.00** | 1.18 / 1.46 / 2.06 / 2.90 | 1.21 / 1.48 / 2.07 / 2.82 |
| | | MAE* | **0.62** / 0.85 / 1.33 / 1.98 | 0.60 / 0.80 / **1.22** / **1.80** | 0.60 / 0.80 / 1.23 / 1.79 | 0.60 / 0.80 / 1.22 / 1.77 |
| | With ESE | RMSE* | 1.52 / 1.75 / 2.47 / 4.46 | 1.23 / 1.60 / 2.14 / 3.05 | **1.17** / 1.46 / **2.03** / 2.68 | **0.98** / 1.13 / 1.36 / 2.01 |
| | | MAE* | 0.78 / 0.90 / 1.49 / 2.29 | 0.66 / 0.82 / 1.27 / 1.83 | 0.59 / 0.80 / **1.12** / 1.58 | 0.46 / 0.56 / 0.89 / 1.46 |
| DeepAR | No ESE | RMSE* | 1.25 / 1.51 / 2.21 / 3.53 | 1.27 / 1.51 / 2.19 / 3.16 | 1.22 / 1.48 / 2.11 / 2.91 | 1.23 / 1.49 / 2.07 / 2.84 |
| | | MAE* | 0.65 / 0.85 / 1.35 / 2.18 | 0.61 / 0.80 / 1.25 / 1.83 | 0.61 / 0.81 / 1.25 / 1.81 | 0.60 / 0.80 / 1.23 / 1.77 |
| | With ESE | RMSE* | 1.54 / 1.78 / 2.50 / 4.59 | 1.24 / 1.58 / 2.18 / 3.14 | 1.19 / 1.47 / 2.05 / 2.69 | 1.01 / 1.14 / 1.37 / 2.03 |
| | | MAE* | 0.82 / 0.92 / 1.52 / 2.36 | 0.68 / 0.84 / 1.31 / 1.87 | 0.60 / 0.81 / 1.15 / 1.60 | 0.46 / 0.57 / 0.89 / 1.48 |
| KVAE | No ESE | RMSE* | 1.28 / 1.55 / 2.20 / 3.53 | 1.24 / 1.54 / 2.13 / 3.14 | 1.20 / 1.49 / 2.11 / 2.93 | 1.22 / 1.49 / 2.06 / 2.83 |
| | | MAE* | 0.63 / 0.84 / 1.35 / 2.07 | 0.60 / 0.80 / 1.23 / 1.82 | 0.60 / 0.81 / 1.25 / 1.82 | 0.61 / 0.81 / 1.23 / 1.77 |
| | With ESE | RMSE* | 1.55 / 1.78 / 2.47 / 4.61 | 1.25 / 1.60 / 2.17 / 3.13 | 1.18 / 1.47 / 2.06 / 2.70 | 1.00 / 1.14 / 1.36 / 2.02 |
| | | MAE* | 0.79 / 0.90 / 1.51 / 2.31 | 0.65 / 0.82 / 1.30 / 1.86 | 0.60 / 0.81 / 1.14 / 1.61 | 0.46 / 0.57 / 0.89 / 1.47 |
| TPGNN | No ESE | RMSE* | 1.22 / 1.51 / 2.21 / 3.47 | 1.25 / 1.50 / 2.17 / 3.07 | 1.18 / 1.48 / 2.10 / 2.96 | 1.23 / 1.47 / 2.04 / 2.81 |
| | | MAE* | 0.66 / 0.86 / 1.31 / 2.13 | 0.60 / 0.80 / 1.23 / 1.82 | 0.60 / 0.81 / 1.26 / 1.81 | 0.60 / 0.80 / 1.22 / 1.76 |
| | With ESE | RMSE* | 1.52 / 1.74 / 2.44 / 4.49 | 1.23 / 1.56 / 2.17 / 3.09 | 1.18 / 1.46 / 2.05 / 2.70 | 1.01 / 1.12 / 1.35 / 1.99 |
| | | MAE* | 0.82 / 0.92 / 1.49 / 2.35 | 0.67 / 0.84 / 1.28 / 1.85 | 0.59 / 0.80 / 1.14 / 1.59 | 0.45 / 0.55 / 0.87 / 1.44 |
| PatchTST | No ESE | RMSE* | 1.23 / 1.55 / **2.17** / 3.46 | 1.28 / 1.52 / 2.16 / 3.06 | 1.21 / 1.46 / 2.04 / 2.91 | 1.22 / 1.46 / 2.05 / 2.78 |
| | | MAE* | 0.65 / **0.83** / 1.32 / 2.05 | 0.61 / 0.80 / 1.23 / 1.81 | 0.61 / 0.80 / 1.23 / 1.80 | 0.60 / 0.80 / 1.22 / 1.74 |
| | With ESE | RMSE* | 1.53 / 1.78 / 2.42 / 4.49 | 1.25 / 1.57 / 2.16 / 3.08 | 1.18 / 1.46 / 2.03 / 2.68 | 0.99 / 1.12 / 1.35 / **1.98** |
| | | MAE* | 0.80 / 0.89 / 1.49 / 2.30 | 0.66 / 0.82 / 1.27 / 1.85 | 0.60 / **0.79** / 1.12 / 1.58 | **0.45** / **0.55** / **0.87** / **1.43** |

## J.2 Comparing Computational Costs - Currency Exchange Rates

Table 18: Comparison of computational cost (in minutes) on exchange rate data with four horizons (1/2/5/10) over 16 currencies. The experiments are repeated for four input lengths: 20, 50, 100 and 200. Row "No ESE" is the method acting alone, while Row "With ESE" is the method augmented with ESE. We can observe that ESE substantially reduces computational cost. Since VAR requires manual parameter tuning, its computational cost is excluded from the comparison.

| Models | | Prediction Horizon = 1/2/5/10 | | | |
|---|---|---|---|---|---|
| | | *Input Length* = 20 | *Input Length* = 50 | *Input Length* = 100 | *Input Length* = 200 |
| ESE | – | 0.21 / 0.21 / **0.22** / **0.23** | 0.21 / **0.21** / 0.23 / **0.23** | 0.22 / **0.21** / **0.23** / **0.23** | 0.22 / **0.21** / **0.24** / **0.24** |
| ARIMA | No ESE | **0.18** / **0.21** / 0.26 / 0.29 | **0.18** / 0.23 / 0.27 / 0.30 | **0.19** / 0.24 / 0.29 / 0.31 | **0.19** / 0.25 / 0.27 / 0.30 |
| | With ESE | 0.22 / 0.22 / 0.24 / 0.24 | 0.22 / 0.22 / **0.22** / 0.23 | 0.24 / 0.24 / 0.25 / 0.26 | 0.24 / 0.24 / 0.24 / 0.26 |
| LSTM | No ESE | 5.33 / 6.55 / 7.65 / 8.22 | 5.32 / 6.56 / 8.11 / 8.99 | 5.71 / 6.81 / 8.46 / 9.28 | 5.78 / 7.22 / 8.93 / 9.55 |
| | With ESE | 0.54 / 0.62 / 0.70 / 0.77 | 0.54 / 0.62 / 0.71 / 0.80 | 0.58 / 0.65 / 0.76 / 0.85 | 0.59 / 0.68 / 0.79 / 0.89 |
| Dlinear | No ESE | 6.22 / 8.00 / 8.27 / 8.88 | 6.16 / 7.52 / 8.89 / 9.74 | 6.33 / 8.25 / 8.93 / 9.88 | 7.05 / 8.13 / 9.40 / 10.02 |
| | With ESE | 0.60 / 0.71 / 0.74 / 0.81 | 0.59 / 0.68 / 0.76 / 0.85 | 0.62 / 0.74 / 0.79 / 0.89 | 0.67 / 0.73 / 0.82 / 0.90 |
| Nlinear | No ESE | 6.34 / 7.69 / 8.83 / 10.11 | 6.39 / 7.75 / 9.01 / 9.81 | 6.71 / 8.30 / 9.08 / 9.85 | 6.94 / 8.28 / 9.18 / 9.84 |
| | With ESE | 0.60 / 0.69 / 0.77 / 0.85 | 0.61 / 0.69 / 0.77 / 0.86 | 0.64 / 0.74 / 0.80 / 0.89 | 0.66 / 0.74 / 0.80 / 0.88 |
| Informer | No ESE | 3.61 / 4.39 / 5.08 / 5.62 | 3.57 / 4.46 / 4.97 / 5.43 | 3.58 / 4.92 / 5.42 / 6.12 | 4.03 / 4.67 / 5.69 / 6.15 |
| | With ESE | 0.43 / 0.48 / 0.54 / 0.59 | 0.43 / 0.49 / 0.52 / 0.57 | 0.45 / 0.53 / 0.57 / 0.64 | 0.48 / 0.52 / 0.58 / 0.64 |
| FiLM | No ESE | 6.77 / 8.17 / 9.41 / 9.76 | 6.53 / 8.51 / 9.32 / 10.32 | 6.69 / 9.15 / 9.36 / 10.44 | 7.26 / 8.80 / 10.84 / 11.94 |
| | With ESE | 0.63 / 0.72 / 0.81 / 0.89 | 0.62 / 0.74 / 0.79 / 0.89 | 0.64 / 0.80 / 0.82 / 0.92 | 0.68 / 0.78 / 0.91 / 1.02 |
| SCINet | No ESE | 8.05 / 9.79 / 11.42 / 12.12 | 7.95 / 9.97 / 11.91 / 13.20 | 8.82 / 10.49 / 12.37 / 13.41 | 8.47 / 10.46 / 12.37 / 13.61 |
| | With ESE | 0.71 / 0.82 / 0.93 / 1.04 | 0.70 / 0.83 / 0.95 / 1.08 | 0.77 / 0.88 / 1.00 / 1.12 | 0.76 / 0.88 / 1.00 / 1.13 |
| DeepAR | No ESE | 4.82 / 6.27 / 7.15 / 7.79 | 5.01 / 6.18 / 7.09 / 7.75 | 5.19 / 6.08 / 7.63 / 8.33 | 5.33 / 6.62 / 7.85 / 8.66 |
| | With ESE | 0.51 / 0.60 / 0.67 / 0.75 | 0.52 / 0.59 / 0.65 / 0.72 | 0.55 / 0.61 / 0.71 / 0.79 | 0.56 / 0.64 / 0.72 / 0.81 |
| KVAE | No ESE | 4.39 / 5.43 / 5.83 / 6.23 | 4.39 / 5.39 / 6.35 / 6.98 | 4.39 / 5.60 / 7.12 / 8.02 | 4.68 / 6.17 / 6.41 / 7.06 |
| | With ESE | 0.48 / 0.55 / 0.58 / 0.64 | 0.48 / 0.54 / 0.61 / 0.67 | 0.50 / 0.58 / 0.68 / 0.77 | 0.52 / 0.61 / 0.63 / 0.70 |
| TPGNN | No ESE | 5.97 / 7.53 / 8.37 / 9.36 | 5.81 / 7.59 / 8.44 / 9.38 | 6.75 / 7.78 / 8.90 / 9.47 | 6.90 / 8.21 / 9.59 / 10.39 |
| | With ESE | 0.58 / 0.68 / 0.74 / 0.83 | 0.57 / 0.68 / 0.74 / 0.83 | 0.64 / 0.71 / 0.79 / 0.86 | 0.66 / 0.74 / 0.83 / 0.92 |
| PatchTST | No ESE | 3.87 / 4.75 / 5.55 / 5.90 | 3.92 / 5.01 / 5.61 / 6.19 | 4.32 / 5.41 / 6.06 / 6.64 | 4.21 / 5.39 / 6.01 / 6.64 |
| | With ESE | 0.45 / 0.51 / 0.57 / 0.62 | 0.45 / 0.52 / 0.56 / 0.62 | 0.49 / 0.56 / 0.61 / 0.67 | 0.49 / 0.56 / 0.60 / 0.67 |

## K The Normalized RMSE and MAE for Exchange Rate

In the case of exchange rate prediction, additional metrics are used:

$$RMSE^* = \frac{\sum_{i=1}^{n} \frac{RMSE_i}{\bar{s_i}}}{n},$$
$$MAE^* = \frac{\sum_{i=1}^{n} \frac{MAE_i}{\bar{s_i}}}{n}, \tag{24}$$

where $RMSE^*$ and $MAE^*$ are the normolized RMSE and MAE, to compensate scale biases. $RMSE_i$ and $MAE_i$ are the RMSE and MAE values for the prediction result of the system $i$, $s_i$. In the scenario of exchange rate prediction, $RMSE_i$ and $MAE_i$ are the RMSE and MAE values for the prediction results of the currency exchange rate $i$. $\bar{s_i}$ is the average target variable for the system $i$.

# L COVID-19 DATA

Table 19: Six columns of COVID-19 daily data

| Variable | Description | Data Type |
|---|---|---|
| Date | Date | Date |
| Population | Total population | Float |
| Active | Active cases | Float |
| Rate | Infection rate | Float |
| New | New confirmed cases | Float |
| Band | Restriction band | Float |

The dataset contains daily COVID-19 data for the 20, 79, 320 regions throughout 2022, in six columns, including date, population, active cases, rates, new cases and risk band level (see Table 19). Note that in this case, the number of "new cases" is the target. The associated attributes about the regions in Victoria and other epidemic related data are collected from main resources, government agencies, e.g. the health department and the Australian bureau of statistics, as detailed in Table 20. These attributes provide comprehensive health and socioeconomic information across 20 and 79 regions throughout 2022, including population health, vaccination rates, and economic conditions.

Table 20: Attributes for regions (only valid when modeling $\mathcal{MS}$ of 20 and 79 regions).

| Attribute | Description | Data Type |
|---|---|---|
| Region | Name of the region | String |
| Type 2 Diabetes Ratio (%) | Percentage of diagnosed Type 2 diabetes cases | Float |
| Multiple Chronic Diseases (%) | Percentage with two or more chronic diseases | Float |
| Chronic Heart Disease (%) | Percentage with chronic heart disease | Float |
| Obesity Rate (%) | Percentage of obese individuals | Float |
| Respiratory Diseases (%) | Percentage with respiratory diseases | Float |
| Asthma Diagnosis (%) | Percentage diagnosed with asthma | Float |
| Daily Smoker (%) | Percentage of daily smokers | Float |
| Vaccine Dose 1 (%) | First dose vaccine coverage | Float |
| Vaccine Dose 2 (%) | Second dose vaccine coverage | Float |
| Vaccine Dose 3 (%) | Third dose vaccine coverage | Float |
| Hospital Employment Ratio | Ratio of hospital-employed individuals | Float |
| Median Weekly Income (AUD) | Median weekly household income in AUD | Float |
| Median Age | Median age of the population | Float |

## L.1 THE RULES OF MERGING REGIONS

To estimate ESE's performance across different levels of granularity, we aggregated the 79 municipalities into 20 larger regions and also partitioned them into 320 smaller regions based on postcodes. The merging rules are that:

- Merged regions must be geographically adjacent;
- The merged attribute is determined as either the sum or the average of the merging regions, depending on the type of attribute. For instance, the sum is appropriate for population, while the average is suitable for the obesity rate.

## M  FULL COMPARISON ON COVID-19 DATA

### M.1  COMPARING PREDICTION PERFORMANCE - COVID-19 DATA

Table 21: Comparison of prediction performance on COVID-19 data with three granularities (20/79/320 regions) with the horizon of 1. RMSE and MAE rows show the means of one method over all regions at each granularity. Row "No ESE" is the method acting alone, while Row "With ESE" is the method augmented with ESE. The experiments are repeated for four input lengths: 10, 20, 50 and 100. We can observe that **(1)** ESE alone outperforms other predictors in many cases; **(2)** ESE consistently boosts the performance of other predictors, with the best results in most columns achieved either by ESE alone or by combining with ESE.

| Models | | Metric | Prediction Horizon = 1 | | | |
| --- | --- | --- | --- | --- | --- | --- |
| | | | *Input Length = 10* | *Input Length = 20* | *Input Length = 50* | *Input Length = 100* |
| ESE | – | RMSE | 53.69 / 57.99 / 4.34 | 52.47 / 54.99 / 4.76 | 62.16 / 54.52 / 4.73 | 84.54 / 55.54 / 4.83 |
| | | MAE | 51.64 / 52.27 / 3.54 | 51.60 / 49.86 / 4.43 | 51.34 / 49.94 / 4.56 | 83.91 / 50.85 / 4.58 |
| VAR | – | RMSE | 45.29 / 63.94 / 6.76 | 54.61 / 78.92 / 8.49 | 77.19 / 84.94 / 8.90 | 95.79 / 90.85 / 8.07 |
| | | MAE | 38.48 / 56.16 / 5.96 | 53.38 / 70.90 / 8.17 | 73.55 / 82.26 / 8.38 | 90.86 / 81.74 / 7.68 |
| ARIMA | No ESE | RMSE | 18.49 / 57.69 / 5.44 | 39.44 / 70.94 / 7.34 | 69.84 / 72.56 / 7.64 | 79.70 / 77.72 / 7.44 |
| | | MAE | 17.47 / 49.22 / 5.02 | 37.64 / 64.31 / 6.24 | 68.05 / 66.87 / 6.54 | 79.59 / 67.41 / 6.24 |
| | With ESE | RMSE | 49.34 / 56.32 / 4.38 | 50.12 / 54.28 / 4.66 | 61.34 / 55.46 / 4.89 | 75.31 / 54.15 / 4.74 |
| | | MAE | 48.41 / 51.21 / 3.46 | 49.16 / 49.12 / 4.39 | 50.34 / 50.45 / 4.47 | 73.64 / 50.65 / 4.41 |
| LSTM | No ESE | RMSE | 16.41 / 46.60 / 4.51 | 37.01 / 59.01 / 5.43 | 57.69 / 60.83 / 5.54 | 78.45 / 61.42 / 6.23 |
| | | MAE | 15.01 / **40.02** / 3.52 | 35.96 / 50.68 / 4.74 | 47.64 / 55.87 / 5.29 | 74.96 / 57.26 / 5.47 |
| | With ESE | RMSE | 48.01 / 56.62 / 4.30 | 48.49 / 56.31 / 4.54 | 60.20 / 55.77 / 4.73 | 72.79 / 55.42 / 4.68 |
| | | MAE | 44.15 / 51.12 / 3.34 | 44.54 / 50.32 / 4.24 | 47.37 / 50.47 / 4.24 | 69.47 / 49.93 / 4.37 |
| Dlinear | No ESE | RMSE | 16.34 / 50.09 / 4.43 | 36.31 / 54.55 / 5.53 | 57.32 / 55.15 / 5.13 | 79.47 / 56.16 / 5.93 |
| | | MAE | 14.96 / 46.04 / 3.61 | 34.78 / 53.78 / 4.53 | 49.63 / 52.95 / 4.83 | 72.41 / 53.95 / 5.10 |
| | With ESE | RMSE | 45.47 / 50.23 / 4.24 | 46.97 / 51.94 / 4.47 | 58.13 / 53.44 / 4.70 | 73.61 / 56.14 / 4.51 |
| | | MAE | 42.74 / 48.19 / 3.16 | 42.17 / 51.63 / 4.14 | 46.82 / 50.42 / 4.36 | 66.94 / 50.12 / 4.34 |
| Nlinear | No ESE | RMSE | **15.01** / **43.28** / **4.13** | **34.78** / 52.69 / 5.34 | 56.74 / 54.22 / 4.97 | 74.96 / 55.62 / 5.47 |
| | | MAE | 14.90 / 43.49 / 3.24 | 33.69 / 51.53 / 4.52 | 47.41 / 51.95 / 4.80 | 70.77 / 52.23 / 4.99 |
| | With ESE | RMSE | 45.23 / 48.32 / 4.16 | 46.03 / 51.64 / 4.39 | 58.14 / 55.01 / 4.71 | 71.64 / 55.14 / 4.47 |
| | | MAE | 40.94 / 44.31 / **3.09** | 41.31 / 50.78 / **4.01** | 45.84 / 50.34 / 4.24 | 69.98 / 51.33 / 4.18 |
| Informer | No ESE | RMSE | 17.03 / 58.96 / 4.86 | 38.99 / 59.85 / 6.07 | 58.31 / 61.23 / 5.71 | 76.84 / 62.01 / 6.05 |
| | | MAE | 14.99 / 47.31 / 3.94 | 35.69 / 53.78 / 5.06 | 46.72 / 58.85 / 5.25 | 71.79 / 60.57 / 5.80 |
| | With ESE | RMSE | 46.25 / 60.32 / 4.31 | 47.34 / 56.33 / 4.48 | 59.42 / 55.17 / 4.80 | 73.43 / 55.48 / 4.60 |
| | | MAE | 43.32 / 51.37 / 3.19 | 43.44 / 50.98 / 4.23 | 48.83 / 49.47 / 4.40 | 71.48 / 49.23 / 4.37 |
| FiLM | No ESE | RMSE | 16.54 / 52.90 / 4.53 | 35.66 / 53.32 / 5.03 | 55.33 / 55.57 / 5.09 | 76.84 / 56.21 / 5.74 |
| | | MAE | 13.85 / 43.21 / 3.34 | 33.58 / 50.66 / 4.94 | 47.45 / 48.60 / 4.57 | 73.85 / 51.34 / 4.96 |
| | With ESE | RMSE | 45.94 / 53.14 / 4.28 | 44.17 / 52.96 / 4.23 | 57.93 / 55.31 / 4.69 | 71.68 / 54.96 / **4.36** |
| | | MAE | 42.45 / 51.67 / 3.09 | 41.96 / 51.14 / 4.14 | **45.83** / 49.66 / 4.32 | 69.15 / 49.01 / **3.96** |
| SCINet | No ESE | RMSE | 15.44 / 50.64 / 4.60 | 35.98 / 57.45 / 5.63 | 58.94 / 59.80 / 6.03 | 79.75 / 55.86 / 5.94 |
| | | MAE | 13.90 / 44.70 / 3.33 | 32.23 / 50.48 / 5.13 | 54.79 / 51.31 / 5.57 | 69.94 / 53.56 / 5.59 |
| | With ESE | RMSE | 44.67 / 53.01 / 4.24 | 45.49 / 52.25 / **4.20** | 58.88 / 54.12 / **4.63** | 71.84 / 54.04 / 4.46 |
| | | MAE | 40.95 / 51.18 / 3.16 | 42.79 / 47.11 / 4.18 | 46.73 / 48.14 / **4.16** | 69.71 / 50.34 / 4.24 |
| DeepAR | No ESE | RMSE | 17.63 / 51.12 / 5.01 | 39.01 / 53.48 / 5.74 | 61.78 / 54.64 / 6.17 | 83.41 / 56.34 / 6.29 |
| | | MAE | 16.41 / 47.32 / 3.94 | 34.68 / 47.54 / 4.93 | 50.74 / 50.31 / 5.64 | 74.65 / 54.32 / 5.04 |
| | With ESE | RMSE | 47.44 / 51.95 / 4.30 | 48.41 / 52.47 / 4.57 | 60.03 / 52.43 / 4.85 | 74.82 / **50.99** / 4.79 |
| | | MAE | 46.41 / 50.33 / 3.38 | 46.94 / 49.16 / 4.29 | 48.34 / 51.02 / 4.38 | 72.91 / 50.96 / 4.46 |
| KVAE | No ESE | RMSE | 16.21 / 48.69 / 4.39 | 35.14 / **48.14** / 4.79 | 54.36 / 52.41 / 4.97 | 70.85 / 53.12 / 5.29 |
| | | MAE | **13.43** / 47.72 / 4.26 | **31.47** / **47.23** / 4.53 | 45.96 / 50.34 / 4.19 | 64.44 / 52.71 / 4.94 |
| | With ESE | RMSE | 44.45 / 52.10 / 4.23 | 44.94 / 51.39 / 4.24 | 58.22 / **52.11** / 4.74 | 71.94 / 52.13 / 4.76 |
| | | MAE | 41.47 / 49.02 / 3.18 | 40.46 / 48.94 / 4.07 | 46.56 / 51.32 / 4.37 | 68.41 / 51.24 / 4.26 |
| TPGNN | No ESE | RMSE | 15.96 / 48.31 / 4.43 | 36.97 / 49.21 / 5.14 | **53.74** / 56.65 / 5.23 | 78.37 / 59.22 / 6.06 |
| | | MAE | 15.12 / 44.12 / 3.84 | 34.12 / 48.36 / 5.03 | 48.71 / 54.79 / 4.57 | 69.86 / 58.13 / 5.41 |
| | With ESE | RMSE | 45.12 / 54.29 / 4.32 | 46.41 / 53.29 / 4.27 | 57.65 / 55.16 / 4.68 | 72.24 / 56.95 / 4.68 |
| | | MAE | 43.56 / 50.41 / 3.34 | 43.46 / 51.32 / 4.26 | 46.07 / 52.96 / 4.36 | 70.64 / 54.99 / 4.24 |
| PatchTST | No ESE | RMSE | 15.67 / 49.46 / 4.43 | 36.23 / 53.87 / 5.23 | 55.43 / 54.49 / 5.07 | **70.01** / 56.44 / 5.11 |
| | | MAE | 14.75 / 43.94 / 3.36 | 31.98 / 47.31 / 4.82 | 49.34 / 53.35 / 4.29 | 63.52 / 52.41 / 4.43 |
| | With ESE | RMSE | 45.82 / 58.19 / 4.23 | 46.09 / 54.66 / 4.33 | 59.12 / 52.58 / 4.65 | 70.74 / 53.71 / 4.51 |
| | | MAE | 42.44 / 51.17 / 3.18 | 41.63 / 49.89 / 4.16 | 48.49 / **47.99** / 4.35 | **65.97** / **47.25** / 4.36 |

Table 22: Comparison of prediction performance on COVID-19 data with three granularities (20/79/320 regions) with a horizon of 2. RMSE and MAE rows show the means of one method over all regions at each granularity. Row "No ESE" is the method acting alone, while Row "With ESE" is the method augmented with ESE. The experiments are repeated for four input lengths: 10, 20, 50 and 100. We can observe that **(1)** ESE alone outperforms other predictors in many cases; **(2)** ESE consistently boosts the performance of other predictors, with the best results in most columns achieved either by ESE alone or by combining with ESE.

| Models | | Metric | Prediction Horizon = 2 | | | |
| --- | --- | --- | --- | --- | --- | --- |
| | | | *Input Length = 10* | *Input Length = 20* | *Input Length = 50* | *Input Length = 100* |
| ESE | – | RMSE | 58.50 / 62.19 / 4.66 | 55.73 / 59.51 / 5.12 | 67.85 / 59.17 / 5.01 | 92.19 / 59.72 / 5.24 |
| | | MAE | 53.96 / 54.90 / 3.66 | 53.41 / 52.00 / 4.61 | 53.54 / 52.05 / 4.80 | 88.29 / 53.45 / 4.78 |
| VAR | – | RMSE | 46.57 / 65.82 / 7.09 | 54.84 / 81.39 / 8.71 | 77.70 / 86.81 / 9.25 | 95.86 / 93.56 / 8.41 |
| | | MAE | 39.12 / 58.72 / 6.20 | 49.67 / 72.96 / 8.33 | 76.63 / 82.57 / 8.60 | 91.64 / 83.38 / 7.83 |
| ARIMA | No ESE | RMSE | 19.20 / 59.31 / 5.60 | 40.38 / 73.22 / 7.57 | 71.86 / 74.51 / 7.88 | 82.07 / 80.34 / 7.72 |
| | | MAE | 18.83 / 53.64 / 5.49 | 39.82 / 69.65 / 6.82 | 64.35 / 72.12 / 7.07 | 86.79 / 73.38 / 6.77 |
| | With ESE | RMSE | 53.74 / 59.54 / 4.61 | 52.55 / 57.58 / 4.85 | 66.92 / 58.84 / 5.16 | 80.58 / 57.66 / 5.07 |
| | | MAE | 51.61 / 53.90 / 3.48 | 50.87 / **49.13** / 4.46 | 55.33 / 50.68 / 4.83 | 75.28 / 54.36 / 4.71 |
| LSTM | No ESE | RMSE | 17.87 / 50.30 / 4.90 | 39.20 / 64.00 / 5.87 | 62.49 / 65.48 / 5.98 | 84.85 / 66.73 / 6.77 |
| | | MAE | 15.52 / **41.36** / 3.64 | 36.76 / 52.00 / 4.83 | 49.20 / 57.05 / 5.46 | 76.35 / 59.12 / 5.62 |
| | With ESE | RMSE | 51.24 / 56.83 / 4.50 | 48.98 / 59.34 / 4.72 | **49.92** / 54.21 / **4.47** | 77.53 / 58.91 / 4.92 |
| | | MAE | 47.15 / 54.57 / 3.51 | 45.34 / 52.11 / 4.43 | 47.81 / 52.14 / 4.37 | 70.29 / 53.19 / 4.42 |
| Dlinear | No ESE | RMSE | 16.65 / 50.90 / 4.54 | 37.07 / 55.80 / 5.68 | 58.62 / 56.70 / 5.24 | 82.09 / 57.09 / 6.12 |
| | | MAE | 15.02 / 46.35 / 3.61 | 34.69 / 53.84 / 4.54 | 49.87 / 53.00 / 4.86 | 72.86 / 53.81 / 5.13 |
| | With ESE | RMSE | 46.21 / 54.20 / 4.28 | 51.02 / 55.62 / **4.48** | 61.34 / 54.35 / 5.17 | 77.71 / 58.76 / 4.80 |
| | | MAE | 43.24 / 51.23 / 3.35 | 44.86 / 52.87 / 4.34 | 47.45 / 54.08 / 4.72 | 75.45 / 53.52 / 4.48 |
| Nlinear | No ESE | RMSE | 16.50 / **47.48** / 4.52 | 37.26 / 57.54 / 5.82 | 62.28 / 59.61 / 5.47 | 81.63 / 60.84 / 5.96 |
| | | MAE | 15.60 / 44.73 / 3.37 | 34.49 / 53.75 / 4.72 | 49.49 / 53.92 / 5.04 | 74.01 / 54.71 / 5.22 |
| | With ESE | RMSE | 45.37 / 51.11 / **4.19** | 50.56 / 53.65 / 4.78 | 62.34 / 57.70 / 4.73 | 76.42 / 59.77 / 4.72 |
| | | MAE | 43.52 / 45.64 / 3.28 | 45.28 / 53.05 / **4.13** | 48.21 / 52.46 / 4.65 | 71.59 / 52.19 / 4.45 |
| Informer | No ESE | RMSE | 17.73 / 60.39 / 5.00 | 40.10 / 61.77 / 6.31 | 60.27 / 63.62 / 5.90 | 78.13 / 64.41 / 6.30 |
| | | MAE | 16.17 / 50.50 / 4.24 | 37.39 / 57.72 / 5.40 | 49.91 / 63.41 / 5.62 | 77.42 / 65.27 / 6.27 |
| | With ESE | RMSE | 47.22 / 63.37 / 4.72 | 48.11 / 61.71 / 4.85 | 64.29 / 58.86 / 4.93 | 76.45 / 55.93 / 4.62 |
| | | MAE | 46.57 / 53.57 / 3.28 | 46.05 / 51.53 / 4.63 | 50.88 / **50.03** / 4.42 | 73.77 / 50.42 / 4.40 |
| FiLM | No ESE | RMSE | 17.87 / 57.78 / 4.95 | 37.75 / 57.75 / 5.45 | 60.11 / 60.73 / 5.52 | 83.54 / 61.18 / 6.21 |
| | | MAE | 14.19 / 43.95 / 3.40 | 34.01 / 51.97 / 5.09 | 48.83 / 50.10 / 4.65 | 76.50 / 53.08 / 5.07 |
| | With ESE | RMSE | 49.00 / 56.48 / 4.50 | 45.53 / 56.94 / 4.62 | 61.44 / 58.91 / 4.82 | 72.65 / 56.50 / **4.45** |
| | | MAE | 45.67 / 53.86 / **3.22** | 45.24 / 52.09 / 4.42 | 47.91 / 52.37 / 4.37 | 69.15 / 49.64 / **4.03** |
| SCINet | No ESE | RMSE | 15.86 / 51.31 / 4.69 | 36.66 / 58.76 / 5.74 | 60.74 / 61.34 / 6.13 | 82.18 / 56.96 / 6.06 |
| | | MAE | 13.87 / 44.53 / 3.33 | 31.96 / 50.10 / 5.10 | 54.91 / 50.48 / 5.57 | 69.80 / 53.10 / 5.50 |
| | With ESE | RMSE | 47.72 / 57.83 / 4.28 | 48.72 / 53.33 / 4.51 | 61.37 / 57.08 / 4.66 | 76.69 / 56.47 / 4.81 |
| | | MAE | 41.42 / 54.89 / 3.45 | 46.17 / 50.55 / 4.40 | **47.36** / 50.44 / **4.34** | 73.07 / 53.05 / 4.39 |
| DeepAR | No ESE | RMSE | 19.68 / 57.99 / 5.62 | 42.85 / 60.17 / 6.41 | 69.39 / 61.15 / 6.94 | 94.20 / 66.58 / 6.32 |
| | | MAE | 17.15 / 54.46 / 4.82 | 40.13 / 58.14 / 5.99 | 62.15 / 56.95 / 6.24 | 91.80 / 63.03 / 6.13 |
| | With ESE | RMSE | 49.47 / 55.85 / 4.65 | 48.83 / **52.82** / 4.91 | 61.54 / 57.59 / 5.00 | 81.55 / **54.68** / 5.08 |
| | | MAE | 47.63 / 54.90 / 3.66 | 47.12 / 51.78 / 4.34 | 51.49 / 51.40 / 4.45 | 76.76 / 52.76 / 4.85 |
| KVAE | No ESE | RMSE | 16.91 / 48.68 / 4.55 | 39.18 / 56.30 / 5.59 | 51.80 / 57.25 / 5.60 | 85.97 / 60.00 / 5.93 |
| | | MAE | 15.95 / 45.77 / 4.22 | 34.55 / 53.34 / 5.11 | 48.42 / 53.14 / 5.32 | 76.52 / 58.67 / 5.82 |
| | With ESE | RMSE | 45.36 / 54.48 / 4.32 | 48.52 / 56.17 / 4.60 | 61.11 / 57.72 / 4.85 | 76.96 / 55.48 / 5.05 |
| | | MAE | 44.62 / 49.37 / 3.25 | 41.34 / 49.19 / 4.23 | 47.84 / 55.87 / 4.62 | 70.77 / 54.17 / 4.59 |
| TPGNN | No ESE | RMSE | 19.71 / 59.24 / 5.43 | 43.37 / 60.64 / 6.31 | 66.11 / 69.84 / 6.40 | 96.12 / 72.72 / 7.50 |
| | | MAE | 17.46 / 50.69 / 4.47 | 37.81 / 55.92 / 5.85 | 56.32 / 62.84 / 5.24 | 81.05 / 67.01 / 6.30 |
| | With ESE | RMSE | 47.09 / 54.99 / 4.60 | 46.68 / 55.39 / 4.55 | 59.55 / 56.25 / 5.07 | 74.76 / 59.39 / 5.07 |
| | | MAE | 44.40 / 53.31 / 3.40 | 43.85 / 54.64 / 4.59 | 50.07 / 57.37 / 4.61 | 73.94 / 59.78 / 4.34 |
| PatchTST | No ESE | RMSE | 16.55 / 52.02 / 4.52 | 36.89 / 57.93 / 5.64 | 56.80 / 58.07 / 5.28 | 72.20 / 61.24 / 5.32 |
| | | MAE | 14.95 / 45.68 / 3.54 | 33.67 / 51.07 / 4.92 | 52.14 / 55.93 / 4.65 | **69.19** / 56.65 / 4.81 |
| | With ESE | RMSE | 46.59 / 61.93 / 4.41 | 46.38 / 57.12 / 4.59 | 60.21 / **52.82** / 4.94 | **71.91** / 56.34 / 4.83 |
| | | MAE | 45.38 / 54.72 / 3.32 | 43.20 / 50.77 / 4.44 | 50.34 / 51.33 / 4.45 | 70.28 / **48.84** / 4.42 |

Table 23: Comparing prediction performance (output horizon is 5) with 12 SOTA methods, in RMSE and MAE, with no ESE and with ESE, with input of 10, 20, 50 and 100 steps, for 20 large regions / 79 regions / 320 sub-regions. We can observe that **(1)** ESE alone outperforms other predictors in many cases; **(2)**ESE consistently boosts the performance of other predictors, with the best results in most columns achieved either by ESE alone or by combining with ESE.

| Models | | Metric | Prediction Horizon = 5 | | | |
| | | | *Input Length = 10* | *Input Length = 20* | *Input Length = 50* | *Input Length = 100* |
|---|---|---|---|---|---|---|
| ESE | – | RMSE | 60.16 / 64.95 / 4.84 | 62.71 / 61.43 / 5.36 | 69.56 / 61.56 / 5.31 | 94.74 / 61.91 / 5.89 |
| | | MAE | 57.09 / 57.70 / 3.90 | 56.03 / 54.75 / 4.90 | 56.79 / 54.99 / 5.02 | 92.14 / 55.90 / 5.04 |
| VAR | – | RMSE | 51.87 / 70.96 / 8.13 | 66.86 / 88.04 / 9.92 | 93.86 / 85.82 / 8.47 | 114.49 / 98.09 / 9.57 |
| | | MAE | 46.98 / 63.25 / 6.65 | 65.47 / 82.92 / 10.41 | 93.13 / 81.51 / 7.85 | 103.24 / 91.26 / 8.70 |
| ARIMA | No ESE | RMSE | 24.38 / 63.88 / 6.03 | 48.50 / 78.15 / 8.11 | 76.90 / 78.30 / 8.40 | 93.27 / 85.53 / 8.23 |
| | | MAE | 22.84 / 57.97 / 5.90 | 42.39 / 75.75 / 7.33 | 69.98 / 75.64 / 7.68 | 88.08 / 79.06 / 7.35 |
| | With ESE | RMSE | 57.41 / 62.42 / 4.83 | 59.51 / 59.68 / 5.63 | 67.52 / 61.44 / 5.42 | 86.86 / 59.99 / 5.46 |
| | | MAE | 54.11 / 60.27 / 3.98 | 57.98 / 57.56 / 5.06 | 55.01 / 53.16 / 5.25 | 82.92 / 54.38 / 5.20 |
| LSTM | No ESE | RMSE | 21.92 / 56.56 / 5.48 | 42.85 / 71.87 / 6.61 | 70.11 / 71.72 / 6.75 | 95.02 / 74.49 / 7.56 |
| | | MAE | 21.47 / 47.61 / 4.21 | 41.09 / 60.44 / 5.66 | 57.12 / 60.51 / 6.31 | 89.23 / 68.40 / 6.56 |
| | With ESE | RMSE | 52.66 / 59.97 / 4.26 | 53.03 / 58.57 / 4.99 | 65.99 / 61.23 / 5.20 | 79.62 / 55.23 / 5.12 |
| | | MAE | 47.76 / 56.15 / 3.57 | 48.02 / 49.10 / 4.58 | 50.94 / 52.25 / 4.58 | 74.63 / 51.82 / 4.71 |
| Dlinear | No ESE | RMSE | 21.30 / 54.25 / 4.77 | 38.59 / 58.90 / 6.00 | 62.04 / 59.02 / 5.56 | 85.64 / 60.76 / 6.39 |
| | | MAE | 19.13 / 49.10 / 3.83 | 36.42 / 57.32 / 4.83 | 52.72 / 57.35 / 5.17 | 77.40 / 57.57 / 5.44 |
| | With ESE | RMSE | 46.57 / 57.56 / 4.59 | 48.10 / 53.39 / 4.57 | 59.82 / 55.13 / 4.82 | **75.42** / 57.53 / 4.64 |
| | | MAE | 43.42 / 53.05 / 3.59 | 42.79 / **48.28** / 4.18 | 47.38 / 51.21 / 4.40 | 72.50 / 50.94 / 4.30 |
| Nlinear | No ESE | RMSE | 20.57 / **52.70** / 4.72 | **38.35** / 60.04 / 6.10 | 64.50 / 60.14 / 5.65 | 85.26 / 63.34 / 6.24 |
| | | MAE | 20.14 / 49.05 / 3.66 | 36.82 / 58.16 / 5.10 | 53.38 / 58.25 / 5.44 | 80.04 / 59.30 / 5.62 |
| | With ESE | RMSE | 48.95 / 56.44 / 4.52 | 49.87 / 56.19 / 4.78 | 62.88 / 59.55 / 5.10 | 77.96 / 59.57 / 4.85 |
| | | MAE | 44.09 / 51.50 / 3.61 | 44.23 / 52.29 / 4.40 | 49.05 / 51.01 / 4.56 | 74.83 / 54.98 / 4.47 |
| Informer | No ESE | RMSE | 21.50 / 61.88 / 5.11 | 42.77 / 62.70 / 6.45 | 65.07 / 66.77 / 6.08 | 85.99 / 74.67 / 6.53 |
| | | MAE | 18.54 / 54.70 / 4.55 | 38.74 / 58.62 / 5.76 | 58.99 / 64.11 / 5.80 | 80.70 / 70.23 / 6.48 |
| | With ESE | RMSE | 48.42 / 63.24 / 4.52 | 49.62 / 58.99 / 4.68 | 62.07 / 57.98 / 5.02 | 76.76 / 57.94 / 4.81 |
| | | MAE | 42.49 / 55.12 / 3.69 | 50.25 / 49.24 / 4.33 | 53.50 / 53.18 / 4.63 | 72.87 / 56.92 / 4.10 |
| FiLM | No ESE | RMSE | 20.93 / 59.13 / 5.11 | 40.10 / 59.53 / 5.34 | 64.73 / 63.47 / 5.40 | 84.06 / 65.31 / 6.06 |
| | | MAE | 15.56 / 48.48 / 3.92 | 37.15 / 57.59 / 4.61 | 54.60 / 55.46 / 4.80 | 79.11 / 56.94 / 5.65 |
| | With ESE | RMSE | 51.53 / 56.42 / 4.78 | 49.45 / 59.36 / 4.89 | 64.78 / 60.65 / 5.26 | 79.93 / 59.30 / 4.88 |
| | | MAE | 42.02 / 51.25 / 3.46 | 41.40 / 50.63 / **4.12** | 45.37 / 49.18 / 4.26 | 68.45 / **48.58** / 3.91 |
| SCINet | No ESE | RMSE | 19.49 / 53.00 / 4.83 | 37.20 / 60.60 / 5.92 | 62.00 / 60.40 / 6.35 | 83.74 / 68.90 / 6.27 |
| | | MAE | 18.17 / 48.82 / 3.62 | **34.36** / 55.28 / 5.58 | 59.97 / 55.20 / 6.10 | 76.46 / 58.84 / 6.14 |
| | With ESE | RMSE | 44.68 / 62.43 / 4.25 | 45.65 / **52.27** / **4.38** | 58.61 / **54.01** / **4.61** | 76.88 / 53.80 / **4.45** |
| | | MAE | 42.48 / 53.13 / 3.50 | 44.36 / 51.17 / 4.14 | 48.72 / 49.91 / 4.32 | **72.42** / 52.25 / 4.41 |
| DeepAR | No ESE | RMSE | 24.58 / 59.55 / 5.82 | 41.94 / 62.23 / 6.66 | 71.72 / 62.27 / 7.17 | 97.04 / 65.39 / 7.34 |
| | | MAE | 23.25 / 55.89 / 4.63 | 37.32 / 56.11 / 5.79 | 59.70 / 55.99 / 6.63 | 87.61 / 64.01 / 5.95 |
| | With ESE | RMSE | 49.95 / 61.98 / **4.17** | 56.22 / 60.73 / 5.29 | 69.70 / 60.76 / 5.23 | 86.63 / **51.41** / 4.59 |
| | | MAE | 47.26 / 56.17 / 3.55 | 53.23 / 51.97 / 4.55 | 47.74 / 51.30 / 4.83 | 83.60 / 50.11 / **3.67** |
| KVAE | No ESE | RMSE | 21.98 / 52.46 / 5.18 | 41.90 / 60.88 / 6.02 | **56.62** / 61.03 / 6.22 | 93.53 / 64.77 / 6.59 |
| | | MAE | 20.17 / 50.87 / 4.96 | 37.02 / 59.10 / 5.66 | 54.27 / 59.16 / 6.05 | 84.68 / 62.54 / 5.21 |
| | With ESE | RMSE | 52.07 / 60.96 / 4.97 | 52.62 / 59.97 / 4.94 | 67.87 / 58.98 / 5.33 | 84.38 / 59.21 / 4.87 |
| | | MAE | 44.21 / 52.23 / **3.37** | 43.04 / 51.84 / 4.33 | 49.51 / 52.70 / 4.63 | 72.53 / 54.24 / 4.52 |
| TPGNN | No ESE | RMSE | 21.41 / 62.40 / 5.69 | 45.44 / 63.23 / 6.63 | 69.04 / 63.32 / 6.73 | 89.21 / 76.22 / 7.77 |
| | | MAE | 19.95 / 56.68 / 4.90 | 41.60 / 61.99 / 6.43 | 62.44 / 61.91 / 5.83 | 79.55 / 74.43 / 6.93 |
| | With ESE | RMSE | 50.67 / 62.96 / 4.83 | 52.04 / 59.53 / 4.79 | 64.50 / 60.52 / 5.05 | 80.70 / 58.73 / 5.23 |
| | | MAE | 48.43 / 53.09 / 3.73 | 48.24 / 49.39 / 4.26 | 51.16 / 50.10 / 4.86 | 78.51 / 55.16 / 4.73 |
| PatchTST | No ESE | RMSE | **19.05** / 55.33 / 4.94 | 40.10 / 62.56 / 5.73 | 58.25 / 60.75 / 5.71 | 84.02 / 66.97 / 6.41 |
| | | MAE | **15.13** / **44.89** / 4.60 | 35.49 / 50.94 / 5.10 | 52.09 / 57.68 / 4.97 | 79.02 / 57.05 / 5.93 |
| | With ESE | RMSE | 50.60 / 65.59 / 4.75 | 52.01 / 62.50 / 4.82 | 65.69 / 60.34 / 5.20 | 79.39 / 60.65 / 4.98 |
| | | MAE | 46.77 / 56.56 / 3.58 | 47.74 / 55.90 / 4.59 | 54.87 / 54.55 / 4.98 | 73.47 / 52.27 / 4.27 |

Table 24: Comparison of prediction performance on COVID-19 data with three granularities (20/79/320 regions) with a horizon of 5. RMSE and MAE rows show the means of one method over all regions at each granularity. Row "No ESE" is the method acting alone, while Row "With ESE" is the method augmented with ESE. The experiments are repeated for four input lengths: 10, 20, 50 and 100. We can observe that **(1)** ESE alone outperforms other predictors in many cases; **(2)** ESE consistently boosts the performance of other predictors, with the best results in most columns achieved either by ESE alone or by combining with ESE.

| Models | | Metric | Prediction Horizon = 10 | | | |
| --- | --- | --- | --- | --- | --- | --- |
| | | | *Input Length = 10* | *Input Length = 20* | *Input Length = 50* | *Input Length = 100* |
| ESE | – | RMSE | 61.89 / 66.57 / 5.99 | 65.98 / 73.11 / 5.49 | 68.22 / 67.66 / 5.42 | 97.34 / 69.27 / 5.55 |
| | | MAE | 59.83 / 60.47 / 5.08 | 59.86 / 57.53 / 5.11 | 62.32 / 63.77 / 5.26 | 95.02 / 64.62 / 5.30 |
| VAR | – | RMSE | 55.18 / 78.13 / 8.14 | 67.77 / 98.17 / 10.58 | 98.19 / 105.90 / 10.79 | 117.09 / 112.45 / 10.45 |
| | | MAE | 49.42 / 67.56 / 7.34 | 64.52 / 87.39 / 10.09 | 90.99 / 101.47 / 9.51 | 112.44 / 100.95 / 9.79 |
| ARIMA | No ESE | RMSE | 25.08 / 71.01 / 6.68 | 56.28 / 87.47 / 9.05 | 86.38 / 89.27 / 9.45 | 98.31 / 87.48 / 9.20 |
| | | MAE | 24.72 / 63.19 / 6.44 | 55.49 / 82.67 / 8.03 | 87.09 / 85.86 / 8.37 | 102.05 / 82.51 / 7.98 |
| | With ESE | RMSE | 61.57 / 64.49 / 5.21 | 63.92 / 69.95 / 5.44 | 64.62 / 63.32 / 6.05 | 91.88 / 61.79 / 5.37 |
| | | MAE | 56.90 / 60.83 / 4.93 | 57.92 / 60.82 / 5.15 | 59.57 / 59.68 / 5.23 | 89.79 / 60.15 / 5.18 |
| LSTM | No ESE | RMSE | 23.07 / 59.44 / 5.75 | 54.39 / 75.39 / 6.93 | 73.72 / 77.31 / 7.04 | 99.83 / 75.18 / 7.94 |
| | | MAE | 20.95 / 51.13 / 4.48 | 53.15 / 64.56 / 6.03 | 60.61 / 71.12 / 6.71 | 95.77 / 64.53 / 7.00 |
| | With ESE | RMSE | 56.40 / 63.37 / 4.60 | 56.89 / 67.97 / 5.29 | 63.76 / 66.16 / 5.52 | 87.54 / 65.82 / 5.44 |
| | | MAE | 51.04 / 52.21 / 4.37 | 51.92 / 58.83 / 4.89 | 55.10 / 59.23 / 4.90 | 83.31 / 58.74 / 5.08 |
| Dlinear | No ESE | RMSE | 22.04 / 58.39 / 5.19 | 50.66 / 63.52 / 6.46 | 67.14 / 64.59 / 6.02 | 92.78 / 63.74 / 6.90 |
| | | MAE | 19.18 / 51.46 / 4.81 | 47.50 / 59.52 / 5.31 | 53.33 / 59.99 / 5.69 | 88.74 / 58.37 / 5.78 |
| | With ESE | RMSE | 48.06 / 60.83 / 4.47 | 56.87 / 65.78 / 4.72 | 62.70 / 57.45 / 4.99 | 80.94 / 60.88 / 4.77 |
| | | MAE | 44.94 / 54.27 / 4.18 | 49.41 / 55.30 / 4.36 | 49.56 / 54.06 / 4.59 | 78.21 / 53.68 / 4.47 |
| Nlinear | No ESE | RMSE | 21.87 / **52.43** / 4.96 | **49.79** / 63.18 / 6.45 | 68.31 / 64.99 / 5.97 | 89.84 / 63.35 / 6.61 |
| | | MAE | 19.06 / 50.22 / 3.76 | 47.39 / 59.69 / 5.25 | 54.71 / 60.34 / 5.54 | 81.92 / 59.71 / 5.80 |
| | With ESE | RMSE | 49.60 / 57.33 / 4.49 | 50.42 / 62.04 / 4.77 | 64.62 / 61.12/ 5.16 | 80.85 / 61.16 / 4.88 |
| | | MAE | 42.36 / 52.52 / 4.09 | **43.05** / 53.79 / 4.24 | 47.97 / 53.50 / 4.40 | 75.94 / 54.23 / 4.34 |
| Informer | No ESE | RMSE | 22.92 / 72.13 / 5.96 | 56.44 / 73.54 / 7.45 | 71.49 / 74.89 / 7.01 | 93.52 / 73.26 / 7.41 |
| | | MAE | 20.71 / 59.39 / 4.94 | 53.23 / 67.45 / 6.36 | 62.63 / 73.56 / 6.56 | 90.27 / 67.35 / 7.26 |
| | With ESE | RMSE | 48.91 / 60.33 / 4.53 | 58.05 / 63.38 / 4.72 | 64.36 / 59.42 / 5.08 | **80.67** / 59.92 / 4.86 |
| | | MAE | 46.65 / 56.45 / 4.30 | 56.98 / 55.95 / 4.55 | 53.44 / 53.93 / 4.73 | 80.24 / 53.83 / 4.70 |
| FiLM | No ESE | RMSE | 23.37 / 67.77 / 5.81 | 52.91 / 68.19 / 6.44 | 70.69 / 70.95 / 6.54 | 98.40 / 68.25 / 7.37 |
| | | MAE | **17.00** / **47.94** / 3.70 | 46.15 / 56.51 / 5.49 | 52.49 / 54.18 / 5.09 | 82.39 / 56.29 / 5.49 |
| | With ESE | RMSE | 53.83 / 61.01 / 5.00 | 55.65 / 62.54 / 5.10 | **59.12** / 65.58 / 5.51 | 86.48 / 65.61 / 5.06 |
| | | MAE | 42.11 / 52.43 / 3.94 | 49.64 / 51.79 / **4.10** | 46.08 / **50.41** / 4.32 | **71.67** / **49.26** / **3.90** |
| SCINet | No ESE | RMSE | **20.48** / 60.85 / 5.51 | 53.32 / 69.22 / 6.80 | 71.08 / 72.02 / 7.23 | 96.06 / 69.10 / 7.17 |
| | | MAE | 19.36 / 56.65 / 4.20 | 50.05 / 63.89 / 6.49 | 69.14 / 64.99 / 7.02 | 88.44 / 63.85 / 7.09 |
| | With ESE | RMSE | 43.69 / 62.07 / **4.42** | 54.73 / 63.95 / 4.88 | 59.43 / **54.41** / **4.56** | 83.61 / **54.32** / **4.66** |
| | | MAE | 42.12 / 59.91 / 3.82 | 52.05 / **51.63** / 4.29 | 48.95 / 50.42 / **4.28** | 74.86 / 52.71 / 4.37 |
| DeepAR | No ESE | RMSE | 24.88 / 61.90 / 6.12 | 55.00 / 66.93 / 6.67 | 75.73 / 73.09 / 6.93 | 99.18 / 67.14 / 7.37 |
| | | MAE | 19.40 / 52.81 / 5.60 | 48.34 / 62.01 / 5.97 | 60.37 / 66.31 / 5.49 | 84.99 / 62.18 / 6.48 |
| | With ESE | RMSE | 52.55 / 53.69 / 4.54 | 59.99 / **60.65** / 4.96 | 63.25 / 60.91 / 5.41 | 87.15 / 58.46 / 5.31 |
| | | MAE | 49.54 / 59.83 / 3.92 | 54.36 / 54.08 / 4.39 | 52.39 / 56.94 / 4.54 | 91.87 / 56.68 / 4.66 |
| KVAE | No ESE | RMSE | 23.27 / 60.72 / 5.76 | 58.95 / 70.89 / 6.95 | 65.42 / 71.62 / 7.03 | 98.45 / 70.75 / 7.60 |
| | | MAE | 22.32 / 58.54 / 5.23 | 53.05 / 67.86 / 6.49 | 62.51 / 67.81 / 7.19 | 97.92 / 68.02 / 7.46 |
| | With ESE | RMSE | 50.00 / 61.58 / 4.72 | 60.46 / 59.77 / **4.71** | 62.97 / 60.81 / 5.41 | 85.18 / 60.70 / 5.44 |
| | | MAE | 42.43 / 59.51 / 4.00 | 51.24 / 52.44 / 4.14 | 49.07 / 55.56 / 4.55 | 78.06 / 55.50 / 4.44 |
| TPGNN | No ESE | RMSE | 23.25 / 64.05 / 5.84 | 56.21 / 64.98 / 6.80 | 71.15 / 75.09 / 6.90 | 96.70 / 65.08 / 8.02 |
| | | MAE | 21.61 / 57.41 / 4.97 | 51.98 / 62.90 / 6.54 | 63.23 / 71.22 / 5.93 | 90.85 / 62.77 / 6.99 |
| | With ESE | RMSE | 53.04 / 64.88 / 5.04 | 59.39 / 61.65 / 4.97 | 65.28 / 64.04 / 5.53 | 87.54 / 64.02 / 5.49 |
| | | MAE | 49.84 / 59.39 / 3.79 | 53.98 / 52.68 / 4.87 | 52.98 / 59.87 / 5.01 | 84.26 / 59.44 / 4.84 |
| PatchTST | No ESE | RMSE | 21.10 / 67.19 / 5.80 | 50.31 / 70.11 / 6.94 | 77.25 / 76.29 / 7.06 | 92.65 / 78.95 / 6.80 |
| | | MAE | 20.53 / 61.18 / 4.64 | 43.36 / 64.12 / 6.28 | 68.05 / 72.54 / 5.90 | 85.67 / 73.31 / 6.10 |
| | With ESE | RMSE | 53.04 / 59.60 / 6.86 | 53.37 / 64.31 / 5.01 | 69.18 / 61.00 / 5.50 | 83.42 / 60.67 / 5.04 |
| | | MAE | 48.56 / 51.74 / **3.62** | 48.91 / 57.96 / 4.70 | 55.34 / 53.04 / 4.91 | 76.01 / 55.41 / 4.83 |

## M.2  PLOTS OF COMPARING PREDICTION PERFORMANCE

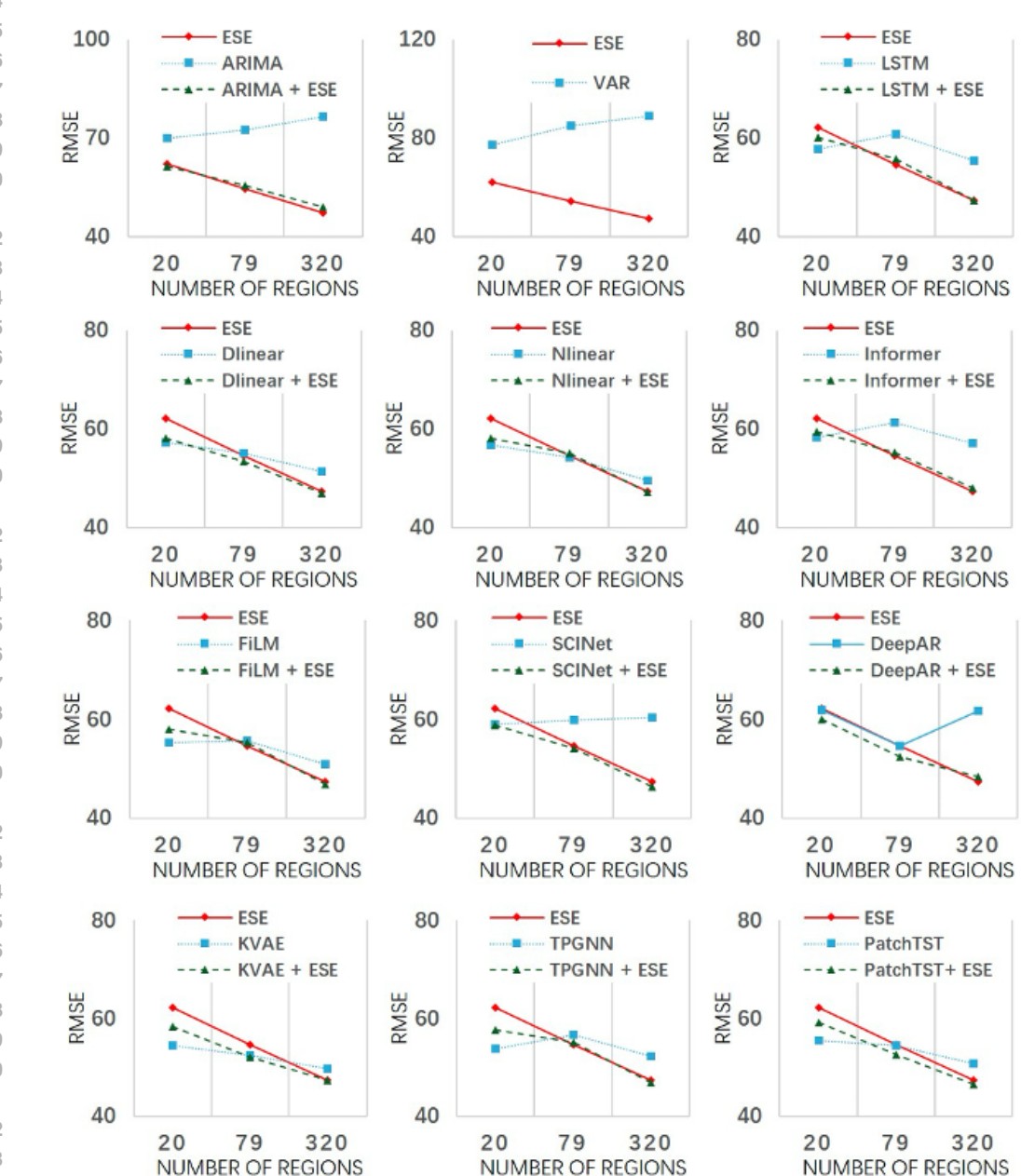

Figure 6: Comparing with 12 SOTA methods in RMSE on different numbers (20, 79 and 320) of systems for COVIDE-19 data (input size = 50, horizon = 1 ).

Figure 6 illustrates ESE's advantage over an increasing number of systems. ESE can perform better with more regions. In comparison, other methods either deteriorate or do not improve as much. Another point to highlight is that when combined with ESE, these 12 methods also show a similar trend as that of ESE alone.

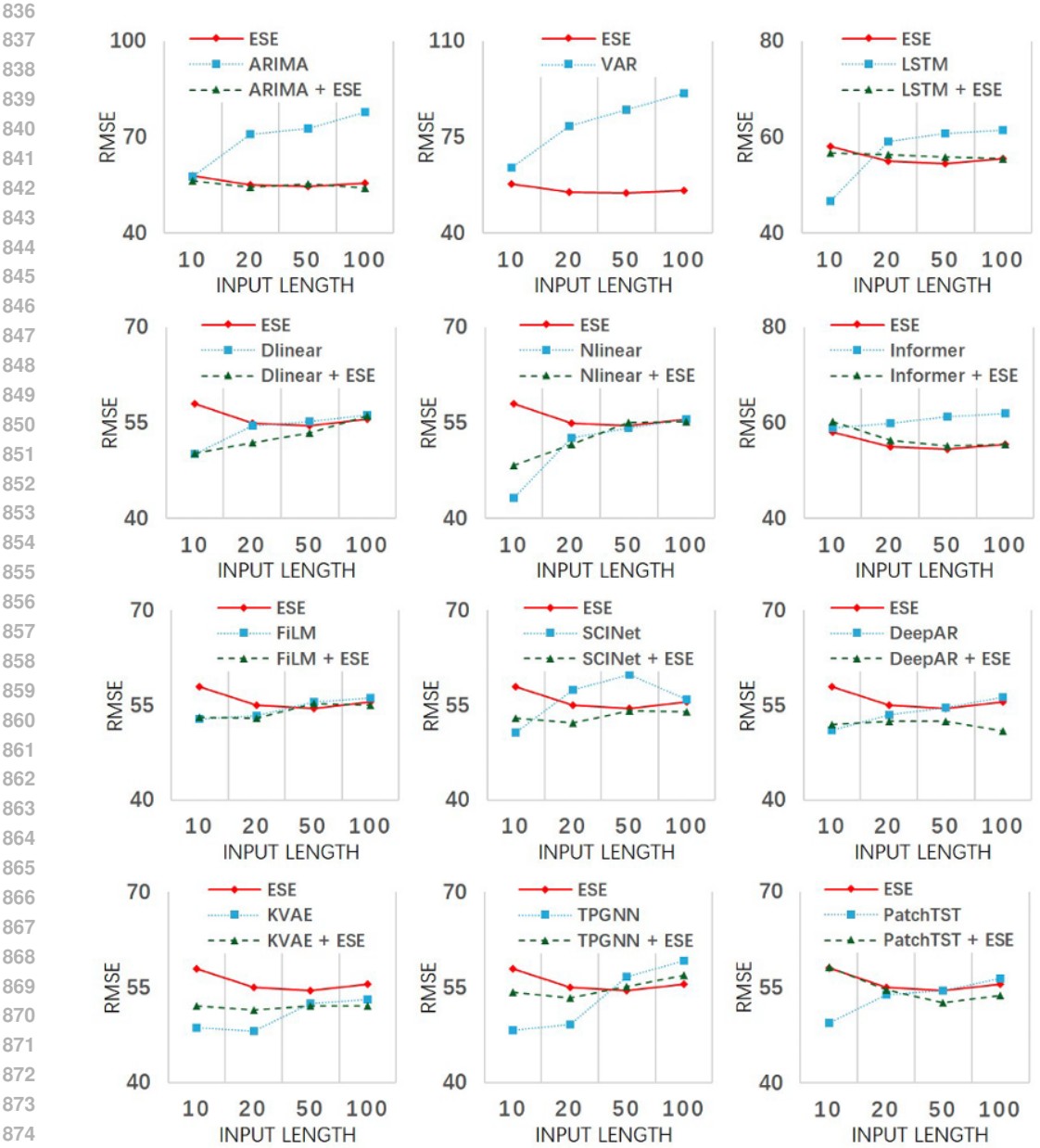

Figure 7: Comparing with 12 SOTA methods in RMSE on different input sizes, 10, 20, 50 and 100 (79 regions, horizon = 1 )

Figure 7 illustrates how ESE handles different input sizes, ranging from 10 to 100. It clearly shows that ESE can handle large inputs as most of the lowest RMSE with input over 50 are either from ESE or SOTA methods combined with ESE.

## M.3 COMPARING COMPUTATIONAL COSTS - COVID-19 DATA

Table 25: Comparison of computational cost (in minutes) on COVID-19 data with three granularities (20/79/320 regions) with a horizon of 1. Row "No ESE" is the method acting alone, while Row "With ESE" is the method augmented with ESE. The experiments are repeated for four input lengths: 10, 20, 50 and 100. We can observe that ESE significantly reduces costs for all methods when integrated. When applied to FiLM and SCINet on the 320-region dataset, ESE enables speedups exceeding 70×. Additionally, ESE demonstrates strong scalability with increasing region count, while most other methods either degrade in performance or show limited improvement.

| Models | | Costs (mins) for Horizon = 1 | | | |
|---|---|---|---|---|---|
| | | *Input Length = 10* | *Input Length = 20* | *Input Length = 50* | *Input Length = 100* |
| ESE | – | 1.19 / 1.49 / **1.71** | 1.23 / 1.43 / **1.82** | 1.22 / 1.45 / **1.97** | 1.31 / 2.10 / **2.28** |
| ARIMA | No ESE | **0.18** / **0.70** / 2.81 | **0.22** / **0.89** / 3.59 | **0.27** / **1.09** / 4.11 | **0.33** / **1.31** / 5.07 |
| | With ESE | 1.20 / 1.50 / 1.72 | 1.24 / 1.44 / 1.83 | 1.23 / 1.46 / 1.99 | 1.33 / 2.12 / 2.30 |
| LSTM | No ESE | 5.06 / 20.68 / 86.88 | 6.14 / 27.76 / 109.29 | 7.77 / 33.81 / 131.74 | 9.23 / 39.96 / 149.26 |
| | With ESE | 1.46 / 1.77 / 1.98 | 1.57 / 1.74 / 2.13 | 1.61 / 1.84 / 2.41 | 1.80 / 2.59 / 2.72 |
| Dlinear | No ESE | 6.20 / 23.96 / 96.44 | 7.07 / 31.61 / 132.31 | 10.28 / 38.58 / 159.46 | 10.40 / 40.98 / 163.57 |
| | With ESE | 1.48 / 1.79 / 2.03 | 1.62 / 1.81 / 2.18 | 1.69 / 1.90 / 2.49 | 1.86 / 2.68 / 2.87 |
| Nlinear | No ESE | 6.04 / 24.91 / 99.01 | 7.40 / 31.37 / 135.11 | 9.57 / 37.83 / 155.00 | 11.56 / 45.67 / 160.09 |
| | With ESE | 1.50 / 1.81 / 2.01 | 1.60 / 1.80 / 2.23 | 1.72 / 1.93 / 2.41 | 1.87 / 2.70 / 2.86 |
| Informer | No ESE | 3.52 / 13.24 / 57.48 | 4.56 / 17.16 / 74.00 | 5.38 / 20.58 / 93.92 | 5.95 / 26.97 / 100.08 |
| | With ESE | 1.36 / 1.67 / 1.88 | 1.43 / 1.67 / 2.03 | 1.48 / 1.73 / 2.25 | 1.66 / 2.40 / 2.62 |
| FiLM | No ESE | 6.63 / 26.22 / 108.97 | 7.62 / 34.91 / 143.09 | 9.66 / 40.59 / 171.36 | 11.70 / 48.55 / 181.06 |
| | With ESE | 1.50 / 1.82 / 2.02 | 1.65 / 1.87 / 2.24 | 1.73 / 1.96 / 2.44 | 1.86 / 2.74 / 2.84 |
| SCINet | No ESE | 7.98 / 30.62 / 127.32 | 10.37 / 40.89 / 151.48 | 12.24 / 49.96 / 189.39 | 14.18 / 62.27 / 206.06 |
| | With ESE | 1.57 / 1.88 / 2.12 | 1.70 / 1.92 / 2.31 | 1.78 / 2.11 / 2.60 | 2.04 / 2.82 / 2.94 |
| DeepAR | No ESE | 5.11 / 19.57 / 76.72 | 6.16 / 23.56 / 94.99 | 7.34 / 31.55 / 131.00 | 9.22 / 37.25 / 130.67 |
| | With ESE | 1.43 / 1.74 / 1.95 | 1.52 / 1.72 / 2.15 | 1.63 / 1.87 / 2.39 | 1.77 / 2.52 / 2.73 |
| KVAE | No ESE | 4.37 / 17.03 / 67.34 | 5.21 / 22.17 / 90.23 | 7.19 / 28.31 / 96.23 | 6.80 / 32.41 / 109.62 |
| | With ESE | 1.41 / 1.71 / 1.92 | 1.49 / 1.69 / 2.07 | 1.53 / 1.78 / 2.30 | 1.67 / 2.50 / 2.63 |
| TPGNN | No ESE | 5.87 / 23.56 / 97.40 | 7.90 / 27.60 / 119.72 | 9.85 / 35.00 / 152.22 | 10.58 / 42.92 / 158.84 |
| | With ESE | 1.49 / 1.80 / 2.00 | 1.62 / 1.82 / 2.19 | 1.72 / 1.91 / 2.48 | 1.81 / 2.65 / 2.86 |
| PatchTST | No ESE | 3.71 / 15.55 / 61.08 | 4.60 / 18.68 / 82.90 | 5.85 / 23.96 / 94.47 | 6.92 / 27.70 / 109.89 |
| | With ESE | 1.38 / 1.69 / 1.90 | 1.48 / 1.69 / 2.08 | 1.53 / 1.78 / 2.29 | 1.79 / 2.46 / 2.66 |

Table 26: Comparison of computational cost (in minutes) on COVID-19 data with three granularities (20/79/320 regions) with a horizon of 2. Row "No ESE" is the method acting alone, while Row "With ESE" is the method augmented with ESE. The experiments are repeated for four input lengths: 10, 20, 50 and 100. We can observe that ESE significantly reduces costs for all methods when integrated. When applied to FiLM and SCINet on the 320-region dataset, ESE enables speedups exceeding 70×. Additionally, ESE demonstrates strong scalability with increasing region count, while most other methods either degrade in performance or show limited improvement.

| Models | | Costs (mins) for Horizon = 2 | | | |
| --- | --- | --- | --- | --- | --- |
| | | $Input\ Length = 10$ | $Input\ Length = 20$ | $Input\ Length = 50$ | $Input\ Length = 100$ |
| ESE | | 1.20 / 1.50 / **1.72** | 1.23 / 1.44 / **1.81** | 1.22 / 1.46 / **1.99** | 1.32 / 2.11 / **2.30** |
| ARIMA | No ESE | **0.18** / **0.70** / 2.71 | **0.21** / **0.94** / 3.78 | **0.29** / **1.02** / 4.79 | **0.35** / **1.30** / 5.32 |
| | With ESE | 1.21 / 1.51 / 1.73 | 1.24 / 1.45 / 1.84 | 1.24 / 1.47 / 1.99 | 1.34 / 2.13 / 2.30 |
| LSTM | No ESE | 5.06 / 20.24 / 88.66 | 7.04 / 25.26 / 104.36 | 7.83 / 33.44 / 132.97 | 10.29 / 41.51 / 153.76 |
| | With ESE | 1.47 / 1.76 / 1.96 | 1.55 / 1.75 / 2.15 | 1.61 / 1.89 / 2.41 | 1.77 / 2.59 / 2.75 |
| Dlinear | No ESE | 6.02 / 24.55 / 95.88 | 7.02 / 32.21 / 127.01 | 10.14 / 36.63 / 140.45 | 10.77 / 48.41 / 165.12 |
| | With ESE | 1.49 / 1.80 / 2.02 | 1.62 / 1.81 / 2.22 | 1.69 / 1.91 / 2.51 | 1.88 / 2.71 / 2.88 |
| Nlinear | No ESE | 6.18 / 23.48 / 98.89 | 7.86 / 32.94 / 124.18 | 10.26 / 37.00 / 150.42 | 11.08 / 43.64 / 186.56 |
| | With ESE | 1.48 / 1.80 / 2.03 | 1.65 / 1.83 / 2.19 | 1.67 / 1.93 / 2.50 | 1.91 / 2.67 / 2.86 |
| Informer | No ESE | 3.53 / 14.52 / 58.83 | 4.67 / 16.95 / 71.12 | 5.40 / 22.88 / 88.41 | 6.54 / 24.66 / 93.47 |
| | With ESE | 1.36 / 1.68 / 1.89 | 1.43 / 1.65 / 2.05 | 1.50 / 1.75 / 2.27 | 1.62 / 2.41 / 2.62 |
| FiLM | No ESE | 6.49 / 25.28 / 102.23 | 7.62 / 32.35 / 138.70 | 10.62 / 42.84 / 167.44 | 11.61 / 43.21 / 198.15 |
| | With ESE | 1.52 / 1.81 / 2.06 | 1.66 / 1.85 / 2.25 | 1.76 / 2.01 / 2.46 | 1.95 / 2.65 / 2.84 |
| SCINet | No ESE | 8.09 / 30.57 / 130.87 | 9.43 / 41.49 / 158.16 | 12.55 / 45.86 / 211.56 | 15.92 / 55.12 / 236.50 |
| | With ESE | 1.59 / 1.91 / 2.10 | 1.70 / 1.91 / 2.34 | 1.83 / 2.03 / 2.64 | 2.09 / 2.89 / 3.05 |
| DeepAR | No ESE | 4.75 / 19.20 / 79.48 | 6.58 / 24.60 / 95.80 | 7.70 / 29.92 / 126.18 | 9.03 / 31.82 / 151.75 |
| | With ESE | 1.45 / 1.73 / 1.96 | 1.54 / 1.76 / 2.14 | 1.63 / 1.81 / 2.34 | 1.73 / 2.55 / 2.71 |
| KVAE | No ESE | 4.02 / 16.90 / 65.41 | 5.57 / 22.04 / 84.81 | 6.08 / 27.94 / 113.13 | 7.36 / 32.06 / 133.49 |
| | With ESE | 1.40 / 1.72 / 1.92 | 1.47 / 1.72 / 2.09 | 1.57 / 1.80 / 2.31 | 1.70 / 2.50 / 2.71 |
| TPGNN | No ESE | 6.09 / 22.68 / 99.67 | 7.21 / 28.38 / 118.50 | 9.36 / 37.78 / 152.74 | 10.76 / 44.22 / 155.24 |
| | With ESE | 1.48 / 1.80 / 2.02 | 1.59 / 1.80 / 2.24 | 1.67 / 1.94 / 2.43 | 1.86 / 2.63 / 2.87 |
| PatchTST | No ESE | 3.77 / 14.77 / 60.33 | 4.83 / 18.83 / 81.41 | 6.31 / 24.19 / 101.68 | 7.29 / 28.69 / 103.71 |
| | With ESE | 1.38 / 1.69 / 1.91 | 1.47 / 1.67 / 2.08 | 1.54 / 1.76 / 2.31 | 1.69 / 2.47 / 2.61 |

Table 27: Comparison of computational cost (in minutes) on COVID-19 data with three granularities (20/79/320 regions) with a horizon of 5. Row "No ESE" is the method acting alone, while Row "With ESE" is the method augmented with ESE. The experiments are repeated for four input lengths: 10, 20, 50 and 100. We can observe that ESE significantly reduces costs for all methods when integrated. When applied to FiLM and SCINet on the 320-region dataset, ESE enables speedups exceeding 70×. Additionally, ESE demonstrates strong scalability with increasing region count, while most other methods either degrade in performance or show limited improvement.

| Models | | Costs (mins) for Horizon = 5 | | | |
| | | $Input\ Length = 10$ | $Input\ Length = 20$ | $Input\ Length = 50$ | $Input\ Length = 100$ |
|---|---|---|---|---|---|
| ESE | | 1.25 / 1.53 / **1.91** | 1.31 / 1.61 / **1.90** | 1.23 / 1.54 / **2.05** | 1.40 / 2.41 / **2.38** |
| ARIMA | No ESE | **0.18** / **0.83** / 3.22 | **0.24** / **1.02** / 3.64 | **0.30** / **1.19** / 4.55 | **0.34** / **1.46** / 5.81 |
| | With ESE | 1.24 / 1.68 / 1.91 | 1.36 / 1.64 / 1.92 | 1.36 / 1.49 / 2.10 | 1.51 / 2.14 / 2.33 |
| LSTM | No ESE | 5.68 / 21.82 / 91.98 | 7.45 / 27.28 / 112.66 | 8.79 / 32.04 / 138.17 | 11.39 / 41.16 / 158.84 |
| | With ESE | 1.67 / 2.04 / 2.23 | 1.68 / 2.08 / 2.49 | 1.92 / 2.07 / 2.76 | 2.01 / 2.70 / 3.24 |
| Dlinear | No ESE | 6.05 / 23.42 / 113.54 | 7.03 / 35.35 / 124.01 | 10.35 / 41.71 / 160.66 | 11.36 / 40.65 / 189.59 |
| | With ESE | 1.57 / 1.88 / 2.22 | 1.78 / 1.85 / 2.34 | 1.83 / 2.11 / 2.68 | 1.97 / 2.64 / 3.05 |
| Nlinear | No ESE | 6.59 / 27.11 / 103.52 | 7.77 / 32.62 / 146.44 | 10.50 / 39.99 / 168.24 | 12.11 / 43.00 / 177.14 |
| | With ESE | 1.56 / 1.88 / 2.23 | 1.73 / 1.84 / 2.48 | 1.78 / 2.07 / 2.72 | 1.95 / 2.75 / 3.01 |
| Informer | No ESE | 4.08 / 14.90 / 64.72 | 4.35 / 17.16 / 76.05 | 5.96 / 24.74 / 99.45 | 6.86 / 26.33 / 108.50 |
| | With ESE | 1.53 / 1.90 / 2.08 | 1.59 / 1.66 / 2.22 | 1.56 / 1.80 / 2.55 | 1.73 / 2.74 / 2.64 |
| FiLM | No ESE | 6.65 / 27.88 / 115.14 | 7.84 / 33.46 / 140.27 | 10.58 / 45.99 / 193.35 | 11.37 / 49.80 / 177.95 |
| | With ESE | 1.55 / 2.07 / 2.24 | 1.87 / 2.05 / 2.45 | 1.93 / 2.00 / 2.52 | 1.94 / 3.01 / 3.13 |
| SCINet | No ESE | 8.52 / 31.75 / 144.09 | 11.46 / 40.04 / 183.36 | 12.63 / 50.63 / 222.10 | 14.04 / 62.88 / 262.88 |
| | With ESE | 1.68 / 2.11 / 2.37 | 1.74 / 1.96 / 2.66 | 1.89 / 2.39 / 2.94 | 1.97 / 3.12 / 3.44 |
| DeepAR | No ESE | 5.07 / 18.75 / 82.37 | 6.46 / 25.93 / 102.85 | 7.53 / 29.79 / 123.68 | 9.52 / 39.91 / 143.83 |
| | With ESE | 1.51 / 1.90 / 1.96 | 1.65 / 1.76 / 2.29 | 1.82 / 2.07 / 2.69 | 1.97 / 2.83 / 3.06 |
| KVAE | No ESE | 4.64 / 19.12 / 66.30 | 5.56 / 20.49 / 91.55 | 7.02 / 24.84 / 120.19 | 9.01 / 31.98 / 153.34 |
| | With ESE | 1.41 / 1.84 / 2.20 | 1.71 / 1.78 / 2.15 | 1.54 / 1.79 / 2.63 | 1.76 / 2.77 / 2.86 |
| TPGNN | No ESE | 6.13 / 26.30 / 111.59 | 8.85 / 34.59 / 123.10 | 9.10 / 37.87 / 167.70 | 10.93 / 47.32 / 167.42 |
| | With ESE | 1.67 / 2.06 / 2.08 | 1.82 / 1.83 / 2.22 | 1.78 / 2.09 / 2.52 | 2.02 / 2.65 / 2.96 |
| PatchTST | No ESE | 4.30 / 16.03 / 67.92 | 5.10 / 21.31 / 91.14 | 6.46 / 25.72 / 117.19 | 7.36 / 27.97 / 132.35 |
| | With ESE | 1.57 / 1.70 / 2.08 | 1.55 / 1.84 / 2.23 | 1.53 / 1.78 / 2.43 | 1.81 / 2.61 / 2.70 |

Table 28: Comparison of computational cost (in minutes) on COVID-19 data with three granularities (20/79/320 regions) with a horizon of 10. Row "No ESE" is the method acting alone, while Row "With ESE" is the method augmented with ESE. The experiments are repeated for four input lengths: 10, 20, 50 and 100. We can observe that ESE significantly reduces costs for all methods when integrated. When applied to FiLM and SCINet on the 320-region dataset, ESE enables speedups exceeding 70×. Additionally, ESE demonstrates strong scalability with increasing region count, while most other methods either degrade in performance or show limited improvement.

| Models | | Costs (mins) for Horizon = 10 | | | |
| --- | --- | --- | --- | --- | --- |
| | | *Input Length* = 10 | *Input Length* = 20 | *Input Length* = 50 | *Input Length* = 100 |
| ESE | | 1.20 / 1.77 / **2.00** | 1.30 / 1.65 / **2.03** | 1.22 / 1.66 / **1.98** | 1.36 / 2.11 / **2.66** |
| ARIMA | No ESE | **0.20** / **0.77** / 3.11 | **0.23** / **0.97** / 3.86 | **0.33** / **1.29** / 4.64 | **0.31** / **1.47** / 5.27 |
| | With ESE | 1.32 / 1.78 / 1.89 | 1.37 / 1.54 / 2.12 | 1.40 / 1.51 / 2.03 | 1.51 / 2.15 / 2.57 |
| LSTM | No ESE | 5.39 / 23.63 / 93.81 | 6.64 / 27.15 / 130.60 | 8.34 / 35.48 / 156.14 | 11.54 / 41.35 / 183.91 |
| | With ESE | 1.60 / 2.03 / 2.17 | 1.65 / 2.04 / 2.47 | 1.77 / 2.09 / 2.65 | 1.84 / 3.00 / 2.98 |
| Dlinear | No ESE | 7.17 / 25.10 / 102.81 | 8.49 / 33.99 / 149.76 | 9.38 / 38.68 / 180.71 | 12.24 / 43.94 / 190.21 |
| | With ESE | 1.68 / 2.10 / 2.35 | 1.70 / 2.05 / 2.33 | 1.94 / 2.23 / 2.51 | 1.96 / 2.75 / 3.27 |
| Nlinear | No ESE | 6.49 / 26.95 / 104.37 | 7.99 / 37.47 / 136.83 | 9.96 / 44.62 / 156.76 | 11.51 / 48.25 / 171.26 |
| | With ESE | 1.58 / 1.99 / 2.07 | 1.85 / 1.82 / 2.45 | 1.84 / 2.12 / 2.45 | 1.96 / 2.72 / 3.04 |
| Informer | No ESE | 3.71 / 15.70 / 62.75 | 4.41 / 20.75 / 86.88 | 5.85 / 23.84 / 89.99 | 6.69 / 27.92 / 97.35 |
| | With ESE | 1.52 / 1.80 / 2.00 | 1.61 / 1.98 / 2.46 | 1.72 / 1.93 / 2.38 | 1.66 / 2.89 / 2.76 |
| FiLM | No ESE | 7.00 / 26.69 / 115.83 | 9.40 / 36.31 / 138.15 | 10.87 / 44.67 / 172.86 | 12.54 / 57.13 / 218.43 |
| | With ESE | 1.76 / 2.19 / 2.14 | 1.97 / 1.99 / 2.55 | 1.81 / 2.00 / 2.63 | 2.00 / 2.99 / 3.26 |
| SCINet | No ESE | 9.11 / 31.92 / 133.02 | 11.90 / 38.60 / 161.36 | 13.38 / 49.13 / 214.69 | 14.99 / 64.88 / 239.29 |
| | With ESE | 1.87 / 1.96 / 2.26 | 1.82 / 2.15 / 2.35 | 1.91 / 2.30 / 2.64 | 2.15 / 3.02 / 3.58 |
| DeepAR | No ESE | 5.75 / 22.47 / 90.77 | 6.66 / 27.13 / 103.54 | 8.19 / 33.24 / 131.73 | 9.22 / 37.33 / 172.25 |
| | With ESE | 1.68 / 1.88 / 2.32 | 1.59 / 2.01 / 2.25 | 1.65 / 2.24 / 2.71 | 1.93 / 2.89 / 2.93 |
| KVAE | No ESE | 4.33 / 18.79 / 73.88 | 4.90 / 25.86 / 97.79 | 7.79 / 30.02 / 123.96 | 7.83 / 28.27 / 144.77 |
| | With ESE | 1.39 / 1.71 / 2.25 | 1.52 / 2.00 / 2.42 | 1.63 / 1.76 / 2.54 | 1.71 / 2.96 / 2.74 |
| TPGNN | No ESE | 5.97 / 24.32 / 104.31 | 7.44 / 34.46 / 155.33 | 10.28 / 45.28 / 144.83 | 12.46 / 50.40 / 172.18 |
| | With ESE | 1.72 / 2.03 / 2.25 | 1.62 / 2.09 / 2.58 | 1.70 / 2.10 / 2.74 | 2.11 / 2.89 / 3.38 |
| PatchTST | No ESE | 4.39 / 15.63 / 66.70 | 4.99 / 20.69 / 89.16 | 7.09 / 24.81 / 109.28 | 6.74 / 33.07 / 118.99 |
| | With ESE | 1.51 / 1.77 / 1.94 | 1.70 / 1.78 / 2.12 | 1.63 / 2.00 / 2.56 | 1.66 / 2.56 / 3.14 |

## N    COMPARING NAIVE OR ADAPTIVE WEIGHTING STRATEGY

Since the equilibrium state ($\mathcal{ES}$) obtained from estimation can be regarded as a form of weighting, one could use naive or adaptive weighting schemes to achieve the same goal as ESE. To address this point, we apply a naive approach on currency exchange rate prediction. Specifically, we compute a set of weights that are then applied to generate forecasts for horizons $h = 1, 2, 5$, and $10$. These weights are calculated using Eq. 25:

$$w_{i,t} = \frac{s_{i,t}}{\sum_{i=1}^{n} s_{i,t}} \tag{25}$$

where $w_{i,t}$ denotes the weight of system $i$ at time $t$ in an n-system $\mathcal{MS}$, and $s_{i,t}$ represents the target value of system $i$ at time $t$. The denominator is the total value across all systems, which in our case is always 1 (because of the aforementioned normalization). The corresponding predictions for all systems are then performed as defined in Eq. 7 of Section 3.

To ensure data diversity, five distinct time points were selected from February, March, April, May, and June 2024. The prediction results of both the naïve approach and our ESE method are reported in terms of RMSE, as shown in the following table.

Table 29: Comparing ESE with Naive Weighting Strategy

| Sample | Method | Date | h=1 | h=2 | h=5 | h=10 |
|--------|--------|------|-----|-----|-----|------|
| 1 | naive | 14-Jun-24 | 6.44 | 6.94 | 12.47 | 32.84 |
|   | ESE | 14-Jun-24 | 6.01 | 7.60 | 10.69 | 15.31 |
| 2 | naive | 17-May-24 | 6.22 | 8.52 | 16.77 | 36.54 |
|   | ESE | 17-May-24 | 5.94 | 7.67 | 10.65 | 15.27 |
| 3 | naive | 22-Apr-24 | 5.52 | 9.32 | 20.97 | 24.74 |
|   | ESE | 22-Apr-24 | 6.14 | 7.61 | 10.58 | 15.42 |
| 4 | naive | 22-Mar-24 | 6.34 | 7.56 | 14.74 | 35.02 |
|   | ESE | 22-Mar-24 | 6.00 | 7.68 | 10.58 | 15.44 |
| 5 | naive | 23-Feb-24 | 5.94 | 8.64 | 29.57 | 39.41 |
|   | ESE | 23-Feb-24 | 6.12 | 7.54 | 10.59 | 15.37 |

The naive approach performs reasonably well on short horizons (e.g., $h = 1$), achieving performance comparable to ESE in general. However, its accuracy deteriorates as the forecasting horizon increases, and it lags significantly behind ESE in all cases at $h = 10$.

This demonstrates the significant advantage of equilibrium estimation over weighting approaches, particularly in long-horizon scenarios. This is because weighting approaches do not account for the impact of attribute changes on future states of these systems. In contrast, our equilibrium estimation method aims to capture these dynamics and adjust the predictions accordingly, leading to higher accuracies. From this, it can be seen that naive weighting will not reproduce the behavior of ESE.

## O    EVALUATING TRUE EQUILIBRIUM STATES

As discussed in Section 3.1 of the main paper, reaching true equilibrium is not required in ESE, which only estimates the equilibrium state. Actually, reaching the true equilibrium state does not provide additional benefits for the prediction. Below shows an example. The reduced forex scenario comprises only the G7 countries. It includes four non-USD currencies: CAD, GBP, EUR, and JPY. This simplified configuration can clearly illustrate two key points: (1) ESE is not limited to modeling fully closed systems; it can effectively accommodate incomplete systems. (2) The ESE-estimated equilibrium can be as effective as the true equilibrium for downstream prediction tasks.

The comparison results are presented in the table below. The true equilibrium states in the table are derived using the differential equation method, which is commonly employed to compute true equilibrium states Bai et al. (2019); Ding et al. (2023); Nash (2024). It is important to note that the differential equation approach does not scale well with a large number of systems, making it difficult to obtain true equilibrium states for scenarios with more than four currencies.

The equilibrium weights computed by both the differential equation method and ESE are expressed as 4-element vectors (all expressed as proportions), representing the four currencies. The predicted values based on these weights are listed in the fourth column, while the rightmost are the observed values with normalized errors (RMSE*). The input consists of 50 time steps with a horizon of $h = 1$. The three time points estimated are Apr. 22, 2024, May. 17, 2024, and Jun. 14, 2024.

Table 30: Comparing true equilibrium state with estimated equilibrium state

| Sample | Equilibrium | State (system weight) | Predicted Values | Observed values + RMSE*s |
|---|---|---|---|---|
| 1 | True by differential eq | (0.0087, 0.0059, 0.0049, 0.9805) | 1.37, 0.94, 0.78, 155.60 | 1.37, 0.93, 0.79, 157.70 |
| | Estimated by ESE | (0.0087, 0.0060, 0.0050, 0.9803) | 1.36, 0.93, 0.78, 153.22 | 0.92 vs. 1.10 |
| 2 | True by differential eq | (0.0084, 0.0057, 0.0048, 0.9811) | 1.36, 0.93, 0.78, 159.38 | 1.36, 0.92, 0.79, 156.37 |
| | Estimated by ESE | (0.0084, 0.0057, 0.0049, 0.9810) | 1.36, 0.93, 0.79, 159.16 | 0.86 vs. 0.61 |
| 3 | True by differential eq | (0.0084, 0.0058, 0.0050, 0.9808) | 1.35, 0.93, 0.81, 157.94 | 1.37, 0.94, 0.81, 154.81 |
| | Estimated by ESE | (0.0084, 0.0058, 0.0050, 0.9808) | 1.38, 0.94, 0.82, 159.78 | 1.28 vs. 1.17 |

As shown in the above comparison, ESE's estimations are closely aligned with true equilibrium states computed via differential equations. Consequently, both approaches lead to comparable predictive accuracy. Their normalized RMSE values (RMSE*), adjusted to account for scale differences in JPY, are reported below the observed values at each time point. Their mean RMSE*s across the three samples are 1.02 vs. 0.96 respectively, indicating no clear predictive advantage in true equilibrium over ESE.

# P MULTIVARIATE-ONLY PREDICTION TOOLS COMBINED WITH ESE

Since certain multi-variate prediction tools cannot perform prediction for a single target variable, such as VAR, Crossformer Zhang & Yan (2023), they do not directly apply to Eq. 8, thus being unsuitable for direct comparison. However, there is a workaround: summing the predicted values from VAR and subsequently applying ESE to compute values for each individual system. Note that this approach involves a second transformation of the numerical values, hence it was not included in the paper. The results are shown below. It includes four currencies: CAD, GBP, EUR, and JPY.

Table 31: Comparing Multivariate-only prediction tools

| | | Predicted Values | $RMSE^*s$ (x 100) |
|---|---|---|---|
| Observed values | | 1.37, 0.93, 0.79, 157.70 | - |
| ESE | | 1.38, 0.93, 0.82, 157.17 | 1.10 |
| VAR | No ESE | 1.38, 0.91, 0.77, 155.67 | 1.18 |
| | With ESE | 1.37, 0.92, 0.81, 155.64 | 1.12 |
| Crossformer <univariate, with attributes> | No ESE | 1.37, 0.92, 0.81, 159.14 | 1.12 |
| | With ESE | 1.35, 0.91, 0.80, 153.65 | 1.80 |

# Q ESE TOLERATES MISSING ATTRIBUTES

ESE downweights attributes that are noisy or absent. When an attribute is absent for a system, it is simply ignored. The study below shows that the impact on performance is small and depends on the relevance of the attribute. Population (Table 15) is weakly related to exchange rates, hence its absence produces minimal change. M2 and Export are more relevant, but even removing them does not significantly collapse performance, further demonstrating ESE's robustness.

Table 32: Comparing Multivariate-only prediction tools

| | $RMSE^*$ (ESE) | $RMSE^*$ (DLinear+ESE) |
|---|---|---|
| No missing attributes | 1.0972 | 0.9495 |
| Removing Germany Population | 1.0976 | 0.9584 |
| Removing EUR M2 | 1.0121 | 0.9742 |
| Removing Germany Export | 1.0120 | 0.9787 |

## R    USE OF LLMs

Large Language Models (LLMs) were used only to check typos and grammar mistakes. No part of the research design, modeling, experiments, analysis, presentation or writing relied on LLMs. All ideas, methods, and conclusions were conceived, implemented, and validated entirely by the authors.

