# OpenReview forum: "Once-for-All: Scalable Simultaneous Forecasting via Equilibrium State Estimation"
_ICLR.cc/2026/Conference — Submitted to ICLR 2026_

### Official Review · Reviewer_kPHS · 2025-10-23

**Soundness:** 3
**Presentation:** 1
**Contribution:** 3
**Rating:** 2
**Confidence:** 2

**Summary:**

The paper introduces a new method called Equilibrium State Estimation (ESE), designed for the simultaneous prediction of interacting systems. Each system is characterized by a set of attributes whose evolution over time serves as covariates in the prediction process. The objective is to predict a quantity of interest across multiple systems by leveraging the dynamics of these covariates.

The prediction task is decomposed into two components:
1. Total quantity prediction – estimating the overall magnitude of the quantity of interest across all systems.
2. Equilibrium ratio estimation – predicting the relative contribution (ratio) of each system to the total quantity.

The central idea is that while the total quantity may vary dynamically, the relative ratios between systems tend to exhibit greater stability and may evolve toward an equilibrium state. By separating the modeling of the total quantity and the equilibrium ratios, the method aims to achieve more stable and interpretable predictions.

A key aspect of the approach is the use of cointegration to calibrate the equilibrium estimation. The algorithm begins with an initial prediction and iteratively updates or biases the estimates through a correction term L until cointegration is achieved in the predicted signals.

Another noteworthy feature of the method is its agnosticism to the specific model used for predicting the total quantity. While the paper describe as default a simple linear predictor for that component, the framework is flexible and can incorporate any time series prediction model, as shown in the experiments.

The authors evaluate their method on both synthetic datasets and real-world applications, including currency exchange rates and COVID-19 spread modeling, and analyze the computational complexity of the approach.

**Strengths:**

The primary strength of the paper is that the proposed method works effectively in practice. The results demonstrate good predictive performance, achieving both reduced computational cost and improved accuracy, which supports the soundness of the approach.

Another strong point is the use of real-world data in addition to synthetic experiments. This empirical grounding gives credibility to the method and shows that it can handle complex, practical scenarios.

The idea of incorporating equilibrium information into the prediction process is particularly interesting. In contexts such as currency exchange rates, equilibrium relationships are known to exist, and explicitly modeling them is conceptually appealing. While one might question whether such equilibria are stable only in the short term, the decision to include this additional information is well-motivated and insightful.

Overall, the framework provides clear benefits: by leveraging equilibrium structure, the method delivers higher accuracy and shorter computation times. The underlying idea is both sound and potentially impactful, with evident relevance for applications where interacting systems evolve toward some form of equilibrium.

**Weaknesses:**

1. Presentation and Clarity

The most significant weakness of the paper is its presentation. The text is filled with typos and inconsistencies (e.g., line 179: “twp” instead of “two”). The notation used in Algorithm 1 is also problematic: line 166 defines the all-ones vector using np.ones, mixing a pseudo-algorithm written in mathematical notation with Python-specific syntax. This not only assumes familiarity with Python but also breaks the consistency of presentation. Such initialization should be expressed purely mathematically.

There is also significant notation confusion. The paper alternates between A, α, and lowercase a, sometimes apparently referring to the same quantity. In particular, α seems to appear without ever being introduced, leading to unnecessary ambiguity. Variables and concepts often appear before being properly defined. Key terms such as cointegration or even the predictor are introduced abruptly, with unclear formulations. For instance, the description of the predictor as using “log-maximum likelihood” and residuals is either somewhat vague; it would be clearer to describe it as a linear predictor, or define clearly each term in equation 7 and the training procedure.

In general, the mathematical writing and textual explanations are weak. Some sentences are simply ill-formed (e.g., line 212: “which would not work without them”), which undermines the paper’s readability. The result is that a first reading generates many unanswered questions and leaves the reader struggling to follow the logic.

2. Structure and Conceptual Consistency

Beyond language, the paper suffers from structural and conceptual inconsistencies. The first three pages, in particular, introduce several ideas, equilibrium state, utility function, deltas, that are poorly connected to the actual method presented later. Section 2 on equilibrium state feels like filler: it could be merged into the introduction without loss of substance. Many of the definitions provided are trivial observations, such as the fact that the ratios sum to 1, implying that their changes sum to 0. While potentially insightful if leveraged, these points are not exploited in the actual method, which ultimately reduces to linear regression.

As a result, there is a disconnect between the theoretical narrative and the implemented algorithm. The “equilibrium state” terminology seems overemphasized relative to what is mathematically demonstrated.

3. Incorrect or Confusing Derivations

The proof in Appendix E seems wrong or underexplained, as it looks like a circular argument. Either the reasoning is incorrect, or it is poorly explained. In both cases, clarification is needed.

Additionally, the proof title includes the independence sign, which is poor formatting.

4. Missing or Undefined Key Concepts

Crucial concepts such as cointegration are introduced without definition or explanation. The equilibrium estimation process itself is described through an algorithm that updates L and converges when cointegration is achieved, but the mechanism and intuition behind this convergence remain obscure. There is no explanation of why such an iterative correction should work or what motivates its structure.

Overall, the paper spends excessive time on an overextended introduction while neglecting to introduce and justify the most central concepts. The result is a confusing read that obscures what is otherwise a potentially interesting and well-motivated method.

5. Mathematical and Notational Ambiguities

Mathematical definitions lack precision. For instance, the state s is never explicitly placed within a space (e.g., ℝⁿ or a manifold). Similarly, the upper and lower bounds U and L are introduced,should they not simply be denoted as min and max.
The notation with sometimes 4 indices ($\alpha'_{1:n,k,t}$ line 197) is heavy and hard to parse. I would recommend using vector notation and denote time dependence in a functional way, i.e. $\alpha_k(t)\in\mathbb{R}^n,\ k\in ${$1,..,m$} etc.


Overall, manuscript is currently not fit for publication. Significant effort to improve the presentation and properly introduce the necessary concepts is needed. This poor presentation makes the evaluation of the method itself difficult. The current evaluation will mostly reflect the quality of the presentation rather than the work in itslef. I am willing to revise my rating if the writing improves significantly.

**Questions:**

It is unclear whether this work is the first to leverage this decomposition between total quantity and contribution between the different ratios. Are there other methods that use similar assumption? In such case benchmarking should also compare with this other methods that also incorporate equilibrium-like assumptions.

Why was this specific formulation chosen for the state estimation? There are some elements of linear regression, as well as the iterative refinement of the correction L. As you mention later, any method can be used as predictor, why wouldn't it be the same for the equilibrium state estimation? For instance an LSTM with a softmax on the last layer?

What is the use of introducing utility functions and overall section 2, that could not be done by the introduction?

In computational complexity, you assert linear complexity, which is achieved because you limit the while loop to E iterations. Potentially, the E necessary to achieve good results could scale poorly with dimension, especially in difficult instances. Do you have guaranties, results, empirical evidence, that this is not the case?

While the idea of an equilibrium for currency exchange is straightforward, what is your justification for COVID data? In general, do you have a framework for when your method would work, beyond the idea of interracting systems. You mention that definitions 1,2,3 must be satisfied, however as I mentioned it hard to connect these to the ESE framework mathematically, beyond general motivation.

---

> ### Author Response · Authors · 2025-11-26
> **Response to Weaknesses 1-5 and Question 1**
>
> Dear Reviewer kPHS, Thank you for your patience. It took us some time to revise the paper in order to address all of your comments. All changes in the revised version are highlighted in RED.
>
> We sincerely appreciate your recognition of the strengths of our work, particularly its effectiveness, novelty, and impact. We would like to clarify a few key points that may have caused some confusion regarding our problem formulation and methodology.
>
> >**w.1 Presentation and Clarity**
>
> We are grateful for your meticulous review. "twp" has been corrected to "two". To clarify the notation, $\mathcal{A}$ denotes the set of all attributes $\alpha$. We have added a note after *Definition 3* to make this explicit (in red). Furthermore, we have enhanced the explanations in the sections on cointegration and the predictor for greater clarity. As suggested, the description "*log maximum likelihood*" has been revised to the more standard: "the parameter $\theta_{t+h}$ is estimated by maximizing the log-likelihood function of a linear autoregressive model for $\mathcal{MS}$."
>
> >**w.2 Structure and Conceptual Consistency**
>
> We have restructured the manuscript to strengthen its logical coherence. The text now emphasizes that the "*equilibrium state*" in our study is purely a statistical concept, describing a proportional relationship toward which multiple systems converge under given attribute values. It is not the complex game-theoretic equilibrium that requires an explicit solution.
>
> Constraints such as the sum of proportions being $1$ and the sum of their changes being $0$ are not trivial observations. In our framework, both the estimated state ($\mathcal{ES}$) and the system state ($\mathcal{ST}$) inherently adhere to these principles. This reinforces the idea that while the overall trend of the system may change, the internal proportional relationships relative to the equilibrium remain stable.
>
> >**w.3 Incorrect or Confusing Derivations**
>
> We agree that the proof in Appendix E can be expanded, so the impression of a circular argument can be avoided.  Hence, we have enhanced the descriptions quite extensively in that section, emphasizing the separability between overall trends and internal proportional distributions.  For example, the opening explanation: "*This section proves a point used in the predictor (Section 3.2): the change in the overall multiple system value $\Delta MS_{t+h}$ is mathematically separable from the dynamics of the proportional states (Equilibrium State) $\gamma_{i,t}$. This allows us to model the overall trend and the internal distribution independently.*"
>
> >**w.4 Missing or Undefined Key Concepts (e.g. cointegration)**
>
> Cointegration is a well known statistical term that we assumed would be familiar to most readers.  In the original text, Line 214, we wrote "*The cointegration test in Algorithm 1 is a common statistical method for assessing the existence of a long-run equilibrium Abadir (2004); Enders & Siklos (2001).*"  That is where cointegration first appears. To make the point clear, that part has been expanded in the revised version.
>
> "The equilibrium estimation ... the mechanism and intuition ... remain obscure." Figs 1 and 2 both serve to intuitively illustrate the concept, which is somewhat unorthodox: the target pattern converges to resemble the attribute pattern in Systems A/B/C, thereby enabling simultaneous predictions.
>
> Thank you for the comment on the paper structure. We have strengthened the introduction of key concepts such as cointegration and equilibrium estimation algorithms.
>
> >**w.5 Mathematical and Notational Ambiguities**
>
> Re: the upper and lower bounds (denoted as $U$ and $L$), we wish to clarify that they represent theoretical extremes, not the observed range in the dataset. For instance, the theoretical minimum for a region's population is 0, whereas its practical minimum in our data is over 100. We have added text to the manuscript to prevent this potential misunderstanding.
>
> For the mathematical notation, we are grateful for the valuable suggestions. We have updated the main text to improve the notations.
>
> >**Q.1 It is unclear whether this work is the first to leverage this decomposition between total quantity and contribution between the different ratios. Are there other methods that use similar assumption? In such case benchmarking should also compare with this other methods that also incorporate equilibrium-like assumptions.**
>
> Our work is the first to leverage this decomposition between total quantity and contribution between the different ratios, to the best of our knowledge. Benchmarking against methods with a similar design would be highly valuable, but despite our efforts, we have not identified such counterparts in the existing literature.  We have compared with naive weighting strategy, which is similar to the idea of decomposition. Nevertheless, there is no published work on naive weighting as well.

---

> ### Author Response · Authors · 2025-11-26
> **Response to Questions 2-5**
>
> >**Q.2 Why was this specific formulation chosen for the state estimation? ...... For instance an LSTM with a softmax on the last layer?**
>
> ESE is designed to handle large scale multiple prediction tasks by avoiding predicting individual systems one by one.  That is the key rationale behind the ensemble-like formulation.  With this formulation, the relative contribution of each system towards this ensemble can be estimated.  Based on that, we can obtain the corresponding prediction without invoking the predictor on every system.
>
> By the question "*why wouldn't it be the same for the equilibrium state estimation*", do you mean the equilibrium state can be estimated by a predictor?  If so, the number of target variables in the state would be the same as the number of systems, which may be many. Predictors are in general not ideal in this case, even just from a computational cost perspective. As shown in our result tables, they are much slower than ESE.  Using them for equilibrium state estimation would be costly, defeating the purpose of introducing ESE (for high efficiency).  Another way of evaluating the equilibrium state is by differential equation or first-order conditions method, which is discussed in ***Appendix O***.
>
> >**Q.3 What is the use of introducing utility functions and overall section 2, that could not be done by the introduction?**
>
> Utility function or payoff function is essential in conventional equilibrium studies, e.g. Nash Equilibrium.  It is mentioned for readers from equilibrium related fields.  As ESE can be viewed from that perspective:  "*From a utility standpoint, the estimation process is to maximize confidence in the cointegration between ST (t−p):t and E S t, until the confidence reaches a level above 0.95.*" (Line 239)
>
> Thank you for the suggestion, we have moved some part from Section 2 into Introduction.  The rest of Section 2 is the key mathematical formulation of multiple prediction tasks, hence not suited for Section 1.
>
> >**Q.4 In computational complexity, you assert linear complexity, ......  Do you have guaranties, results, empirical evidence, that this is not the case?**
>
> We would like to clarify the confusion about the maximum epoch parameter E. It is actually always $\infty$ in our study, not limiting the convergence process.  We have clarified that in the paper (Line 233).  The linear complexity is due to the ESE algorithm itself.  As stated in Line 428, "*shown in Line 3 of Algorithm 1, the number of iterations is proportional to the time step p, indicating its linear time complexity.*"  Hence, the linear time complexity is guaranteed, regardless of dimensions and difficulties.
>
> >**Q.5 While the idea of an equilibrium for currency exchange is straightforward, ...... however as I mentioned it hard to connect these to the ESE framework mathematically, beyond general motivation.**
>
> The justification for COVID-19 data is based on the paper’s core premise: ESE models interacting systems whose dynamics influence one another. In the COVID-19 setting, changes in one region’s case numbers affect its neighboring regions through mobility, shared health resources, and policy spillovers. These interdependencies naturally form an ensemble whose proportions evolve collectively.  Our ESE does not assume the epidemic is at equilibrium; rather, it estimates the latent equilibrium state implied by current attributes (e.g., population, regional bands), then forecasts deviations from it. This is precisely why ESE performs strongly on COVID-19 data: the method captures cross-regional influence patterns that are difficult to capture by conventional single-system or multivariate models.
>
> As discussed in Limitations (Page 10), "*ESE is based on equilibrium, so not suitable for scenarios that do not satisfy Definitions 1, 2, and 3, e.g. multiple unrelated systems.*"  For systems that are not related, they may have very different sets of attributes, for example, a school and a restaurant, which are difficult to model together with ESE.
>
> Definitions 1–3 are a key part of ESE’s mathematical framework. They formalize the mathematical expression of ESE, particularly to ensure that the attributes used in ESE calculations are consistent and comparable across systems. They do not impose any structural restrictions.  Simply put, these definitions mean that, to facilitate ESE prediction, all systems should use the same set of attributes. In addition, the input data should NOT be raw values, but the proportions within the ensemble. Hence, these definitions are essential to our ESE formulation.  We have updated the paper on Page 3 to emphasize this point.
>
> ---
>
> **We hope this clarification has addressed your concerns and better highlighted our contributions. We would be grateful if you could kindly reconsider the score. Thank you.**

---

### Official Review · Reviewer_QEQA · 2025-11-01

**Soundness:** 3
**Presentation:** 3
**Contribution:** 2
**Rating:** 4
**Confidence:** 4

**Summary:**

This paper proposes **Equilibrium State Estimation (ESE)** for simultaneous forecasting across many interacting systems (e.g., currencies or regions). Instead of forecasting each target independently, this paper estimates a shared **latent equilibrium** over systems from attributes, enforces proportionality/conservation constraints, and then allocates the ensemble trend back to each system. A damped iterative solver is used; cointegration tests provide a stopping criterion. For prediction, this paper either uses a parametric trend term multiplied by equilibrium proportions or plugs in an external forecaster to provide the ensemble trend. Experiments on synthetic dynamics, **G20 exchange rates**, and **COVID-19** incidence at multiple granularities show that ESE is competitive alone and often improves error when paired with strong baselines (ARIMA, LSTM, Informer, PatchTST, SCINet, etc.), while offering substantial runtime gains and scaling roughly linearly with systems, attributes, and timesteps.

**Strengths:**

- **Originality.** This paper casts multi-target forecasting as **equilibrium inference** over an ensemble of interacting systems, then distributes forecasts in one pass—distinct from multivariate series that treat variables within a single system. The explicit proportionality constraints and equilibrium-led allocation are conceptually clean.
- **Quality.** The framework specifies definitions, constraints, a **cointegration-based** stopping rule, and a clear prediction layer (Eq. 7) or a plug-in mode with external predictors (Eq. 8). The procedure is easy to integrate and analyze.
- **Clarity.** This paper carefully differentiates ESE from multivariate, multi-compartment, and multi-target settings, provides an algorithm box, and uses intuitive figures/tables across datasets and granularities.
- **Significance.** Reported results indicate strong **scalability** (empirical linear cost), competitive or improved accuracy, and **large speedups** when ESE is paired with SOTA forecasters—particularly as system count grows.

**Weaknesses:**

- **Assumptions.** Reliance on conservation-style proportion constraints and fixed ensemble membership may limit applicability where systems enter/exit or totals are non-conservative.
- **Attribute dependence.** Performance hinges on attribute quality/availability; robustness to noisy or missing attributes is not fully characterized.
- **Theory.** The cointegration threshold and damping are reasonable but heuristic; formal guarantees on consistency, bias, and convergence would strengthen the method.
- **Baselines & compatibility.** Some multivariate models (e.g., VAR) are not paired with ESE; a broader set of multi-target baselines would clarify fairness and generality.
- **Evaluation under shift.** More tests under distribution/regime shifts (e.g., shocks, policy changes) would build confidence in real-world deployment.

**Questions:**

1. **Constraint realism:** How does performance change if conservation is violated or if systems enter/exit? Could soft penalties replace hard constraints?
2. **Attribute robustness:** Please provide sensitivity to attribute noise/missingness and to the estimation of attribute effects; consider regularized or Bayesian estimators.
3. **Stopping & damping:** Ablate the cointegration threshold and damping schedule; is there an automated criterion that balances accuracy and stability?
4. **Shift robustness:** Evaluate under regime changes and non-stationary attributes; compare stability with multivariate baselines.
5. **Compatibility map:** Clarify which multi-target models can be paired with ESE and whether hybrids (e.g., VAR for trend, ESE for allocation) are feasible.
6. **Compute profiling:** Release code and report end-to-end wall-clock, memory, and parameter counts to substantiate linear scaling and speedups.

---

> ### Author Response · Authors · 2025-11-22
> **Response to Weaknesses 1 to 4**
>
> Dear Reviewer QEQA, we sincerely appreciate your recognition of the strengths of our work, specifically its originality, quality, clarity and significance. Below, please see our responses, particularly to clarify the concerns.
>
> >**W1.  Assumptions. Reliance on conservation-style proportion constraints and fixed ensemble membership may limit applicability where systems enter/exit or totals are non-conservative.**
>
> Similar to the response to ***W5 of Reviewer VFff***, the constraints are functional, not structural.  It is the mathematical formulation of ESE, just to ensure that attributes included in ESE are consistent across systems. We do not impose any limitations, e.g., excluding certain attributes or forcing the inclusion of others.
>
> The no-enter/exit assumption is also just for mathematical rigor, to ensure the composition of the ensemble remains consistent during a particular time window for calculation. Using a flea market as an example, where vendors come and go on a daily basis, such inconsistency in system composition is undesirable for predicting individual vendors' sales based on a 10-day window.
>
> The composition may change in the long run.  As shown in Table 6, ESE can work with both G7 and G20 compositions to predict exchange rates, even once the composition transitions between these two settings.  This G7/G20 example also shows that ESE can work with incomplete composition, as G7/G20 only represent about 32% and 75% of the entire global trade.
>
> >**W2. Attribute dependence. Performance hinges on attribute quality/availability; robustness to noisy or missing attributes is not fully characterized.**
>
> The analysis on robustness to **noisy attributes** is present in the main text (Table 4 "Robustness analysis of ESE", and Subsection "Robustness", Lines 419-425, Page 8),
>
> **Missing attributes:** Similar to the response to ***W5 of Reviewer VFff***, an attribute will be simply ignored if it is absent for a system. The table below shows that the impact of missing attributes on performance is rather small.
>
> || RMSE* (ESE) | RMSE* (DLinear+ESE)|
> |:-:|:-:|:-:|
> |No missing attributes |1.0972|0.9495|
> |Removing Germany Population|1.0976|0.9584|
> |Removing Germany M2|1.0121|0.9742|
> |Removing Germany Export|1.0120|0.9787|
>
> >**W3.  Theory. The cointegration threshold and damping are reasonable but heuristic; formal guarantees on consistency, bias, and convergence would strengthen the method.**
>
> We thank the reviewer for raising this point. In response, we wish to clarify that ESE already incorporates formal convergence guarantees in **Appendix D**.  The convergence condition is mathematically grounded in established cointegration theory—the iterative estimator converges precisely when a long-run equilibrium relationship is satisfied. The residual stationarity test used as the stopping criterion is a standard inferential procedure in statistics.
>
> The damping coefficient $\lambda$ is not a heuristic choice: it acts solely as a relaxation parameter to stabilize iterative updates and does not influence the final fixed point. The resulting equilibrium estimate depends entirely on the underlying cointegration structure, not the specific value of $\lambda$ in Algo.1.  Similar to the response to ***W3 of Reviewer VFff***, the convergence process requires a $\lambda$ no greater than 0.5.  A choice less than 0.5 will require more epochs and more time, yet still reach the same convergence point, as shown below.
>
> |$\lambda$ Value|Num. of Epochs|Time (sec)|
> |:-:|:-:|:-:|
> |0.5|14|9.37|
> |0.4|18|11.13|
> |0.3|21|12.44|
> |0.2|27|14.42|
> |0.1|36|16.61|
>
> Thus, $\lambda$=0.5 is the optimal choice. We will add the above in **Appendix D.3**.  In addition, we will update the paper to highlight another point: the consistency of the estimator itself follows from the maximum likelihood formulation given in Eq. (6).
>
> >**W4.  Baselines & compatibility. Some multivariate models (e.g., VAR) are not paired with ESE; a broader set of multi-target baselines would clarify fairness and generality.**
>
> Since VAR cannot perform prediction for a single target variable, it does not apply to Eq.(8), thus being unsuitable for direct comparison. However, there is a workaround: summing the predicted values from VAR and subsequently applying ESE to compute values for each individual system. Note that this approach involves a second transformation of the numerical values, hence was not included in the paper.  The results are shown below.  Another multivariate method Crossformer raised by **Reviewer P3FP** is also included.  This will be added in the updated manuscript.
>
> |G7||Predicted Values|RMSE*s (x 100)|
> |:-:|:-:|:-:|:-:|
> |Observed values||*1.37, 0.93, 0.79, 157.70*|-|
> |ESE||1.38, 0.93, 0.82, 157.17|1.10|
> |VAR|No ESE|1.38, 0.91, 0.77, 155.67|1.18|
> ||With ESE|1.37, 0.92, 0.81, 155.64|1.12|
> |Crossformer <univariate, with attributes>|No ESE|1.37, 0.92, 0.81, 159.14|1.12|
> |Crossformer <multivariate, with attributes>|With ESE|1.35, 0.91, 0.80, 153.65|1.80|

---

> ### Author Response · Authors · 2025-11-22
> **Response to Weakness 5 & Questions 1 - 6**
>
> >**W5. Evaluation under shift. More tests under distribution/regime shifts (e.g., shocks, policy changes) would build confidence in real-world deployment**
>
> We agree that robustness under shifts is essential. This is actually an advantage of our ESE. We would like to clarify that the manuscript already evaluates ESE under multiple forms of significant distributional change, and these experiments demonstrate that ESE remains stable across varying regimes.
> 1. *Real-world regime shifts: COVID19 case studies (**Table 3, Tables 21–24**)*: The COVID19 experiments naturally embody severe distributional shifts due to abrupt policy changes and infection waves. Despite these extreme non-stationarities, ESE consistently improves prediction accuracy over baselines, demonstrating resilience to real-world shocks.
> 2. *Attribute perturbation and misspecification (**Table 4 & Lines 419–425**)*: The noise injection experiments introduce substantial perturbations, representing a controlled distributional shift. ESE degrades smoothly rather than abruptly, further showing robustness.
> 3. *Regime shift (G7/G20 in **Table 6**)*: Table 6 evaluates ESE when the ensemble change between the relatively homogeneous but incomplete G7 countries and the more heterogeneous G20 group. ESE remains stable and effective under this major regime shift, indicating robustness to shifts in system relationships and compositions.
> ---
> >**Q1. Constraint realism: How does performance change if conservation is violated or if systems enter/exit? Could soft penalties replace hard constraints?**
>
> As detailed in our response to **W1**, these are functional, not structural, and simply ensure attribute consistency within a prediction window. ESE handles changing or incomplete ensembles well in practice, as demonstrated in G7/G20 compositions in Table 6. We will clarify that these assumptions support calculation but do not limit ESE’s applicability.
>
> >**Q2. Attribute robustness: Please provide sensitivity to attribute noise/missingness and to the estimation of attribute effects; consider regularized or Bayesian estimators**
>
> As explained in our response to **W2**, these analyses are in the main text (Table 4 and Lines 419–425), showing smooth degradation under substantial perturbations. For missing attributes, ESE simply ignores absent features, as shown in the table in W2.
>
> In addition, our model already possesses a form of built-in regularization. The cointegration test and the damping factor $\lambda$ prevent overfitting to spurious fluctuations in attributes. The adaptive weight $\psi_k$ acts as a natural selector, suppressing attributes with unstable relationships.
>
> >**Q3. Stopping & damping: Ablate the cointegration threshold and damping schedule; is there an automated criterion that balances accuracy and stability?**
>
> *Cointegration threshold* analysis can be found in **Appendix D, Lines 893-902**, which revealed that ESE is flexible with the threshold value, which has only a minor impact if it stays in the range around 0.05.
>
> *Damping schedule*: As noted in our response to **W3**, λ value 0.5 is the optimal choice as it ensures numerical stability. All λ ≤ 0.5 converge to the same equilibrium, with smaller values only increasing runtime but not affecting accuracy.
>
> >**Q4. Shift robustness: Evaluate under regime changes and non-stationary attributes; compare stability with multivariate baselines**
>
> As addressed in responses to **W4** and **W5**, ESE has been evaluated under multiple forms of shifts. These include real-world COVID19 shocks (**Table 3, Tables 21–24**), attribute perturbation (**Table 4 & Lines 419–425**), and structural changes in ensemble composition such as the G7/G20 regime shift (**Table 6**). Across all these scenarios, ESE remains stable and consistently outperforms baselines, including multivariate models, VAR and Crossformer.
>
> The main text also includes a dedicated comparison with multivariate baselines (**Table 5 & Lines 426-431**).
>
> >**Q5.Compatibility map: Clarify which multi-target models can be paired with ESE and whether hybrids (e.g., VAR for trend, ESE for allocation) are feasible**
>
> As stated in Line 239, "any prediction models can be integrated".  Multi-target models that do not support univariate prediction can also be combined with a workaround, as shown in the response to **W4**.
>
> >**Q6. Compute profiling: Release code and report end-to-end wall-clock, memory, and parameter counts to substantiate linear scaling and speedups**
>
> Code is released (see **Abstract Line 025**).  The runtimes in the paper are wall-clock time.  We will add memory usage, which increases marginally with the number of systems and attributes.
>
> |#Attributes|2|4|6|8|10|
> |:-:|:-:|:-:|:-:|:-:|:-:|
> |G7|141.93|141.57|141.9375|141.47|142.23|
> |G20|144.60|144.17|144.51|144.21|145.14|
> ---
> **We hope our response has clarified the confusion and addressed your concerns. We would greatly appreciate it if you could kindly reconsider your score. Thank you.**

---

### Official Review · Reviewer_VFff · 2025-11-01

**Soundness:** 3
**Presentation:** 3
**Contribution:** 3
**Rating:** 6
**Confidence:** 1

**Summary:**

The paper describes Equilibrium State Estimation, that jointly predicts the future states of multiple interacting systems. This work first estimates a latent "equilibrium state" for the entire ensemble of systems, which represents a balanced distribution of their target values based on their current attributes. The prediction is then generated based on the deviation of the current state from this estimated equilibrium, combined with the overall trend of the ensemble. The authors claim that it can provide significant speedup while maintaining or improving accuracy.

**Strengths:**

· The idea of using a estimated equilibrium state as a basis for multi-system forecasting is innovative.

· The paper's strongest empirical claim is the dramatic computational speedup. The linear complexity w.r.t. the number of systems, attributes, and time steps is clearly demonstrated in the complexity analysis.

· The appendices do a good job of distinguishing the proposed task from related but different concepts like multi-variate forecasting and multi-compartment models, which helps to precisely define the paper's contribution.

· The paper is overall well written with most concepts.

**Weaknesses:**

· The core assumption is that the ensemble of systems has a meaningful, estimable equilibrium state. While the intuition is grounded in concepts like Nash equilibrium, the direct application to non-adversarial, prediction-based tasks needs stronger theoretical justification. The paper would benefit from a more formal discussion on the existence and uniqueness of such an equilibrium in the contexts presented.

· It seems to me that the equilibrium state is estimated from the attributes, and then the prediction is made based on the deviation from the equilibrium state. However, the attributes are also the primary drivers of the system's evolution. There is a risk of a circular argument where the model is effective simply

because it's using the attributes to create a target (the equilibrium) and then predicting towards it. A deeper ablation study or analysis is needed to disentangle the contribution of the equilibrium concept from simply using the attributes for a clever form of proportional forecasting.

· The initialization of equilibrium state and the choice of the damping coefficient=0.5 seem arbitrary. A sensitivity analysis or a justification for these choices is missing. How sensitive are the final results to these initial conditions and hyperparameters?

· The paper lacks a crucial ablation study. What is the performance of the "Predictor" component (Eq. 7) alone, using a simple or naive weighting (e.g., last observed proportion) instead of the estimated equilibrium state? Appendix N compares to a naive weighting, but this should be integrated into the main experiments to isolate the value added by the complex equilibrium estimation process itself.

· The method may not be generalized. The constraints (Definitions 1-3) are quite strict: all systems must have the same attribute set, and the sum of proportions must be 1. This limits the applicability of ESE. How would it perform if some systems had missing attributes? Or if the ensemble was not "closed"? The G7 vs. G20 analysis (Table 6) hints at this but doesn't fully address the brittleness of the method to its core assumptions.

**Questions:**

Please see the weaknesses and address them.

---

> ### Author Response · Authors · 2025-11-21
> **Response to Weaknesses 1 - 2**
>
> Dear Reviewer VFff, we would like to express our gratitude for your support.  Below, please see our responses, particularly to clarify **a major misunderstanding**.
>
> >**W1: ... the intuition is grounded in concepts like Nash equilibrium, the direct application to non-adversarial, prediction-based tasks needs stronger theoretical justification. ... a more formal discussion on the existence and uniqueness of such an equilibrium in the contexts presented.**
>
> Thank you for this thoughtful observation. We would like to clarify that a game-theoretic equilibrium, e.g., Nash equilibrium, does require formal existence and uniqueness guarantees, however, ESE uses a local, data-derived equilibrium that is re-estimated at every time window.  So it **does not assume or depend on** a global, unique, or game-theoretic equilibrium.
>
> Nevertheless, a formal examination of equilibrium states in ESE is included. **Appendix D** shows the mathematical description of the equilibrium, the validation criterion, and the convergence analysis. These analyses show that the equilibrium used in ESE is empirically estimable, time-varying, and does not require global uniqueness or persistence.
>
> More importantly,  **Appendix O – Evaluating True Equilibrium States** is dedicated to formal analysis of true equilibrium states.  We explicitly test whether enforcing a true global equilibrium improves predictive performance or not. The results show that:
> - the true/global equilibrium does not yield better predictions,
> - enforcing a fixed equilibrium can even degrade performance, and
> - allowing the equilibrium to be local and time-varying, as done in ESE, is consistently more effective.
>
> To make this distinction clearer, we will update the main text to emphasize that the equilibrium concept in ESE is statistical not game-theoretic, and mention Appendix O early, as the current version refers to it on Page 9, in the footnote "*Additional analyses on weighting strategy and true equilibrium are viewable in Appendix N and O*".
>
> >**W2: ... There is a risk of a circular argument where the model is effective simply because it's using the attributes .... A deeper ablation study or analysis is needed to disentangle the contribution of the equilibrium concept from simply using the attributes for a clever form of proportional forecasting.**
>
> Thank you for this thoughtful concern. We clarify that ESE does not use attributes in a circular manner. The equilibrium in ESE means a statistically validated, residual-based construct, not a direct transformation of attributes into a target. Several parts of the paper, including Appendices D, N, and O, explicitly disentangle the contribution of ESE from simple attribute-driven forecasting.
>
> 1. *Appendix N directly evaluates naive vs. adaptive weighting:* **Appendix N** specifically addresses this concern. It compares ESE against naive weighting schemes that simply use attributes or heuristic combinations. The results show that:
> - naive attribute-based weighting performs substantially worse,
> - adaptive equilibrium weighting yields consistent improvement,
> - demonstrating that ESE’s gains do not arise from cleverly reweighting attributes.
>
> 2. *Additional ablations demonstrate non-circularity*: Beyond Appendix N, several ablations in the paper show that ESE's improvements do not come from simply turning attributes into a target:
> - the study on attribute noise (Lines 419–425; Table 4) shows that substantial perturbations to attributes produce only modest degradation, inconsistent with a model that relies directly on attributes for target construction.
> - the analysis on cointegration threshold (Appendix D.2) indicates that ESE behaves smoothly under threshold changes, indicating no direct attribute usage.
> - the analysis on true equilibrium (Appendix O) illustrates that a "true equilibrium" does not improve performance, confirming that the benefit comes from ESE’s dynamically estimated equilibrium, not simplistic mapping of attributes to the target.
>
> 3. *The equilibrium is statistically validated, not attribute-defined:* While attributes participate in estimating the local equilibrium, the equilibrium state is accepted only if a residual-based test is passed (Appendix D, Lines 818–832). If an attribute does not form a stable long-run relationship with the target, the test fails and the adaptive equilibrium weight $\psi_k$ would collapse toward zero (Lines 211–212), removing the attribute from influencing the prediction. This mechanism prevents circularity: attributes cannot "force" the target unless they statistically support a stable, equilibrium-forming relationship.
>
> To make this point clearer and avoid similar confusion, we will revise the manuscript to highlight that: (1) ESE’s equilibrium is not an attribute-derived proxy target, (2) naive attribute weighting does not reproduce ESE’s behavior (Appendix N), and (3) the equilibrium contributes structural information not present in attributes alone.

---

> ### Author Response · Authors · 2025-11-21
> **Response to Weaknesses 3 - 5**
>
> >**W3: The initialization of equilibrium state and the choice of the damping coefficient=0.5 seem arbitrary. A sensitivity analysis or a justification for these choices is missing....**
>
> Thank you for raising this. We clarify that both the initialization and the choice of $\lambda$=0.5 are not arbitrary, but grounded in the ESE’s iterative update rule.
>
> The initialization does not materially affect convergence or predictive performance, because ESE recomputes the equilibrium in every local window. After a few epochs, the equilibrium estimate is entirely determined by (1) the cointegration-validated equilibrium regression, and (2) the recursive update rule.
>
> As for the damping coefficient being 0.5, the choice follows directly from the stability constraints of ESE’s iterative update in Line 4 of Algorithm 1:  $\mathcal{L}^{[e]} ← \mathcal{ES}\_{t}^{[e]} - \mathcal{ST}\_{t'} + (\mathcal{L}^{[e]}) \cdot \lambda$.  In this update, the term $\mathcal{ES}\_{t}^{[e]} - \mathcal{ST}\_{t'} $ has a maximum magnitude of 1 due to normalization. Therefore:
> - If $\lambda$ >0.5,  the recursive update may grow in magnitude rather than contract, causing $L^{[e]}$ to diverge instead of converge.
> - If $\lambda$ <0.5, the update becomes conservative, slowing down convergence unnecessarily.  Below is an empirical comparison of different $\lambda$ values under 0.5. Smaller $\lambda$ values take longer to converge, increasing total computation time without affecting the final equilibrium, as all settings converged to the same final state.
>
> |$\lambda$ Value |Num. of Epochs|Time (sec)|
> |:-:|:-:|:-:|
> |0.5|14|9.37|
> |0.4|18|11.13|
> |0.3|21|12.44|
> |0.2|27|14.42|
> |0.1|36|16.61|
>
> Thus, $\lambda$=0.5 is the maximum stable damping factor that still ensures efficient convergence. We will add the above in **Appendix D.3**.
>
> >**W4: The paper lacks a crucial ablation study. What is the performance ... using a simple or naive weighting ...? Appendix N compares to a naive weighting, but this should be integrated into the main experiments ...**
>
> Thank you for this valuable suggestion. We agree that isolating ESE's contribution is important. In fact, the paper already includes such an ablation in Appendix N, which shows that naive weights perform substantially worse than equilibrium estimation, as they fail to capture meaningful long-run relationships. The gains of ESE cannot be reproduced by simply applying the Predictor with proportional weighting.
>
> In the updated version, we will summarize the key results of Appendix N in the main paper and clarify more explicitly ESE's contribution through this ablation of isolating equilibrium estimation.
>
> >**W5: The constraints (Definitions 1-3) are quite strict: all systems must have the same attribute set, and the sum of proportions must be 1. This limits the applicability of ESE. How would it perform if some systems had missing attributes? Or... not "closed"? ... (Table 6) hints at this but doesn't fully address the brittleness of the method to its core assumptions.**
>
> We fully agree that generalizability is important. In fact, ESE is designed to be generalizable, flexible, and robust.
>
> 1. *The constraints are functional, not structural requirements:* Definitions 1–3 specify the mathematical formulation of ESE to ensure that attributes included in ESE are consistent and comparable across systems.  They facilitate the calculation and do not impose structural limitations, e.g., excluding certain attributes or forcing the inclusion of others.
>
> 2. *ESE tolerates missing attributes.* ESE downweights attributes that are noisy or absent. When an attribute is absent for a system, it is simply ignored.  The study below shows that the impact on performance is small and depends on the relevance of the attribute. Population (Table 15) is weakly related to exchange rates, hence its absence produces minimal change. M2 and Export are more relevant, but even removing them does not collapse performance, further demonstrating ESE's robustness.
>
> || RMSE* (ESE) | RMSE* (DLinear+ESE)|
> |:-:|:-:|:-:|
> |No missing attributes |1.0972|0.9495|
> |Removing Germany Population |1.0976|0.9584|
> |Removing Germany M2|1.0121|0.9742|
> |Removing Germany Export|1.0120|0.9787|
>
> 3. *The proportion-sum constraint (summing to 1) is simply a normalization step that prevents numerical bias*. It is not a domain limitation and does not restrict the kinds of systems or attributes ESE can handle.
>
> 4. *ESE is not brittle on not-closed ensembles*: Table 6 (G7 vs. G20) demonstrates ESE performs well even when the ensemble is more heterogeneous (G20) and more open (G7), indicating that ESE remains stable at different settings.  Furthermore, Table 3 and Tables 21 -24 on COVID prediction show ESE can work with different granularities as well.
>
> ---
>
> **Hope our responses have clarified all confusions and adequately addressed your concerns. We would greatly appreciate if you could kindly increase your score. Thank you.**

---

> > ### Comment · Reviewer_VFff · 2025-11-25
> >
> > Thanks for the response and answering the question in detail.
> >
> > I will maintain my positive score.

---

### Official Review · Reviewer_P3FP · 2025-11-03

**Soundness:** 3
**Presentation:** 3
**Contribution:** 1
**Rating:** 2
**Confidence:** 5

**Summary:**

The paper introduces a novel framework called Equilibrium State Estimation (ESE) for simultaneous forecasting across multiple interdependent systems. Each system’s relative proportion to the ensemble total is modeled via an equilibrium state ES_t, estimated iteratively until cointegration between observed and equilibrium series is achieved. Forecasts from any base model (e.g., ARIMA, LSTM, Informer) are redistributed according to these equilibrium proportions, ensuring global consistency.

The approach is tested on synthetic data and two real-world problems: FX rate forecasting (16 currencies) and COVID-19 infection forecasting (320 regions in Australia). Results show improved accuracy and computational efficiency when ESE is added to baseline univariate models.

**Strengths:**

1. Interpretability: ESE yields equilibrium proportions interpretable as long-run relationships among series (e.g., currency parity, infection equilibrium).
2. Scalability: Computational efficiency is a strong advantage; ESE avoids large multivariate training and scales linearly in the number of systems.
3. Consistency: The paper demonstrates consistent accuracy gains and significant runtime reduction compared to single-model baselines.
4. Clarity: Formulations are precise, and algorithms are clearly presented with explicit steps and convergence tests.

**Weaknesses:**

1. Restricted baseline comparison: The method is compared mostly against univariate models. No experiments include multivariate or cointegration-based systems (VAR, VECM, DeepVECM, Crossformer), making it unclear whether ESE offers an advantage beyond equilibrium-aware models.
2. Static equilibrium assumption: ESE treats equilibrium adjustment as static. It does not learn or adaptively correct deviations over time (e.g., through error-correction dynamics), limiting performance in regime-switching or non-stationary settings.
3. Lack of theoretical guarantees: No formal convergence or consistency results are provided for the iterative equilibrium estimation process.

**Questions:**

Could the authors clarify how sensitive the ESE procedure is to misspecified attributes or cointegration test thresholds, whether it can be extended with dynamic or learned error-correction mechanisms such as neural ECM, and provide a clear theoretical justification with conditions under which ESE offers advantages over multivariate time-series models like VAR or VECM?

---

> ### Author Response · Authors · 2025-11-20
> **Clarifications on Weaknesses 1 to 3**
>
> Dear Reviewer P3FP, please see our responses below, which clarify a few points of confusion regarding our ESE method.
>
> >**W1: Restricted baseline comparison: ... No experiments include multivariate or cointegration-based systems (VAR, VECM, DeepVECM, Crossformer), making it unclear whether ESE offers an advantage beyond equilibrium-aware models.**
>
> **VAR**: we respectfully clarify that VAR is included in our experiments. It appears as is the first non-ESE method in Table 2, Table 3, Table 16, Table 17, Table 21, Table 22, Table 23, and Table 24. Moreover, we devoted one subsection to "Multi-variate Prediction" (Section 4.2, Lines 426-438 and Table 5 on Pages 8-9), where the advantages of ESE over multivariate approaches are clearly demonstrated.
>
> **VECM**, Vector Error Correction Model, is a derivative of VAR, with the prerequisite that all target time series must satisfy the cointegration assumption. If the target time series are not cointegrated, the error correction term becomes undefined and the correction mechanism cannot operate [1][2]. Our ESE approach does not rely on this prerequisite, nor is conintegration enforced in our datasets.  Therefore VECM is not suitable for direct comparison.
>
> Regarding **DEEP VECM**, we have been unable to locate the relevant literature. Could you kindly provide a reference, implementation, or tool in which it is embedded?
>
> **Crossformer**: thank you for raising this.  On our data, it is not as accurate as other baseline methods, e.g. Dlinear and PatchTST (see the table below).  Moreover, we evaluated three scenarios of Crossformer predicting exchange rates of G7 currencies, CAD, GBP, EUR, and JPY: (1) multiple single-variate predictions; (2) multivariate prediction without using attribute data; (3) multivariate with attribute data.  None of these scenarios showed superior performance.  Nevertheless, we will consider adding Crossformer to the main results.
>
> |G7 Currencies||Predicted Values| RMSE*s (x100)|Time (sec)|
> |:-:|:-:|:-:|:-:|:-:|
> |Observed values||*1.37, 0.93, 0.79, 157.70*|-|-|
> |ESE||1.38, 0.93, 0.82, 157.17|1.10|**9.38**|
> Dlinear|No ESE|1.38, 0.94, 0.81, 158.34|1.10|37.88|
> ||With ESE|1.38, 0.94, 0.81, 158.30|**0.95**|21.57|
> PatchTST |No ESE|1.37, 0.94, 0.81, 158.13|1.09|80.52|
> ||With ESE|1.38, 0.94, 0.80, 158.60|1.04|30.55|
> Crossformer (1)|No ESE|1.37, 0.92, 0.81, 159.14|1.12|129.28|
> |<multi-prediction, with attributes>|With ESE|1.40, 0.94, 0.82, 159.10|2.13|139.13|
> |Crossformer (2)|No ESE|1.43, 0.86, 0.74, 149.78|5.81|23.65|
> |<multi-variate, without attributes>|With ESE|1.32, 0.89, 0.78, 149.83|3.82|33.79|
> |Crossformer (3) |No ESE|1.41, 0.82, 0.86, 153.62|6.54|172.32|
> |<multi-variate, with attributes>|With ESE|1.35, 0.91, 0.80, 153.65|1.80|182.24|
>
> [1] Robert F. Engle and C. W. J. Granger, Econometrica, Vol. 55, No. 2 (1987), pp. 251-276
>
> [2] Gianluca Cubadda, Marco Mazzali, The vector error correction index model: representation, estimation and identification, The Econometrics Journal, Vol. 27, Issue 1, Jan 2024, pp. 126–150
>
> >**W2: Static equilibrium assumption: ESE treats equilibrium adjustment as static. It does not learn or adaptively correct deviations over time ..., limiting performance in regime-switching or non-stationary settings.**
>
> We appreciate your observation but respectfully clarify that **ESE does not require** a static equilibrium assumption. In ESE, the "equilibrium" at time $t$ is not a fixed global point, but a local point derived from the current data window. As time progresses and new data arrive, ESE re-estimates the equilibrium $\gamma^*_{1:n, t}$ for the new $t$, allowing it to evolve dynamically. Hence, ESE is fully compatible with non-stationary environments and does not impose any static-equalibrium restriction. We are now clarifying this point in the manuscript to avoid further confusion.
>
> >**W3: Lack of theoretical guarantees: No formal convergence or consistency results are provided for the iterative equilibrium estimation process.**
>
> We thank you for the comment. However, the claim that "*no formal convergence or consistency results are provided*" does not reflect what is already included in the manuscript.  Line 216 states "*The convergence analysis and the cointegration equations are in Appendix D*". **Appendix D - Cointegration and Convergence in ESE**, provides a formal convergence analysis, including the progression of p-values at each epoch (D.1) and the impact of different p-value thresholds (D.2) .
>
> Lines 818–832 detail the cointegration equation and the residual-based stationary test used to verify long-run equilibrium. The mathematical foundation of the convergence criterion used in ESE is defined there (Line 820): $\mathcal{ST}_t = \beta_0 + \beta_1 \mathcal{ES}_t^{[e]} + \epsilon_t$.
>
> In addition, Lines 211-212 explain that convergence will fail when $\psi_t$=0, which effectively disables the influence of all attributes in Eq. 5. This provides a theoretical characterization of the convergence conditions.

---

> ### Author Response · Authors · 2025-11-20
> **Response to the Questions**
>
> >**Q.1: Could the authors clarify how sensitive the ESE procedure is to misspecified attributes or cointegration test thresholds**
>
> ESE is designed to be robust to imperfect or noisy attributes. As shown in **Appendix D**, the convergence mechanism adaptively evaluates cointegration strength at each epoch through residual-based stationarity tests. When an attribute $k$ is weakly informative or misspecified, the estimated equilibrium coefficient of that attribute $\psi_k$ is adjusted accordingly within each local window, reducing its influence in the equilibrium estimation.
>
> Lines 419-425 and Table 4 show our experiments on attribute misspecification by adding perturbations.  The impact on performance is rather limited, illustrating that ESE is not sensitive to substantial noise injected into multiple attributes.
>
> Regarding cointegration thresholds, **Appendix D.2** systematically examines how different p-value thresholds of cointegration impact the performance. The results show smooth degradation rather than abrupt failure, indicating that ESE is not sensitive to the specific threshold chosen. Because ESE does not rely on a single optimal cutoff, it avoids overfitting to any particular threshold.
>
> >**Q.2: whether it can be extended with dynamic or learned error-correction mechanisms such as neural ECM**
>
> As noted in our response to **W2**, ESE does NOT rely on a static or globally fixed equilibrium.  Instead, it estimates a local, time-varying equilibrium from each data window and updates it dynamically as new data arrive. This design already aligns ESE with the spirit of dynamic error-correction.
>
> Regarding the possibility of incorporating neural ECM, we have conducted a focused literature search but have been unable to locate a standard or widely adopted formulation of a "*neural ECM*" in the cointegration/error-correction modeling literature. We would be grateful if the reviewer could share a reference or source.
>
> That said, the ESE framework can conceptually support learned ECM as extensions. Because ESE produces a time varying equilibrium estimate and corresponding residuals, these components could serve as supervision targets or correction signals for a learned nonlinear adjustment module. For example, a neural network could model a nonlinear mapping from recent observations, or from the estimated residuals, to short term corrective updates. Exploring such learned correction mechanisms could be an interesting extension. However, it is not directly related to the core contribution of ESE, thus falls outside the scope of this study.
>
> >**Q.3: and provide a clear theoretical justification with conditions under which ESE offers advantages over multivariate time-series models like VAR or VECM?**
>
> As clarified earlier, VAR is included in our experiments, where it consistently underperforms ESE across Tables 2–3, 16–17, and 21–24. Section 4.2 (Lines 426–438; Table 5) further shows that ESE offers advantages in multivariate settings.  The theoretical reason is that VAR assumes a fixed linear dependency among all series. Its parameters are constant across the entire sample and require stationarity of the multivariate vector.
>
> In contrast, ESE does not assume stationarity.  It re-estimates a local equilibrium at each time window, thus adapting to time-varying, non-stationary, or regime-switching environments.  In addition, attribute relevance is adaptive.  The coefficient $\psi_k$ modulates the influence of attributes, yielding a well-defined theoretical condition under which convergence will fail when all $\psi$ collapse to zero (Lines 211–212).  Therefore, whenever the equilibrium relationships evolve over time or exhibit local patterns not captured by a VAR structure, ESE is theoretically favored.
>
> As stated in our response to **W1**, VECM requires cointegration across all target series. If cointegration does not hold, the error-correction term is undefined and VECM would not work.  Our data of multiple systems do not enforce cointegration; hence VECM is not applicable.
>
> In comparison, ESE does not require cointegration and is robust to noisy attribute data. Thus, ESE has theoretical advantages when cointegration is weak or absent, but local equilibrium tendencies exist.
>
> ---
>
> We sincerely appreciate your recognition of the strengths of our work, specifically its interpretability, scalability, consistency, and clarity.
>
> **We hope that our responses have clarified the misunderstandings and fully addressed your concerns. We would greatly appreciate your reconsideration of the score. Thank you.**

---

### Comment · Area_Chair_vUcw · 2025-11-26
**Reminder to Engage!**

Dear Reviewers,

We are one week away from the end of the discussion period and the review responses have been posted. If you have not done so already, please read the response and check if the authors have addressed your concerns. Also please acknowledge the review by responding and stating how the response (and updated manuscript if provided) does or does not change your evaluation of the work. Earlier responses allow for meaningful engagement and potential for further clarification.

-Area Chair

---

### Meta-Review · Area_Chair_kns8 · 2026-01-07

**Summary:**

The paper introduces Equilibrium State Estimation (ESE), a framework for jointly predicting time-series data drawn from multiple interacting systems. The initial reviews were mixed. While most reviewers found the core idea promising, they raised several concerns, some of which were admittedly not fully well grounded. Two issues were consistently noted across multiple reviewers: (1) the validity of the underlying equilibrium assumption, and (2) the lack of clarity in the presentation of the technical components.

In the rebuttal, the authors provided additional clarification of the equilibrium assumption and revised the paper to improve overall presentation. However, the AC found that presentation issue still outstanding.

After reviewing the paper, the reviews, and the rebuttal, and in light of the overall lukewarm ratings, the AC recommends rejection. This decision is driven largely by concerns about the clarity and presentation of the paper, which make it difficult for the broader ML community to comprehend the technical contributions.

**Reviewer Concerns:**

Unfortunately, the topic of this paper falls outside the primary expertise of the AC, as the paper was reassigned following the ICLR incident. As a result, it is difficult for the AC to independently assess whether the rebuttal fully addresses the validity of the equilibrium assumption. The AC has some basic knowledge about Nash equilibrium and recently worked on time-series modeling, yet found the paper very challenging to follow. Although the mathematical formulation itself is relatively simple, the writing assumes substantial prior knowledge about the topic, and the introduction of key concepts does not follow the way commonly used in the ML community. It is likely many review comments indeed stem from the presentation of the paper. The AC thus concluded that the lack of clarity is still outstanding.

**Reviewer Scores:**

The AC also found it difficult to predict how the reviewers will update their scores. To the AC's best knowledge, Reviewer kPHS, who primarily complained about the presentation, is unlikely to change the current negative rating (2). Reviewer VFff, who is most positive about the paper (current rating 6), has the lowest confidence. The remaining reviewers did not provide particularly constructive comments.

---

### Decision · Program_Chairs · 2026-01-26

Reject